# Lipid peroxidation and type I interferon coupling fuels pathogenic macrophage activation causing tuberculosis susceptibility

Shivraj M Yabaji[1], Vadim Zhernovkov[2], Prasanna Babu Araveti[1], Suruchi Lata[1], Oleksii S Rukhlenko[2], Salam Al Abdullatif[3], Arthur Vanvalkenburg[4,5], Yuriy O Alekseyev[6], Qicheng Ma[7], Gargi Dayama[7], Nelson C Lau[1,7], W Evan Johnson[4,5], William R Bishai[8], Nicholas A Crossland[1,6], Joshua D Campbell[3], Boris N Kholodenko[2,9,10], Alexander A Gimelbrant[11], Lester Kobzik[12], Igor Kramnik[1,13,14]*

[1]The National Emerging Infectious Diseases Laboratory, Boston University, Boston, United States; [2]Systems Biology Ireland, School of Medicine, University College Dublin, Dublin, Ireland; [3]Department of Medicine, Boston University Chobanian and Avedisian School of Medicine, Boston, United States; [4]Rutgers University, New Jersey Medical School, Center for Data Science, New Brunswick, United States; [5]Department of Medicine, Division of Infectious Disease, New Jersey Medical School, Rutgers University, New Brunswick, United States; [6]The Department of Pathology and Laboratory Medicine, Boston University Chobanian Avedisian School of Medicine, Boston, United States; [7]Department of Biochemistry, and Cell Biology and Genome Science Institute, Boston University Chobanian and Avedisian School of Medicine, Boston, United States; [8]Center for TB Research, Johns Hopkins School of Medicine, Baltimore, United States; [9]Conway Institute of Biomolecular and Biomedical Research, University College Dublin, Dublin, Ireland; [10]Department of Pharmacology, Yale University School of Medicine, New Haven, United States; [11]Altius Institute for Biomedical Sciences, Seattle, United States; [12]Cellecta, Inc., Mountain View, United States; [13]Pulmonary Center, The Department of Medicine, Boston University Chobanian and Avedisian School of Medicine, Boston, United States; [14]Dept. of Microbiology, Boston University Chobanian and Avedisian School of Medicine, Boston, United States

*For correspondence:
ikramnik@bu.edu

## eLife Assessment

Yabaji et al. reports a **fundamental** study highlighting the mechanistic connection for susceptibility to TB infection via the sst1 locus, this was shown to involve increased IFN and Myc production causing the down-regulation of anti-oxidant defence genes and chronic lipidation. Ultimately, lipid peroxidation may underlie infectivity and macrophage dysfunction. Overall, the data presented are **compelling**, supported by a well designed multi-omics approach and the findings will be of broad interest to researchers investigating the molecular mechanisms of TB infection.
[Editors' note: this paper was reviewed by Review Commons.]

**Abstract** A quarter of the human population is infected with *Mycobacterium tuberculosis*, but less than 10% of those infected develop pulmonary TB. We developed a genetically defined sst1-susceptible mouse model that uniquely reproduces a defining feature of human TB: the development of necrotic lung granulomas and determined that the sst1-susceptible phenotype was driven by the aberrant macrophage activation. This study demonstrates that the aberrant response of the sst1-susceptible macrophages to prolonged stimulation with TNF is primarily driven by conflicting Myc and antioxidant response pathways leading to a coordinated failure (1) to properly sequester intracellular iron and (2) to activate ferroptosis inhibitor enzymes. Consequently, iron-mediated lipid peroxidation fueled superinduction of Ifnβ and sustained the type I interferon (IFN-I) pathway hyperactivity that locked the sst1-susceptible macrophages in a state of unresolving stress and compromised their resistance to Mtb. The accumulation of the aberrantly activated, stressed, macrophages within the granuloma microenvironment led to the local failure of anti-tuberculosis immunity and tissue necrosis. The upregulation of the Myc pathway in peripheral blood cells of human TB patients was significantly associated with poor outcomes of TB treatment. Thus, Myc dysregulation in activated macrophages results in an aberrant macrophage activation and represents a novel target for host-directed TB therapies.

## Introduction

Thousands of years of co-evolution with modern humans have made *Mycobacterium tuberculosis* (Mtb) arguably the most successful human pathogen (*Orgeur and Brosch, 2018*). It currently colonizes approximately a quarter of the global population (*World Health Organization, 2022*). Most Mtb-infected people develop latent TB, in which the host responses either eliminate or sequester the bacteria inside a granuloma structure (*Pagán and Ramakrishnan, 2018*; *Pai et al., 2016*). However, about 5–10% of Mtb-infected individuals will eventually develop active TB either within a year after the primary infection or later in their life after re-activation of persistent bacteria or re-infection (*Behr et al., 2021*; *Drain et al., 2018*; *Horsburgh and Rubin, 2011*; *Reichler et al., 2018*). A plethora of genetic, developmental, and environmental factors contribute to TB progression in individuals that initially resisted the pathogen (*Cohen et al., 2022*; *Simmons et al., 2018*).

Infection of the lung is central to Mtb's evolutionary success because it allows the pathogen to spread among human hosts via aerosols. After systemic dissemination from primary lesions, Mtb can be found in many human organs (*Bussi and Gutierrez, 2019*; *Ulrichs et al., 2005*). However, approximately 85% of the disease develops in the lungs (*Pai et al., 2016*). Necrotic lesions are the major pathologic manifestation of pulmonary TB ranging from central necrosis in organized granulomas during primary TB to massive necrotizing pneumonia and the formation of cavities in post-primary pulmonary TB (*Hunter, 2011*; *Pai et al., 2016*). The necrotic lung lesions develop in immunocompetent hosts despite the presence of active T-cell-mediated immune response (*Cohen et al., 2022*).

Existing mechanistic concepts explaining the lesion necrosis fall into two main categories: (i) *inadequate* local immunity that allows exuberant bacterial replication and production of virulence factors that drive tissue necrosis, vs. (ii) *excessive* effector immunity that results in immune-mediated tissue damage (*Ernst, 2018*; *O'Garra et al., 2013*; *Ogongo and Ernst, 2023*). Although both scenarios are credible, they are mechanistically distinct and would require different therapeutic strategies. Therefore, in-depth understanding of mechanisms of pulmonary TB progression in immunocompetent hosts is necessary for accurate patient stratification and for the development of personalized approaches to immune modulation (*DiNardo et al., 2021*).

Mouse models have been successfully used for mechanistic studies of Mtb infection, although classical inbred mouse strains routinely used in TB research, such as C57BL/6 (B6) and BALB/c, do not develop human-like necrotic TB lesions (*Apt and Kramnik, 2009*). Nevertheless, even in these models, Mtb predominantly replicates in the lungs irrespective of the route of infection (*North and Jung, 2004*). Mouse models that recapitulate the necrotization of pulmonary TB lesions have also been developed (reviewed in *Kramnik and Beamer, 2016*). We have previously found that C3HeB/FeJ mice develop necrotizing granulomas after infection with virulent Mtb and mapped several genetic loci of TB susceptibility using a cross of the C3HeB/FeJ with the resistant B6 mice (*Kramnik et al., 1998*; *Kramnik et al., 2000*; *Yan et al., 2006*). A single locus on chromosome 1, *sst1* (super-susceptibility to tuberculosis 1), was specifically responsible for the control of the necrotization of TB

lesions (*Sissons et al., 2009*). The *sst1*-susceptible mice develop necrotic lung lesions irrespective of the route of infection – aerosol, intravenous, or intradermal (*Kramnik et al., 1998*; *Kramnik et al., 2000*; *Pan et al., 2005*; *Pichugin et al., 2009*; *Yabaji et al., 2025b*) – thus demonstrating a common underlying mechanism.

We further found that the *sst1* locus primarily controls innate resistance to intracellular pathogens (*Boyartchuk et al., 2004*; *He et al., 2013*; *Pan et al., 2005*; *Yan et al., 2007*), and the *sst1*-mediated susceptibility was associated with the hyperactivity of the type I interferon (IFN-I) pathway in activated macrophages in vitro and in vivo (*Bhattacharya et al., 2021*; *He et al., 2013*; *Ji et al., 2019*). Of note, the hyperactivity of the IFN-I pathway has been associated with TB susceptibility and the disease progression both in human patients and experimental models (*Donovan et al., 2017*; *Moreira-Teixeira et al., 2018*; *O'Garra et al., 2013*; *Stanley et al., 2007*). However, mechanisms that underlie the IFN-I hyperactivity and their roles in susceptibility to TB were insufficiently elucidated. Thus, the *sst1*-mediated susceptibility recapitulates the morphologic and mechanistic hallmarks of human TB disease and provides a genetically defined mouse model to study both the upstream mechanisms responsible for the IFN-I pathway hyperactivity (*Bhattacharya et al., 2021*) and its downstream consequences (*Kotov et al., 2023*).

The *sst1* locus encodes the Sp110 (*Pan et al., 2005*) and Sp140 (*Ji et al., 2021*) proteins – known as interferon-inducible chromatin binding proteins (*Fraschilla and Jeffrey, 2020a*). The expression of mRNAs encoding both proteins is greatly diminished in mice that carry the *sst1* susceptibility allele, and protein expression is undetectable for both (*Bhattacharya et al., 2021*; *Ji et al., 2021*). Both proteins were shown to be involved in regulation of type I interferon pathway (*Ji et al., 2021*; *Lee et al., 2013*). The overexpression of Sp110b in macrophages increased their resistance to intracellular bacteria in vitro (*Pan et al., 2005*), while the *Sp140* gene knockout dramatically increased the mouse susceptibility to several intracellular bacteria including virulent Mtb (*Ji et al., 2021*). It was suggested that Sp140 plays a dominant role in TB susceptibility, via direct regulation of the interferon beta gene (*Ifnb1*) mRNA stability (*Witt et al., 2024*). In humans, the *Sp140* hypomorphic alleles were associated with susceptibility to multiple sclerosis and Crohn's disease (*Matesanz et al., 2015*; *Mehta et al., 2017*). Moreover, the *Sp140* mutations predicted the responsiveness of Crohn's disease patients to anti-TNF therapy, suggesting its role in regulating TNF response. Mechanistically, Sp140 was implicated in silencing 'lineage-inappropriate' and developmental genes, maintenance of heterochromatin in activated macrophages (*Mehta et al., 2017*) and downregulating transcriptional activity by inhibiting topoisomerases (*Amatullah et al., 2022*).

Previously, we found that prolonged TNF stimulation of the *sst1*-susceptible B6.Sst1S macrophages in vitro uniquely induced an aberrant response that was characterized by the *Ifnb1* super-induction and a coordinated upregulation of interferon-stimulated genes (ISGs), markers of proteotoxic stress (PS), and the integrated stress response (ISR). The upregulation of all these pathways was prevented by a reactive oxygen scavenger (BHA; *Bhattacharya et al., 2021*), suggesting that oxidative stress was driving the aberrant activation of the *sst1*-susceptible macrophages. Current literature provides ample evidence of the crosstalk between the above pathways: (1) stress kinase activation by oxidative stress (*Blaser et al., 2016*; *Kamata et al., 2005*); (2) promotion of type I interferon (IFN-I) responses by stress kinases (*Boccuni et al., 2022*; *Buskiewicz et al., 2016*; *Karin and Gallagher, 2005*); (3) suppression of AOD by IFN-I (*Lei et al., 2021*; *Riedelberger et al., 2020*); (4) suppression of IFN responses and AOD by Myc (*Levy and Forman, 2010*; *Torti and Torti, 2002*; *Zimmerli et al., 2022*; *Figure 1A*). However, the dynamic interactions and regulatory dependencies of these pathways in homeostasis and disease-specific contexts remain poorly understood.

In this study, we specifically addressed connectivity of the AOD and IFN-I pathways in *sst1*-susceptible macrophages persistently activated by TNF, a cell state relevant to TB granuloma microenvironment (*Flynn and Chan, 2022*). We determined that their aberrant response to prolonged TNF stimulation was primarily fueled by conflicting Myc and anti-oxidant responses leading to *Ifnb1* superinduction and the IFN-I pathway hyperactivity that locked the *sst1*-susceptible macrophages in a state of persistent oxidative stress. This unresolving stress compromised the macrophage resistance to virulent Mtb in vitro, and the accumulation of stressed macrophages within TB lesions was associated with the failure of Mtb control within pulmonary TB lesions and their necrotization in vivo.

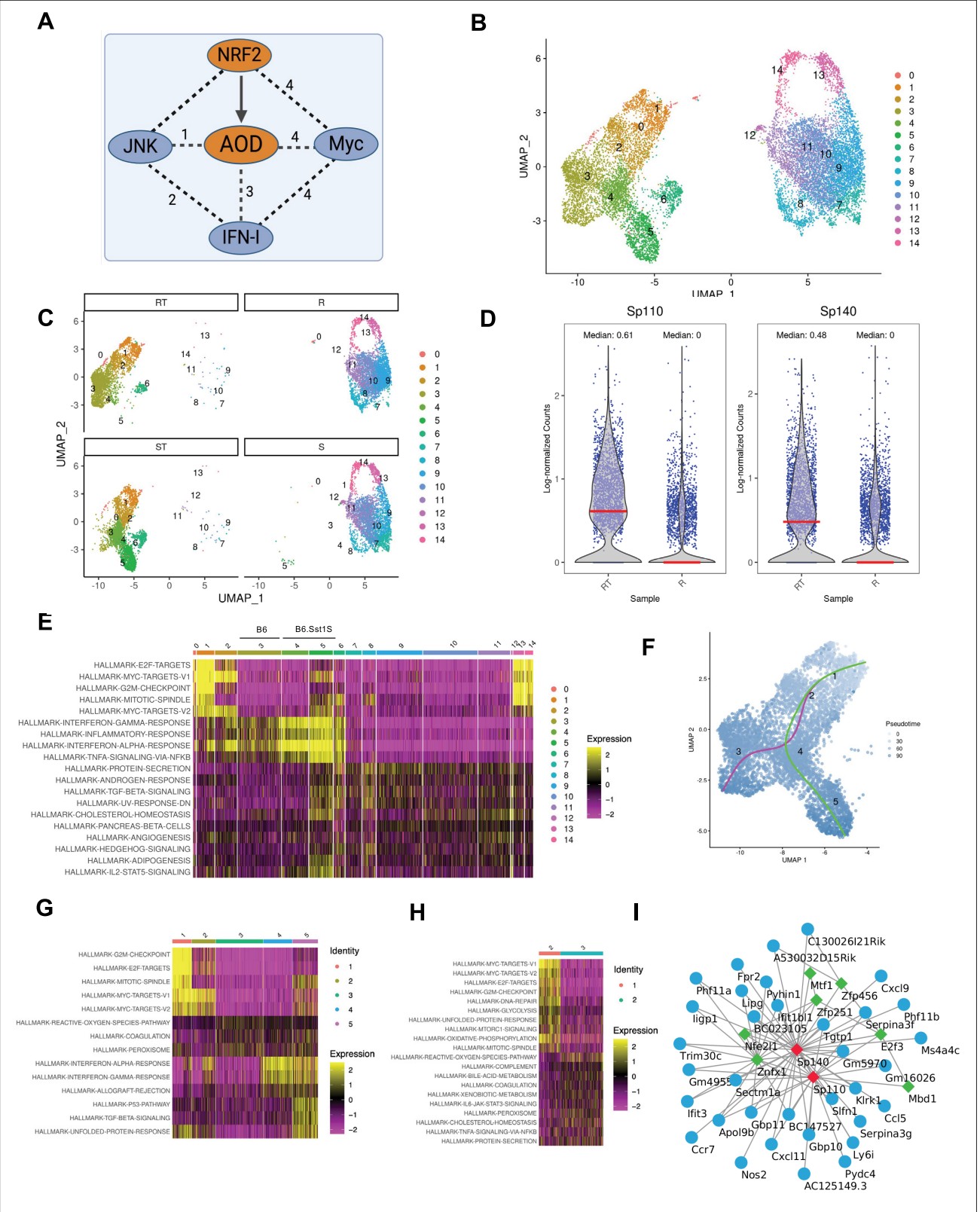

**Figure 1.** Single-cell RNAseq analysis of the population dynamics of B6 and B6. Sst1S macrophages after TNF stimulation. (**A**) Connectivity of antioxidant defense (AOD) with the Myc-, Nrf2-, JNK-, and IFN-I-regulated pathways: (1) stress kinase activation by oxidative stress; (2) promotion of IFN-I responses by stress kinases; (3) suppression of AOD by IFN-I; (4) inhibition of Nrf2, AOD, and IFN responses by Myc. (**B and C**) scRNA-seq analysis (UMAP and individual clusters) of B6 (R) and B6.Sst1S (S) BMDMs either naive (R and S) or after 24 hr of stimulation with TNF (RT and ST, respectively).

*Figure 1 continued on next page*

*Figure 1 continued*

(**D**) Expression of the *sst1*-encoded *Sp110* and *Sp140* genes in the population of either naïve (R) or TNF-stimulated (RT) B6 BMDMs. (**E**) Heatmap showing differentially expressed pathways in all cell clusters identified using scRNA-seq. Rows represent pathways and columns represent individual clusters with color intensity indicating the relative expression. (**F**) Reconstruction of the activation trajectories of TNF-stimulated resistant (RT) and susceptible (ST) macrophage populations using pseudotime analysis. Magenta line indicates B6 and green line indicates B6.Sst1S BMDMs. (**G**) Heatmap showing differentially expressed pathways in subpopulations 1–5 identified using pseudotime analysis. Rows represent pathways and columns represent individual subpopulations with color intensity indicating the relative expression. (**H**) Pathway heatmap representing transition from subpopulation 2 to unique subpopulation 3 in TNF-stimulated B6 macrophages. (**I**) The *Sp110* and *Sp140* gene regulatory network analysis. The mouse macrophage gene regulatory network was inferred using the GENIE3 algorithm from mouse macrophages gene expression data sets obtained from Gene Expression Omnibus (GEO). First neighbors of *Sp110/Sp140* genes were selected to infer a subnetwork of *Sp110/Sp140* co-regulated genes. Green nodes represent transcription factors, blue nodes denote their potential targets.

The online version of this article includes the following figure supplement(s) for figure 1:

**Figure supplement 1.** Single-cell RNAseq analysis of the B6 and B6.Sst1S macrophages after TNF stimulation.

## Results

### The *sst1* locus controls diverse trajectories of macrophage activation by TNF

To explore the spectrum of TNF responses in the B6 and B6.Sst1S BMDMs, we compared gene expression in the BMDM populations either naive, or after 24 hr of stimulation with TNF (10 ng/mL) using single-cell RNA sequencing (scRNA-seq; *Figure 1B*). The naive macrophage populations of both backgrounds were similar. All BMDMs responded to TNF stimulation, as evidenced by the de novo formation of clusters 1–6, where cluster 3 appeared exclusively in the resistant (B6) and clusters 4 and 5 in the susceptible (B6.Sst1S) macrophage populations (*Figure 1C*, *Figure 1—figure supplement 1A*). After TNF stimulation, the expression of the *sst1*-encoded *Sp110* and *Sp140* genes coordinately increased only in the B6 macrophages (*Figure 1D*, *Figure 1—figure supplement 1B*). The IFN pathway was dramatically upregulated in TNF-stimulated B6.Sst1S cells that lacked the *Sp110* and *Sp140* expression, but it was upregulated to a lesser degree in cluster 3 exclusive for TNF-stimulated *Sp110/140*-positive B6 macrophages (*Figure 1E*, *Figure 1—figure supplement 1C*).

To determine the relatedness of the diverse macrophage subpopulations, as they emerge during TNF stimulation, we performed trajectory analysis that clearly demonstrated partial overlap and the divergence between the trajectories of TNF-stimulated resistant (RT) and susceptible (ST) subpopulations (*Figure 1F*). Subpopulations sp1 and sp2 of the B6 and B6.Sst1S macrophages were closely related and were characterized by the upregulation of cell division pathways in both backgrounds (*Figure 1G*). Cell cycle analysis demonstrated that TNF stimulation increased the fraction of macrophages in S phase (*Supplementary file 1*).

The ST and RT activation trajectories diverged as the RT transitioned from sp2 to sp3, but the ST transitioned from sp2 to sp4 and 5. The transition from sp2 to sp3 in the resistant B6 BMDMs was characterized by an increase of the G1/S ratio, the downregulation of anabolic pathways involved in cell growth and replication (E2F and Myc), and the upregulation of anti-oxidant genes (*Figure 1G and H*). In contrast, in the susceptible B6.Sst1S macrophages, the sp2 transitioned to subpopulations sp4 and sp5 that were characterized by the decreased G1/S ratios as compared to sp3, signifying an increased G1 – S transition (*Supplementary file 2*). The sp4 represented an intermediate state between sp2 and sp5 and was primarily characterized by the upregulation of the IFN pathway (*Figure 1G*), as evidenced by the upregulation of known IFN-I pathway activation markers *Rsad2*, *IL1rn*, *Cxcl10*; (*Figure 1—figure supplement 1C and D*).

The ST macrophages in sp5 uniquely demonstrated a coordinated upregulation of the IFN, TGFβ, Myc, E2F, and stress response (p53 and UPR) pathways (*Figure 1G*). They upregulated the ISR genes (*Ddit3/Chop10, Atf3, Ddit4, Trib1, Trib3, Chac1*) in parallel with markers of immune suppression and apoptosis (*Cd274/PD-L1, Fas, Trail/Tnfsf10, Id2*, and a pro-apoptotic ligand – receptor pair *Tnfsf12/Tweak* and *Tnfrsf12a/Tweak* receptor; *Figure 1—figure supplement 1D*). Of note, IFN-I pathway genes were upregulated in all TNF-stimulated B6.Sst1S macrophages in agreement with the paracrine effect of Ifnβ, whose increased production by TNF-stimulated B6.Sst1S macrophages we have described previously (*Bhattacharya et al., 2021*). Interestingly, the expression of *Irf7* and several

IFN-inducible genes, such as *B2m*, *Cxcl10*, and *Ube2l6* was reduced in the sp5 cells, suggesting partial dampening of their IFN-I responsiveness (*Figure 1—figure supplement 1C*).

Taken together, the single-cell trajectory analysis revealed that sp5 represented a terminal state of the aberrant B6.Sst1S macrophage activation by TNF that was characterized by the coordinated upregulation of stress, pro-apoptotic, and immunosuppression genes. Unexpectedly, the stress escalation coincided with paradoxical activity of Myc and E2F pathways. Transition to this state in the susceptible macrophage population was preceded by IFN-I pathway upregulation (sp2 – sp4). In contrast, the sp2 – sp3 transition in the wild type macrophages was coincident with upregulation of both the *Sp110* and *Sp140* genes and was accompanied by the termination of cell cycle and the upregulation of antioxidant defense pathways. Therefore, we concluded that in resistant TNF-stimulated macrophages, the *sst1* locus-encoded genes promoted the activation of the AOD pathway either directly or by suppressing the IFN-I pathway.

To begin exploring the hierarchy and crosstalk of these pathways, we used an unbiased computational approach to define the *Sp110* and *Sp140* regulatory networks. First, we inferred a mouse macrophage gene regulatory network using the GENIE3 algorithm (*Huynh-Thu and Geurts, 2018*) and external gene expression data for mouse macrophages derived from Gene Expression Omnibus (GEO) (*Clough and Barrett, 2016*). This network represents co-expression dependencies between transcription factors and their potential target genes, calculated based on mutual variation in expression level of gene pairs (*Huynh-Thu et al., 2010*; *Zhernovkov et al., 2019*). This analysis revealed that in mouse macrophages, the *Sp110* and *Sp140* genes co-expressed with targets of Nfe2l1/2 (Nuclear Factor Erythroid 2 Like 1/2) and Mtf (metal-responsive transcription factor) TFs that are involved in regulating macrophage responses to oxidative stress and heavy metals, respectively (*Figure 1I*). Taken together, our experimental data and the unbiased network analysis suggested that in TNF-stimulated macrophages, the *sst1*-encoded *Sp110* and/or *Sp140* gene(s) might be primarily involved in regulating AOD.

## Dysregulated AOD activation in B6.Sst1S macrophages

Next, we compared the expression of upstream regulators of AOD in B6 and B6.Sst1S macrophages during TNF activation. Our previous studies demonstrated that the earliest differences between the B6 and B6.Sst1S BMDMs occurred between 8 and 12 hr of TNF stimulation, concomitant with the upregulation of the Sp110 protein in the B6 macrophages and heat shock proteins in the mutant cells (*Bhattacharya et al., 2021*). Comparing the time course of major transcriptional regulators of AOD in TNF-stimulated B6 and B6.Sst1S macrophages during this critical time interval, we observed higher upregulation of Nrf2 protein in TNF-stimulated B6 BMDMs (*Figure 2A*). The Nrf1 levels were not substantially upregulated after TNF stimulation and were similar in B6 and B6.Sst1S BMDMs (*Figure 2—figure supplement 1A and B*).

The Nrf2 difference was observed both in cytoplasmic and nuclear fractions (*Figure 2B and C*, respectively). The levels of Nrf2 negative regulators Keap1 and β-TrCP (*Figure 2—figure supplement 1A*), and Bach1 (*Figure 2B and C*) were similar in both backgrounds and did not notably change after TNF stimulation. Quantitative microscopy confirmed that at 12 hr of TNF stimulation, the cytoplasmic and nuclear Nrf2 levels significantly increased in B6 but not in B6.Sst1S BMDMs (*Figure 2D and E* and *Figure 2—figure supplement 1C*). In contrast, the levels of *Nfe2l1/2* mRNA induced by TNF were higher in the mutant macrophages, suggesting post-transcriptional regulation (*Figure 2F*). Therefore, we measured the rates of Nrf2 protein degradation 6–8 hr after TNF stimulation but found no difference (*Figure 2G and H*).

Using EMSA, we demonstrated that binding activity of nuclear Nrf2 to its target DNA at 8 and 12 hr after TNF stimulation was greater in the resistant B6 BMDMs (*Figure 2I*). To identify core pathways controlled by the *sst1* locus during this critical period, we compared global mRNA expression profiles of the B6 and B6.Sst1S macrophages after 12 hr of TNF stimulation using bulk RNA-seq. This analysis confirmed that the *Sp110* and *Sp140* genes were strongly upregulated by TNF stimulation exclusively in the B6 macrophages (*Figure 2J*). Gene set enrichment analysis (GSEA) of genes differentially expressed between TNF-stimulated B6.Sst1S and B6 macrophages at this critical junction revealed that the IFN response, Myc, E2F target gene, Hypoxia, UV response, and DNA repair pathways were upregulated in the mutant macrophages, while the detoxification of reactive oxygen species, cholesterol homeostasis, fatty acid metabolism, and oxidative phosphorylation, peroxisome,

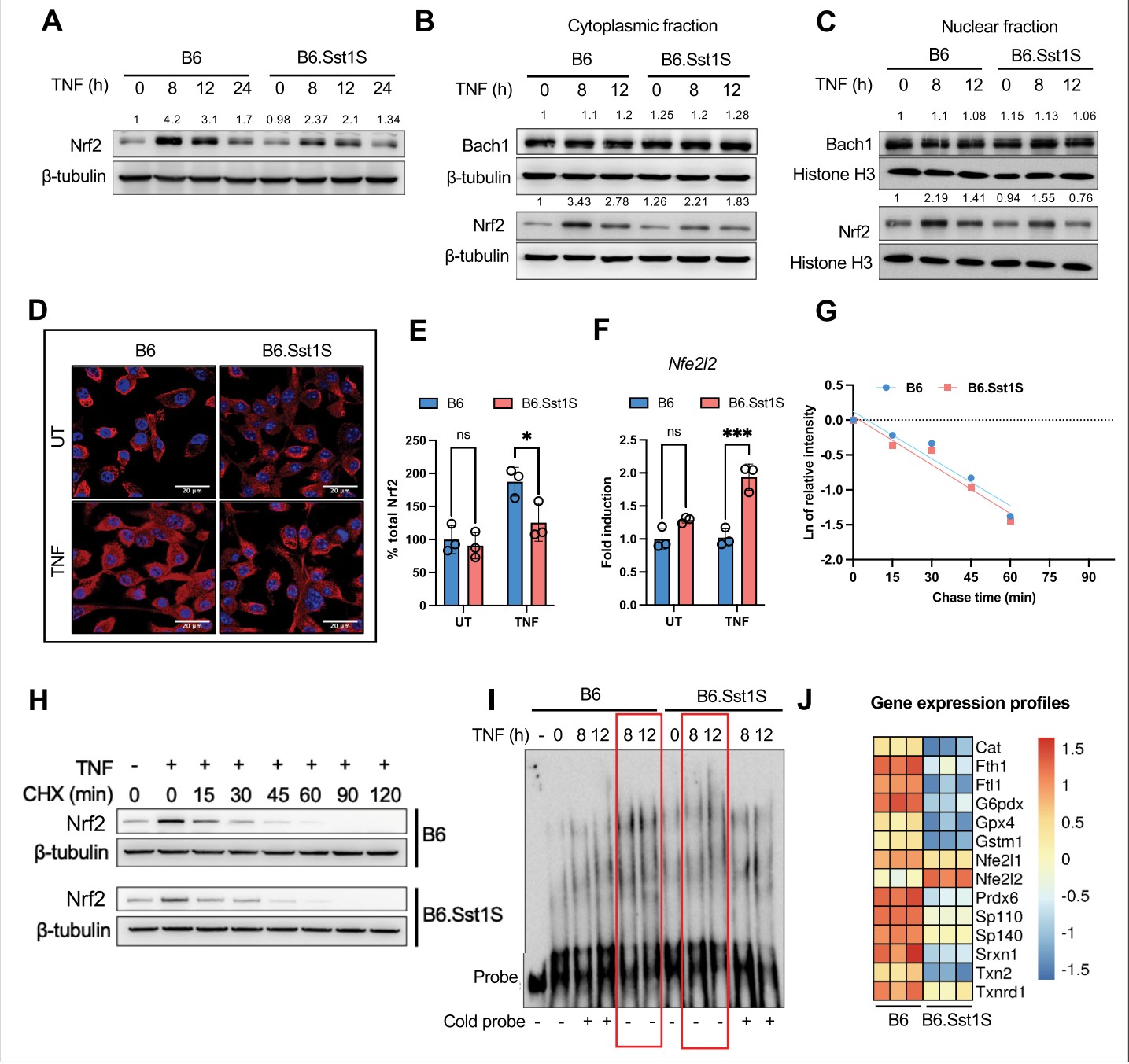

**Figure 2.** Gene expression profiling comparing B6 and B6.Sst1.S BMDMs stimulated with TNF and regulation of NRF2. (**A**) Total level of Nrf2 protein in B6 and B6.Sst1S BMDMs stimulated with TNF (10 ng/mL) for 8, 12, and 24 hr (western blotting). Average densitometric values from two independent experiments were included above the blot. (**B**) Cytoplasmic Nrf2 and Bach1 proteins in B6 and B6.Sst1S BMDMs stimulated with TNF (10 ng/mL) for 8 and 12 hr (western blotting). Average densitometric values from two independent experiments were included above the blot. (**C**) Nuclear Nrf2 and Bach1 protein levels in B6 and B6.Sst1S BMDMs treated with TNF (10 ng/mL) for 8 and 12 hr (western blotting). Average densitometric values from two independent experiments were included above the blot. (**D and E**) Confocal microscopy of Nrf2 protein in B6 and B6.Sst1S BMDMs stimulated with TNF (10 ng/mL) for 12 hr (scale bar 20 µm). The data shows staining with Nrf2-specific antibody and performed area quantification using ImageJ to calculate the Nrf2 total signal intensity per field. Each dot in the graph represents the average intensity of 3 fields in a representative experiment. The experiment was repeated three times. (**F**) B6 and B6.Sst1S BMDMs were stimulated with TNF (10 ng/mL) for 8 hr. The *Nfe2l2* mRNA levels were quantified using quantitative RT-PCR. Fold induction was calculated by DDCt method, and b-actin was used as internal control and normalized the fold change using B6 UT. (**G and H**) The Nrf2 protein stability in TNF-stimulated (10 ng/mL) B6 and B6.Sst1S BMDMs. BMDMs were stimulated with TNF. After 6 hr, 25 µg/ mL of cycloheximide (CHX) was added and cells were harvested after 15, 30, 45, 60, 90, and 120 min. The Nrf2 protein levels after TNF stimulation and

*Figure 2 continued on next page*

*Figure 2 continued*

degradation after cycloheximide addition were determined by western blotting. I - Linear regression curves of Nrf2 degradation after addition of CHX. Band intensities were measured by densitometry using ImageJ. No significant difference in the Nrf2 half-life was found: B6: 15.14±2.5 min and B6.Sst1S: 13.35±0.6 min. (I) Nuclear Nrf2 binding to target sequence. Nuclear extracts were prepared from BMDMs treated with TNF (10 ng/mL) for 8 and 12 hr. The binding activity of Nrf2 was monitored by EMSA using biotin-conjugated Nrf2-specific probe (hot probe, red frames). Competition with the unconjugated NRF2 probe (cold probe) was used as specificity control. (J) Anti-oxidant genes co-regulated with *Sp110* and *Sp140* after stimulation with TNF (10 ng/mL) for 12 hr. The heatmap was generated using FPKM values obtained from RNA-seq expression profiles of B6.Sst1S and B6 BMDMs after 12 hr of TNF stimulation. The data are presented as means ± standard deviation (SD) from three to five samples per experiment, representative of three independent experiments. The statistical significance was performed by two-way ANOVA using Tukey's multiple comparison test (Panel E, **F**). Significant differences are indicated with asterisks (*, p<0.05; **, p<0.01; ***, p<0.001; ****, p<0.0001).

The online version of this article includes the following source data and figure supplement(s) for figure 2:

**Source data 1.** PDF file containing original western blots for *Figure 2A, B, C and H* and EMSA for *Figure 2I* indicating the relevant bands and treatments.

**Source data 2.** Original files for western blot analysis displayed in *Figure 2A, B, C and H* and EMSA for *Figure 2I*.

**Figure supplement 1.** Antioxidant response of TNF-stimulated B6 and B6.Sst1.S BMDMs.

**Figure supplement 1—source data 1.** PDF file containing original western blots for *Figure 2—figure supplement 1A*, indicating the relevant bands and treatments.

**Figure supplement 1—source data 2.** Original files for western blot analysis displayed in *Figure 2—figure supplement 1A*.

and lysosome pathways were downregulated (*Supplementary file 3*). Functional pathway profiling using KEGG and Reactome databases also highlighted the upregulation of genes involved in oxidative stress and cellular senescence in mutant macrophages. In contrast, the wild-type macrophages upregulated genes involved in detoxification of reactive oxygen species, inhibition of ferroptosis, and peroxisome function (*Supplementary file 4*). Supporting these findings, the total antioxidant capacity of the B6 macrophages after TNF stimulation increased to significantly higher levels, as compared to the B6.Sst1S (*Figure 2—figure supplement 1D*).

Transcription factor binding site analysis of genes specifically upregulated by TNF in B6, but not B6.Sst1S, macrophages (B6-specific cluster) revealed an enrichment of Nfe2l1/Nfe2l2, Bach1, and Mafk sequence motifs, that is binding sites of transcription factors regulating AOD. In contrast, over-representation of E2F, Egr1, and Pbx3 transcription factor binding sites was found for genes in the Sst1S-specific cluster (*Supplementary file 5*). A master regulator analysis using Virtual Inference of Protein Activity by Enriched Regulon Analysis (VIPER) algorithm also revealed a key role for Nfe2l1/2 (NF-E2-like) transcription factors (TFs) as regulators of genes differentially induced by TNF in B6 and B6.Sst1S BMDMs (*Supplementary file 6*).

To further investigate this inference, we analyzed the expression of a gene ontology set 'response to oxidative stress' (GO0006979, 416 genes) and observed clear separation of these genes in two clusters in a *sst1*-dependent manner (*Figure 2J* and *Figure 2—figure supplement 1G*). This analysis demonstrated that the response to oxidative stress in the *sst1* mutant macrophages was dysregulated, but not paralyzed. For example, the upregulation of well-known Nrf2 target genes Heme oxygenase 1 (*Hmox1*) and (NAD(P)H quinone dehydrogenase 1) (*Nqo1*) was similar in B6 and B6.Sst1S BMDMs (*Figure 2—figure supplement 1E and F*, respectively).

A subset of antioxidant defense genes whose expression was concordant with *Sp110* and *Sp140* in B6 macrophages represented genes that are known to be involved in iron storage (ferritin light and heavy chains, *Ftl* and *Fth*), ROS detoxification and maintenance of redox state (*Cat, G6pdx, Sod2, Gstm1, Gpx4, Prdx6, Srxn1, Txn2,* and *Txnrd1*; *Figure 2J*). We hypothesized that their coordinate downregulation in TNF-stimulated B6.Sst1S macrophages sensitized the mutant cells to iron-mediated oxidative damage and played a pivotal role in shaping their divergent activation trajectory.

## Persistent TNF stimulation of B6.Sst1S macrophages leads to increased accumulation of lipid peroxidation products and IFN-I pathway hyperactivity

To test this hypothesis, first we explored the intracellular iron storage. Both the ferritin light (*Ftl*) and heavy (*Fth*) chain genes were dysregulated in TNF-stimulated B6.Sst1S BMDMs. While the *Fth* mRNA was upregulated by TNF in B6 BMDMs, it remained at a basal level in the B6.Sst1S cells (*Figure 3A*).

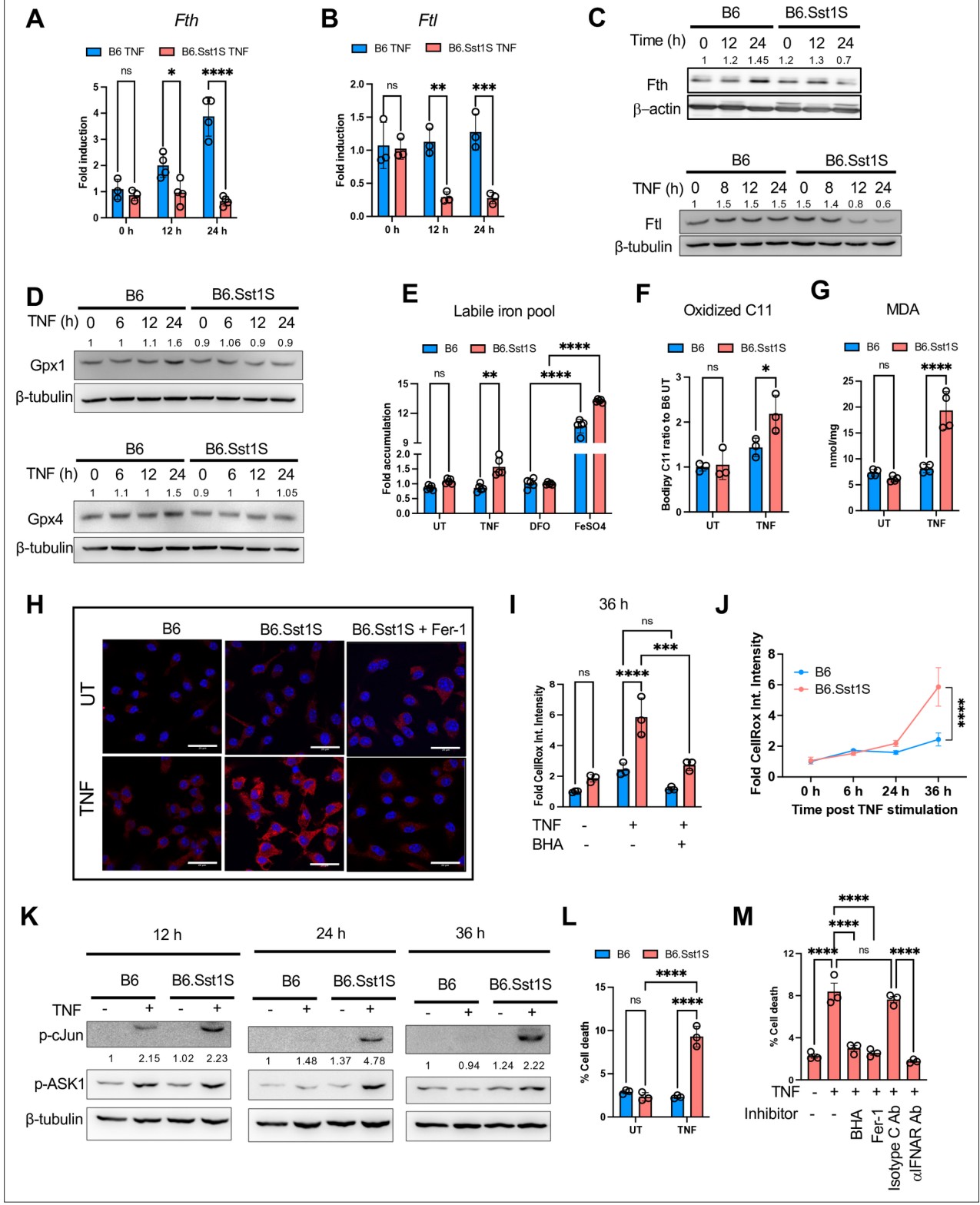

**Figure 3.** Regulation of iron and lipid peroxidation in B6 and B6.Sst1.S BMDMs. (**A and B**) The expression of *Fth* and *Ftl* genes in B6 and B6.Sst1S BMDMs treated with 10 ng/mL TNF for 12 and 24 hr was determined using qRT-PCR. Fold induction was calculated normalizing with B6 untreated control using ΔΔCt method, and 18 S was used as internal control. (**C**) The Fth and Ftl protein levels in B6 and B6.Sst1S BMDMs treated with 10 ng/mL TNF for 0, 12, and 24 hr (Fth) and 0, 8, 12, and 24 hr (for Ftl; western blot). Average densitometric values from three independent experiments were included above the blot. (**D**) The Gpx1 and Gpx4 protein levels in B6 and B6.Sst1S BMDMs stimulated with TNF (10 ng/mL) for 0, 6, 12, and 24 hr

*Figure 3 continued on next page*

*Figure 3 continued*

(western blot). Average densitometric values from two independent experiments were included above the blot. (**E**) The labile iron pool (LIP) in TNF-stimulated B6 and B6.Sst1S BMDMs was treated with 10 ng/mL TNF for 24 hr. UT - untreated control. The LIP was determined using the Calcein AM method and represented as fold change as compared to B6 untreated. DFO was used as a negative control, and FeSO4 was used as a positive control. (**F**) The lipid peroxidation levels were determined by fluorometric method using C11-Bodipy 581/591. BMDMs from B6 and B6.Sst1S were treated with 10 ng/mL TNF for 30 hr. UT - untreated control. (**G**) Production of lipid peroxidation metabolite malondialdehyde (MDA) by B6 and B6.Sst1S BMDMs treated with 10 ng/mL TNF for 30 hr. UT - untreated control. (**H**) The accumulation of the intracellular lipid peroxidation product 4-HNE in B6 and B6.Sst1S BMDMs treated with 10 ng/mL TNF for 48 hr. The lipid peroxidation (ferroptosis) inhibitor, Fer-1 (10 μM), was added 2 hr post TNF stimulation in B6.Sst1S macrophages. The 4-HNE adducts accumulation was detected using 4-HNE-specific antibody and confocal microscopy. (**I**) Reactive oxygen species (ROS) levels were observed using the CellROX assay and quantified by automated microscopy in B6 and B6.Sst1S BMDMs either treated with TNF (10 ng/mL) or left untreated for 36 hr. BHA (100 μM) was used as a positive control. Data are presented as fold mean fluorescence intensity (MFI) normalized by B6 UT, representing ROS levels. (**J**) Time course of ROS accumulation in B6 and B6.Sst1S BMDMs during TNF-stimulated condition. Reactive oxygen species (ROS) levels were observed using the CellROX assay after 0, 6, 24, and 36 hr of TNF stimulation and quantified by automated microscopy. (**K**) Induction of c-Jun and ASK1 phosphorylation by TNF in B6 and B6.Sst1S BMDMs. The B6 and B6.Sst1S BMDMs were treated with TNF (10 ng/ml) or left untreated for 12, 24, and 36 hr and the c-Jun and ASK1 phosphorylation was determined by western blot. Average densitometric values from two independent experiments were included above the blot. (**L**) Cell death in B6 and B6.Sst1S BMDMs stimulated with 50 ng/mL TNF for 48 hr. Percent of dead cells was determined by automated microscopy using Live-or-DyeTM 594/614 Fixable Viability stain (Biotium) and Hoechst staining. (**M**) Inhibition of cell death of B6.Sst1S BMDMs stimulated with 50 ng/mL TNF for 48 hr using IFNAR1 blocking antibodies (5 μg/mL), isotype C antibodies (5 μg/mL), Butylated hydroxyanisole (BHA, 100 μM), or Fer-1 (10 μM). Percent cell death was measured as in L. The data are presented as means ± standard deviation (SD) from three to five samples per experiment, representative of three independent experiments. Statistical analysis was performed using two-way ANOVA followed by Šídák's multiple comparison test (Panels **A, B, F, G, and M**) and Tukey's multiple comparison test (Panels **E, I, J, L**). Statistical significance is indicated by asterisks: *$p0.05$, **$p<0.01$, ***$p<0.001$, ****$p<0.0001$.

The online version of this article includes the following source data and figure supplement(s) for figure 3:

**Source data 1.** PDF file containing original western blots for *Figure 3C, D and K*, indicating the relevant bands and treatments.

**Source data 2.** Original files for western blot analysis displayed in *Figure 3C, D and K*.

**Figure supplement 1.** Regulation of TNF-induced ROS, labile iron pool, and lipid peroxidation in B6 and B6.Sst1.S BMDMs.

The *Ftl* mRNA level was significantly reduced after TNF stimulation in B6.Sst1S macrophages but remained at the basal level in B6 (*Figure 3B*). Accordingly, the expression of Ftl protein was reduced in B6.Sst1S BMDMs after 12 hr of TNF stimulation, and both Ftl and Fth proteins were reduced at 24 hr (*Figure 3C*). In parallel, the levels of Gpx1 and Gpx4 proteins were also substantially reduced by 24 hr in B6.Sst1S (*Figure 3D*). The glutathione peroxidase 4 (Gpx4) protein plays a central role in preventing ferroptosis because of its unique ability to reduce hydroperoxide in complex membrane phospholipids and, thus, limit self-catalytic lipid peroxidation (*Stockwell et al., 2017*). Thus, TNF-stimulated B6.Sst1S BMDMs had reduced intracellular iron storage capacity accompanied by the decline of the major lipid peroxidation inhibitor Gpx4. Accordingly, we observed increases in an intracellular labile iron pool (LIP, *Figure 3E*), an intracellular accumulation of oxidized lipids (*Figure 3F*), a toxic terminal lipid peroxidation (LPO) products malondialdehyde (MDA, *Figure 3G*) and 4-hydroxynonenal (4-HNE) adducts (*Figure 3H*, *Figure 3—figure supplement 1A*). Treatment with the LPO inhibitor Ferrostatin-1 (Fer-1) prevented the 4-HNE adducts accumulation (*Figure 3H*, *Figure 3—figure supplement 1F*). The levels of the LIP and LPO remained significantly elevated in B6.Sst1S macrophages after 48 hr of TNF stimulation (*Figure 3—figure supplement 1B-F*). We also observed an increased ROS production in the B6.Sst1S macrophages during persistent TNF stimulation (*Figure 3I and J*, and *Figure 3—figure supplement 1G*). The ASK1 - JNK – cJun stress kinase axis was upregulated in B6.Sst1S macrophages, as compared to B6, 12–36 hr of TNF stimulation, that is during the aberrant activation stage confirming persistent stress (*Figure 3K*). By 48 hr of TNF stimulation, we noted moderate cell death in B6.Sst1S macrophage cultures (*Figure 3L*, *Figure 3—figure supplement 1H*). Treatments with Fer-1, the antioxidant butylated hydroxyanisole (BHA), or IFNAR1 blockade each prevented the cell death (*Figure 3M*), suggesting that both persistent stress and macrophage death were mediated by ROS, lipid peroxidation, and IFN-I pathway.

Next, we wanted to test whether the IFN-I pathway hyperactivity in TNF-stimulated B6.Sst1S macrophages was responsible for the initial dysregulation of the *Ftl*, *Fth*, and AOD gene expression, that is at 8–12 hr of TNF stimulation. The IFNAR1 blockade, however, did not restore the Nrf2 and Ftl protein levels, or the *Fth*, *Ftl*, and *Gpx1* mRNA levels to the wild type B6 levels (*Figure 4A-C*). Additionally, we observed that LPO production was increased as early as 6 hr under TNF-stimulated

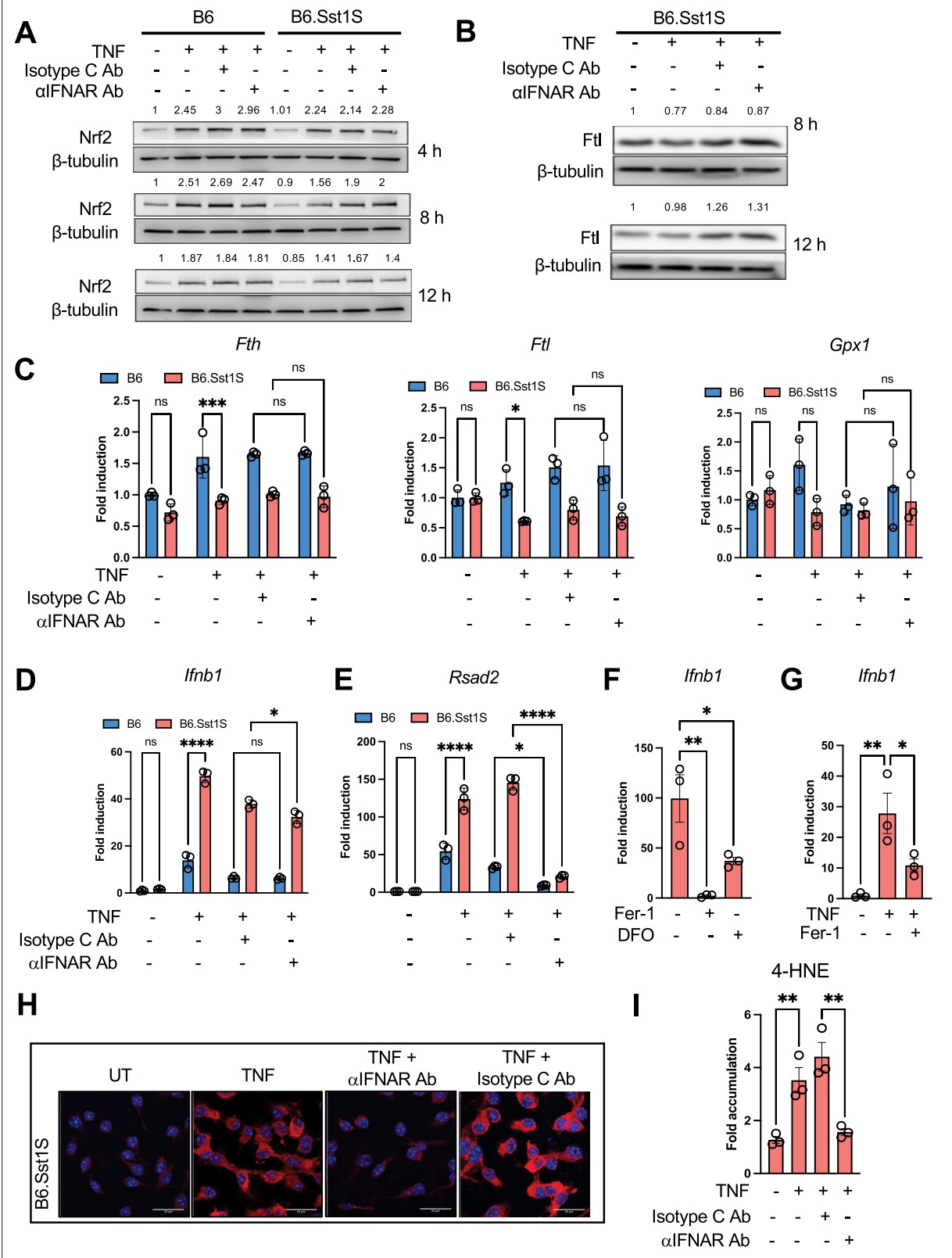

**Figure 4.** Crosstalk of the IFN-I and AOD pathways. (**A**) IFNAR1 blockade does not enhance Nrf2 upregulation in TNF-stimulated B6.Sst1S macrophages. B6 and B6.Sst1S BMDMs were treated with 10 ng/m TNF, with or without IFNAR1-blocking antibodies or isotype control (Isotype C Ab) at 5 μg/mL concentration for 4, 8, and 12 hr. Nrf2 protein levels were quantified by western blot. Average densitometric values from two separate experiments were included above the blot. (**B**) IFNAR1 blockade does not increase Ftl expression in TNF-stimulated B6.Sst1S macrophages. B6.Sst1S

*Figure 4 continued on next page*

*Figure 4 continued*

BMDMs were treated with 10 ng/mL TNF, with or without IFNAR1-blocking antibodies (5 µg/mL) or Isotype C Ab (5 µg/mL), for 8 and 12 hr. Ftl protein levels were quantified by western blot. Average densitometric values from two independent experiments were included above the blot. (**C**) IFNAR1 blockade does not increase mRNA levels of *Fth*, *Ftl*, and *Gpx1*. B6 and B6.Sst1S BMDMs were treated with 10 ng/mL TNF, with or without IFNAR1-blocking antibodies or Isotype C Ab, for 12 hr. Blocking antibodies (5 µg/mL) or isotype C antibodies (5 µg/mL) were added 2 hr after TNF stimulation. Fold induction was calculated using B6 untreated control as average one-fold by utilizing the ΔΔCt method with β-actin as the internal control. (**D and E**) IFNAR1 blockade reduces *Rsad2* mRNA levels (**E**) but does not affect *Ifnb1* mRNA levels (**D**) B6 and B6.Sst1S BMDMs were treated with 10 ng/mL TNF, with or without IFNAR1-blocking antibodies or Isotype C Ab for 16 hr. Blocking antibodies (5 µg/mL) or isotype C antibodies (5 µg/mL) were added 2 hr after TNF stimulation. Fold induction was calculated using B6 untreated control as average onefold by utilizing the ΔΔCt method with β-actin as the internal control. (**F**) Lipid peroxidation inhibition prevents the superinduction of *Ifnb1* mRNA. B6.Sst1S BMDMs were treated with 10 ng/mL TNF, and the lipid peroxidation inhibitor (Fer-1) was added 2 hr post-TNF stimulation. *Ifnb1* mRNA levels were measured using qRT-PCR after 16 hr of TNF treatment. Fold induction was calculated using untreated control as average onefold by utilizing the ΔΔCt method with 18 S as the internal control. (**G**) Lipid peroxidation inhibition reverses the superinduction of *Ifnb1* mRNA. B6.Sst1S BMDMs were stimulated with 10 ng/mL TNF for 18 hr, then the LPO inhibitor (Fer-1) was added for the remaining 12 hr. *Ifnb1* mRNA levels were measured using qRT-PCR. Fold induction was calculated using untreated control as average onefold by utilizing the ΔΔCt method with 18 S as the internal control. (**H and I**) IFNAR1 blockade reduces 4-HNE adduct accumulation in B6.Sst1S BMDMs treated with TNF (10 ng/mL) for 48 hr. Blocking antibodies (5 µg/mL) or Isotype C Ab (5 µg/mL) were added 2 hr post-TNF stimulation. The data are presented as means ± standard deviation (SD) from three to five samples per experiment, representative of three independent experiments. The statistical significance was performed by two-way ANOVA using Tukey's multiple comparison test (Panels C-E), Ordinary one-way ANOVA using Bonferroni's multiple comparison test (Panels F-G and I). Significant differences are indicated with asterisks (*, $p<0.05$; **, $p<0.01$; ***, $p<0.001$; ****, $p<0.0001$).

The online version of this article includes the following source data and figure supplement(s) for figure 4:

**Source data 1.** PDF file containing original western blots for *Figure 4A and B*, indicating the relevant bands and treatments.

**Source data 2.** Original files for western blot analysis displayed in *Figure 4A and B*.

**Figure supplement 1.** Regulation of lipid peroxide production and type I IFN expression in TNF-stimulated B6 and B6.Sst1.S BMDMs.

**Figure supplement 2.** The transient upregulation of transposon mRNAs in B6 macrophages after TNF treatment is affected in the B6.Sst1S.

condition in both B6 and B6.Sst1S prior to *Ifnb1* superinduction (*Figure 4—figure supplement 1A and B*). These data suggest that type I IFN signaling does not initiate LPO in our model.

Of note, the IFNAR1 blockade did not prevent the *Ifnb1* mRNA superinduction (*Figure 4D*), thus rejecting a hypothesis that the *Ifnb1* superinduction in B6.Sst1S macrophages was driven via an Ifnβ – IFNAR1 positive feedback (*Ivashkiv and Donlin, 2014*). The mRNA expression of the interferon-inducible gene *Rsad2*, however, was suppressed, demonstrating the efficiency of the IFNAR1 blockade (*Figure 4E*).

In contrast, treatment of B6.Sst1S macrophages with Fer-1 or the iron chelator DFO during initial TNF stimulation inhibited both the *Ifnb1* and *Rsad2* mRNAs upregulation (*Figure 4F*, *Figure 4—figure supplement 1C* respectively). Importantly, Fer-1 treatment also reduced the *Ifnb1* and *Rsad2* levels in B6.Sst1S macrophages when added at 18 hr after TNF stimulation, that is during established aberrant response (*Figure 4G*, *Figure 4—figure supplement 1D*). Thus, lipid peroxidation was involved in both the initial *Ifnb1* superinduction and in maintenance of the and IFN-I pathway hyperactivity driven by prolonged TNF stimulation.

Next, we wanted to test whether continuous IFN-I signaling was required for the accumulation of 4-HNE adducts during prolonged macrophage activation. Indeed, the blockade of type I IFN receptor (IFNAR1) after 2, 12, or 24 hr after TNF stimulation prevented the 4-HNE adducts accumulation at 48 hr (*Figure 4H and I*, and *Figure 4—figure supplement 1E-G*). Thus, *Ifnb1* super-induction and IFN-I pathway hyperactivity in B6.Sst1S macrophages follow the initial LPO production and maintain and amplify it during prolonged TNF stimulation.

The *sst1*-encoded *Sp110* and *Sp140* genes were described as interferon-induced genes, and Sp140 protein was implicated in maintenance of heterochromatin silencing in activated macrophages (*Fraschilla and Jeffrey, 2020a*; *Mehta et al., 2017*). Therefore, we hypothesized that their deficiency in the TNF-stimulated B6.Sst1S mutants may lead to the upregulation of silenced transposons and, thus, trigger the Ifnβ upregulation via intracellular RNA sensors, that is by an autonomous mechanism unrelated to the AOD dysregulation. We examined the transcriptomes of B6 and B6.Sst1S macrophages before and after TNF stimulation for the presence of persistent viruses or transposons using a custom bioinformatics pipeline (*Ma et al., 2021*). No exogenous mouse viral RNAs were detected. A select set of mouse LTR-containing endogenous retroviruses

(ERVs; *Jayewickreme et al., 2021*), and non-retroviral LINE L1 elements were expressed at a basal level before and after TNF stimulation, but their levels in the B6.Sst1S BMDMs were similar to or lower than those seen in B6 (*Figure 4—figure supplement 2A-C*). We also tested the accumulation of dsRNA using deep sequencing of macrophage small RNAs and failed to detect evidence of transposon-derived dsRNAs (*Figure 4—figure supplement 2D*). We concluded from these findings that the majority of the basal transposon RNAs in macrophages exist primarily as single-stranded mRNAs that evade triggering interferon pathway. The above analyses allowed us to exclude the overexpression of persistent viral or transposon RNAs as a primary mechanism of the IFN-I pathway hyperactivity.

Taken together, the above experiments allowed us to reject the hypothesis that IFN-I hyperactivity caused the *sst1*-dependent AOD dysregulation. In contrast, they established that the hyperactivity of the IFN-I pathway in TNF-stimulated B6.Sst1S macrophages was itself driven by the *initial* dysregulation of AOD and iron-mediated lipid peroxidation. During prolonged TNF stimulation, however, the IFN-I pathway was upregulated, possibly via ROS/LPO-dependent JNK activation, and acted as a potent amplifier of lipid peroxidation.

## Hyperactivity of Myc in susceptible macrophages after TNF stimulation fuels lipid peroxidation

We wanted to determine whether the AOD genes were regulated by the *sst1*-encoded genes directly or indirectly via an intermediary regulator. Previously, we identified two transcription factors whose binding activities were exclusively upregulated in the susceptible macrophages after 12 hr of TNF stimulation: Myc and HSF1 (*Bhattacharya et al., 2021*). The bulk RNA-seq analysis at this timepoint (12 hr) also demonstrated the upregulation of Myc pathway along with E2F target genes and stress responses specifically in TNF-stimulated B6.Sst1S BMDMs, as compared to B6 (*Figure 2*). Of note, the scRNA-seq analysis above also demonstrated the association of Myc and stress response pathways in the mutant cells at 24 hr (*Figure 1*). Therefore, we hypothesized that in susceptible macrophages, Myc might be involved in the dysregulation of AOD and iron storage.

First, we observed that Myc was regulated in an *sst1*-dependent manner: in TNF-stimulated B6 wild-type BMDMs, *c-Myc* mRNA was downregulated, while in the susceptible macrophages, c-*Myc* mRNA was upregulated (*Figure 5A*). The c-Myc protein levels were also higher in the B6.Sst1S cells in unstimulated BMDMs and 6–12 hr of TNF stimulation (*Figure 5B*). Next, we tested whether suppression of Myc activity could 'normalize' the susceptible phenotype using Myc-Max dimerization inhibitor 10058-F4 (F4). Indeed, this treatment increased the levels of Fth and Ftl proteins in TNF-stimulated susceptible macrophages (*Figure 5C*) and decreased the LIP (*Figure 5D*). Accordingly, the levels of MDA, oxidized lipid,s and 4-HNE adducts also significantly decreased (*Figure 5E-G*, *Figure 5—figure supplement 1A and B*), as well as the levels of *Ifnb1*, *Rsad2*, and the ISR markers *Trib3* and *Chac1* (*Figure 5H*). Possibly, the ISR activation serves as an alternative pathway of Myc inhibition, as it is known to inhibit the oncogene protein translation (*Wolfe et al., 2014*).

Next, we wanted to determine whether the upregulation of Myc is driven by TNF alone or in synergy with CSF1, a growth factor that also stimulates Myc. In vitro, we observed the upregulation of Myc shortly after the addition of fresh CSF1-containing media, but no difference in the Myc protein dynamics between B6 and B6.Sst1S BMDMs in the absence of TNF (*Figure 5—figure supplement 1C*). In addition, we tested whether CSF1R inhibitors could prevent the superinduction of *Ifnb1* and *Rsad2* mRNAs in TNF-stimulated B6.Sst1S macrophages. The CSF1R inhibitors were used at concentrations that did not cause macrophage death (*Figure 5—figure supplement 1D*). Neither of these inhibitors prevented the *Ifnb1* and *Rsad2* upregulation (*Figure 5—figure supplement 1E and F*). Because Myc induction by CSF1 and TNF is conducted via distinct relays of receptor signaling and transcription factors, we concluded that the *sst1* locus specifically controls Myc expression induced by inflammatory signaling. Indeed, Myc promoter has multiple predicted NF-κB and/or AP-1 transcription factors binding sites. To test this hypothesis, we used specific JNK inhibitor D-JNK1, but it only partially reduced Myc protein levels in TNF-stimulated B6.Sst1S macrophages (*Figure 5I and J*). Because both *Sp110* and *Sp140* mRNAs and proteins are upregulated during extended TNF stimulation, one of them may participate in feedback regulation of TNF-induced Myc, either directly or via yet unknown intermediates.

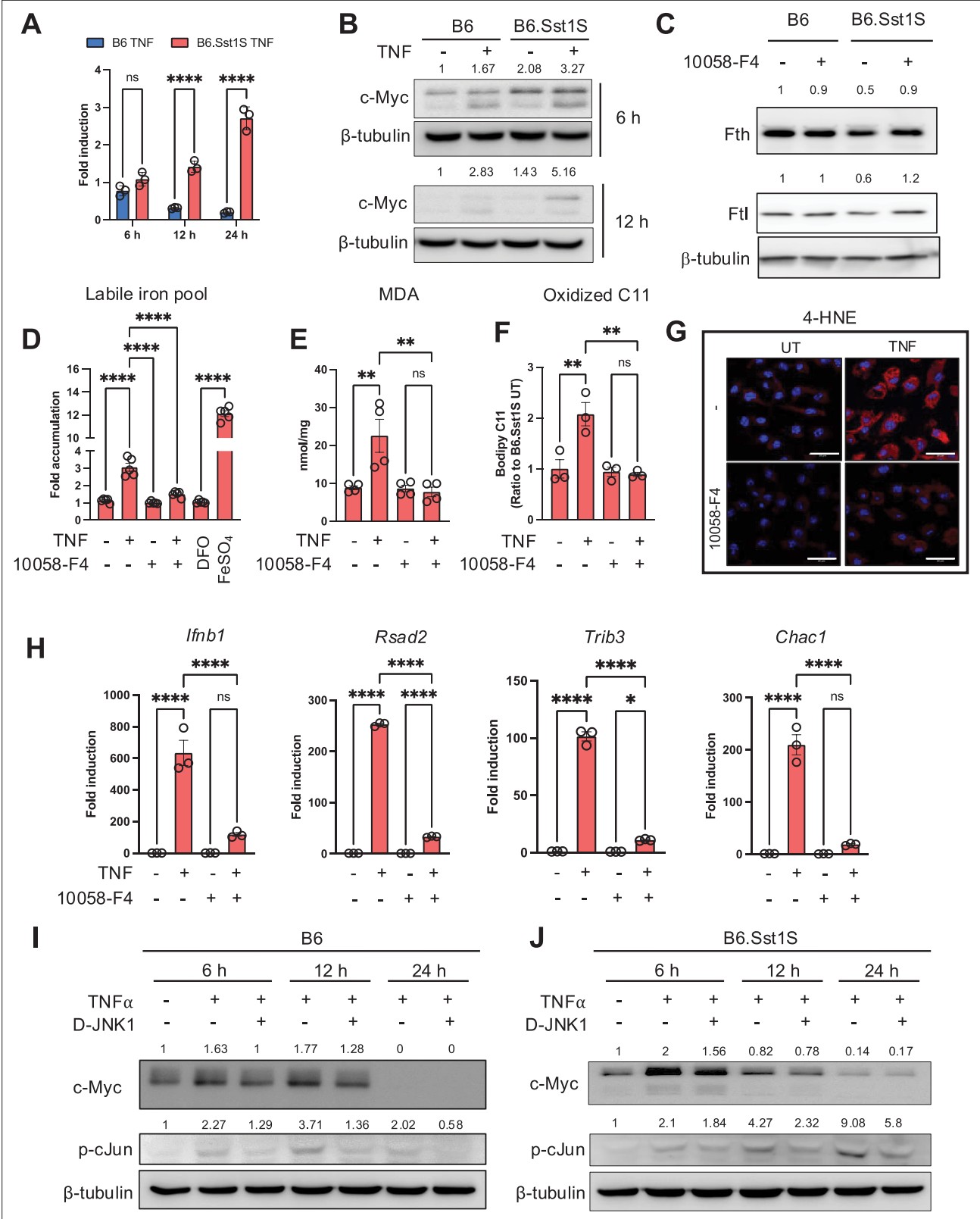

**Figure 5.** Myc dysregulation drives the aberrant state of macrophage activation. (**A**) The lack of *Myc* mRNA downregulation after prolonged TNF stimulation in B6.Sst1S macrophages. BMDMs from B6 and B6.Sst1S were treated with 10 ng/mL TNF for 6, 12, and 24 hr. Expression of *Myc* was quantified by the ΔΔCt method using qRT-PCR and expressed as a fold induction compared to the untreated B6 BMDMs. β-actin was used as the internal control. (**B**) Myc protein levels expressed by B6 and B6.Sst1S BMDMs during the course of stimulation with TNF(10 ng/mL) for 6 and 12 hr.

*Figure 5 continued on next page*

*Figure 5 continued*

(western blot). Average densitometric values from two independent experiments were included above the blot. (C) Myc inhibition restored the levels of Fth and Ftl proteins in TNF-stimulated B6.Sst1S macrophages to the B6 levels. B6 and B6.Sst1S BMDMs were treated with 10 ng/mL TNF alone or in combination with Myc inhibitor, 10058-F4 (10 µM) for 24 hr. 10058-F4 was added 2 hr post TNF stimulation. Protein levels of Fth and Ftl were observed using western blot. Average densitometric values from two independent experiments were included above the blot. (D) Myc inhibition decreased the labile iron pool in TNF-stimulated B6.Sst1S macrophages. B6.Sst1S BMDMs were treated with 10 ng/mL TNF or left untreated for 48 hr. The 10058-F4 inhibitor was added 2 hr post TNF stimulation. The labile iron pool (LIP) was measured using the Calcein AM method and represented as fold change as compared to untreated control. DFO was used as a negative control, and FeSO4 was used as a positive control. (E and F) Myc inhibition reduced lipid peroxidation in TNF-stimulated B6.Sst1S BMDMs. Cells were treated with 10 ng/mL TNF in the presence or absence of 10058-F4 for 48 hr. The inhibitor was added 2 hr post TNF stimulation. The MDA production was measured using commercial MDA assay (E) The lipid peroxidation was measured by fluorometric method using C11-Bodipy 581/591 (F). (G) B6.Sst1S BMDMs were treated as above in E. The accumulation of lipid peroxidation product, 4-HNE after 48 hr was detected by confocal microscopy using 4-HNE-specific antibody. The 4-HNE adducts accumulation was quantified using ImageJ and plotted as fold accumulation compared to untreated group. (H) The BMDMs from B6.Sst1S were treated with 10 ng/mL TNF alone or in combination with Myc inhibitor, 10058-F4 (10 µM) for 24 hr. 10058-F4 was added 2 hr post TNF stimulation. Expression of *Ifnb1*, *Rsad2*, *Trib3*, and *Chac1* was quantified by the ΔΔCt method using qRT-PCR and expressed as a fold induction compared to the untreated group. 18 S was used as the internal control. (I and J) B6 (I) and B6.Sst1S (J) BMDMs were treated with TNF (10 ng/ml) for 6, 12, and 24 hr in the presence or absence of JNK inhibitor D-JNK1 (2 µM). The cells were harvested and the protein levels of c-Myc and p-cJun were determined by western blotting. JNK inhibitor D-JNK1 was added 2 hr post TNF stimulation. Average densitometric values from two independent experiments were included above the blot. The data are presented as means ± standard deviation (SD) from three to five samples per experiment, representative of three independent experiments. The statistical significance was performed by two-way ANOVA using Šídák's multiple comparison test (Panel A) and ordinary one-way ANOVA using Šídák's multiple comparison test (Panels D-F and H). Significant differences are indicated with asterisks (*, p<0.05; **, p<0.01; ***, p<0.001; ****, p<0.0001).

The online version of this article includes the following source data and figure supplement(s) for figure 5:

**Source data 1.** PDF file containing original western blots for *Figure 5B, C, I, and J* indicating the relevant bands and treatments.

**Source data 2.** Original files for western blot analysis displayed in *Figure 5B, C, I, and J*.

**Figure supplement 1.** Effects of Myc and CSF1R inhibition on B6.Sst1S macrophage activation by TNF.

**Figure supplement 1—source data 1.** PDF file containing original western blots for *Figure 5—figure supplement 1C* indicating the relevant bands and treatments.

**Figure supplement 1—source data 2.** Original files for western blot analysis displayed in *Figure 5—figure supplement 1C*.

## Myc hyperactivity and lipid peroxidation compromise the cell autonomous and T-cell-mediated control of Mtb infection by B6.Sst1S macrophages

Next, we tested whether the described facets of the aberrant macrophage activation conferred by the *sst1S* allele were relevant to Mtb susceptibility. After macrophage infection with virulent Mtb in vitro, gradual accumulation of LPO product 4-HNE adducts was observed in BMDMs of both B6 and B6Sst1S genetic backgrounds at 3–5 days post infection (dpi). It occurred either in the presence or absence of exogenous TNF (*Figure 6A*, and *Figure 6—figure supplement 1A and B*). By day 5 post infection, TNF stimulation significantly increased LPO accumulation only in the B6.Sst1S macrophages (*Figure 6A*). Both Mtb-infected and bystander non-infected B6.Sst1S macrophages showed 4-HNE adduct accumulation (*Figure 6B*).

TNF stimulation improved the control of Mtb growth in B6 but not in B6.Sst1S macrophages, as quantified using an Mtb replication reporter strain (*Figure 6B*, *Figure 6—figure supplement 1C and D*) and quantitative PCR of Mtb genomes (*Figure 6C*). The LPO inhibition using Fer-1 improved the survival of the Mtb-infected BMDMs (*Figure 6D-E*) and prevented the intracellular Mtb growth (*Figure 6F*) during the five-day in vitro infection. The iron chelator DFO also significantly reduced the Mtb growth, although restricting iron availability may also directly affect the bacterial replication. Importantly, the survival of Mtb-infected BMDMs was also improved, and the bacterial loads were significantly reduced by the Myc inhibitor, F4 (*Figure 6G and H*). Of note, the effects of the lipid peroxidation inhibitors became prominent between days 3 and 5 post infection (*Figure 6—figure supplement 1E*), suggesting that these inhibitors do not directly boost the bacterial killing by activated macrophages, but rather prevent macrophage damage.

Next, we tested whether the *sst1* susceptible allele compromised responsiveness of Mtb-infected macrophages to mycobacteria-specific T cells. The immune T cells were isolated from the regional lymph nodes of the resistant B6 mice vaccinated with live attenuated BCG vaccine and added either to the B6 or B.Sst1S BMDM monolayers infected with Mtb the day before. The BMDMs were either

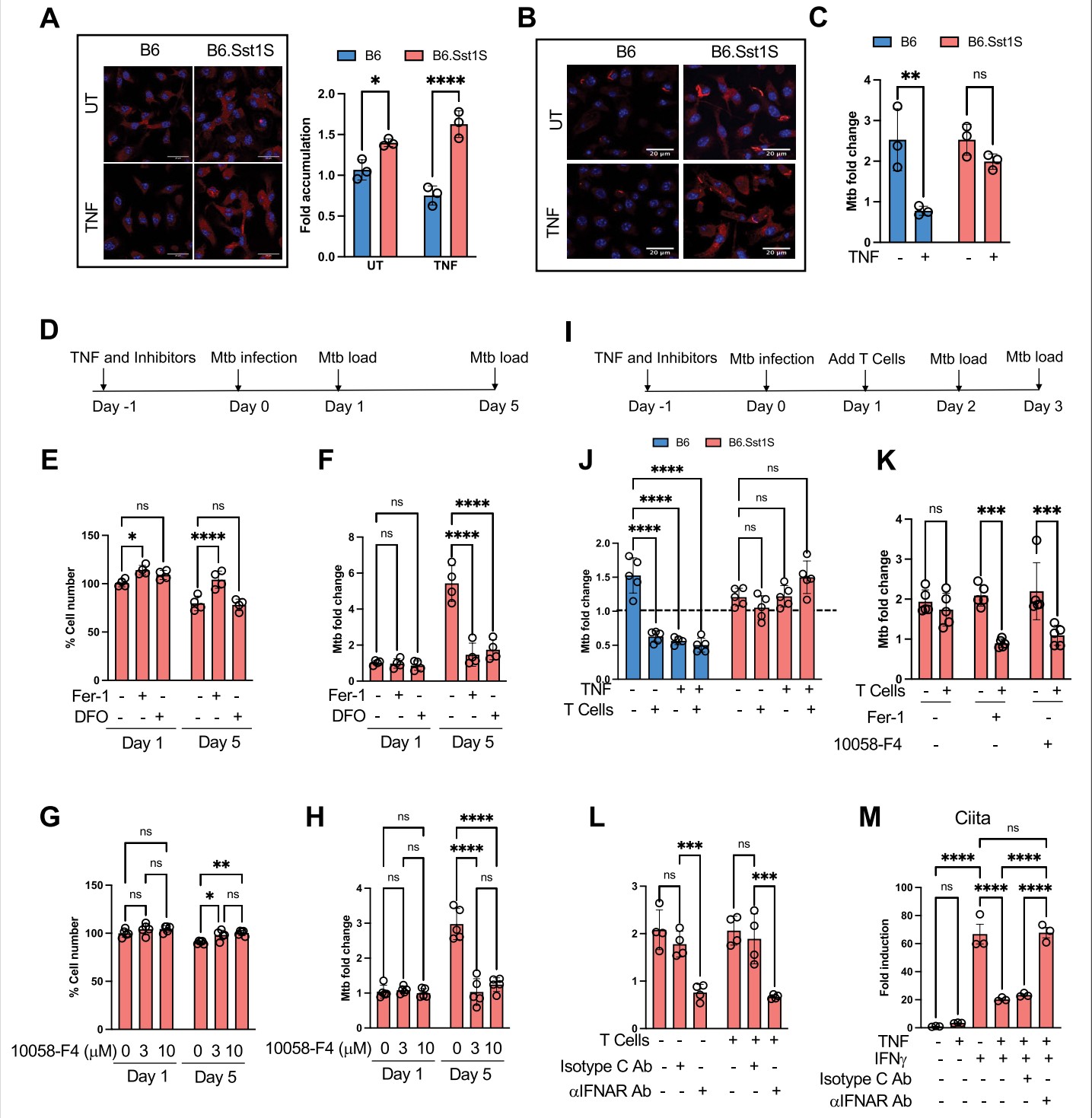

**Figure 6.** Myc and lipid peroxidation compromise control of intracellular Mtb by the B6.Sst1S macrophages. (**A**) Accumulation of 4-HNE adducts in Mtb-infected B6 and B6.Sst1S macrophage monolayers infected with Mtb. BMDMs were either treated with 10 ng/mL TNF or left untreated (UT), and subsequently infected with Mtb at MOI = 1. 4-HNE adducts were detected by confocal microscopy using 4-HNE-specific antibody 5 dpi. The 4-HNE accumulation was quantified at 5 dpi using ImageJ and plotted as fold accumulation compared to untreated B6 (UT). (**B**) Naïve and TNF-stimulated B6 and B6.Sst1S BMDMs were infected with Mtb Erdman reporter strain (SSB-GFP, *smyc*'::mCherry) for 5 days. The accumulation of 4-HNE adducts was detected in both Mtb-infected and non-infected B6.Sst1S cells at day 5 p.i. (**C**) Naive and TNF-stimulated B6.Sst1S BMDMs were infected with Mtb at MOI = 1. At days 4 post infection Mtb load was determined using a qPCR-based method. (**D**) Testing the effects of LPO and Myc inhibitors on the Mtb-infected B6.Sst1S BMDM survival and Mtb control: experimental design for panels E – H. (**E and F**) Prevention of iron-mediated lipid

*Figure 6 continued on next page*

*Figure 6 continued*

peroxidation improves the survival of and Mtb control by the B6.Sst1S macrophages. BMDMs were treated with 10 ng/mL TNF alone in combination with Fer-1 (3 µM) or DFO (50 µM) for 16 hr and subsequently infected with Mtb at MOI = 1. The inhibitors were added after infection for the duration of the experiment. At days 1 and 5 post infection, total cell numbers were quantified using automated microscopy (**E**) and Mtb loads was determined using a qPCR-based method (**F**). The percentage cell number were calculated based on the number of cells at Day 0 (immediately after Mtb infection and washes). The fold change of Mtb was calculated after normalization using Mtb load at Day 0 after infection. (**G and H**) Myc inhibition improves the survival and Mtb control by B6.Sst1S macrophages. BMDMs were treated with 10 ng/mL TNF alone or in combination with 3 µM or 10 µM 10058-F4 for 16 hr and subsequently infected with Mtb at MOI = 1. At days 1 and 5 post infection, total cell numbers were quantified using automated microscopy (**G**) and Mtb loads was determined using a qPCR-based method (**H**). The percentage cell number were calculated based on the number of cells at Day 0 (immediately after Mtb infection). The fold change of Mtb was calculated after normalization using Mtb load at Day 0 after infection and washes. (**I**) LPO and Myc inhibitors improve Mtb control by B6.Sst1S BMDMs co-cultured with BCG-induced T cells: experimental design for panels J – L. (**J**) Differential effect of BCG-induced T cells on Mtb control by B6 and B6.Sst1S macrophages. BMDMs of both backgrounds were treated with 10 ng/mL TNF or left untreated and subsequently infected with Mtb at MOI = 1. T lymphocytes purified from lymph nodes of BCG vaccinated B6 mice were added to the infected macrophage monolayers 24 hr post infection. The Mtb load was calculated by qPCR-based method after 2 days of co-culture with T lymphocytes (3 days post infection). The dotted line indicates the Mtb load in untreated cells at day 2 post infection. (**K**) Inhibition of Myc and lipid peroxidation improves control of Mtb by B6.Sst1S macrophages co-cultured with immune T cells isolated from BCG-vaccinated B6 mice. BMDMs were pretreated with 10 ng/mL TNF alone or in combination with either Fer-1 (3 µM) or 10058-F4 (10 µM) for 16 hr and subsequently infected with Mtb at MOI 1. At 24 hr post infection the lymphocytes from BCG immunized B6 mice were added to the infected macrophage monolayers. The Mtb loads were determined by qPCR based method after 2 days of co-culture with T cells (3 days post infection). (**L**) Inhibition of type I IFN receptor improves control of Mtb by B6.Sst1S macrophages. BMDMs were pretreated with 10 ng/mL TNF alone or in combination with IFNAR1 blocking Ab or isotype C Ab for 16 hr and subsequently infected with Mtb at MOI 1. At 24 hr post infection the lymphocytes from BCG immunized B6 mice were added to the infected macrophage monolayers. The Mtb loads were determined by qPCR-based method after 2 days of co-culture with T cells (3 days post infection). (**M**) TNF stimulation inhibits and IFNAR1 blockade restores the response of B6.Sst1S macrophages to IFNγ. BMDMs were pretreated with TNF (10 ng/mL) for 18 hr and the IFNAR1 blocking Abs or isotype C Ab were added 2 hr after TNF. Subsequently, IFNγ (10 U/mL) was added for additional 12 hr. The expression of the IFNγ-specific target gene *Ciita* was assessed using qRT-PCR. 18 S was used as internal control. The data are presented as means ± standard deviation (SD) from three to five samples per experiment, representative of three independent experiments. The statistical significance was performed by two-way ANOVA using Bonferroni's multiple comparison test (Panels **A, C, J, and K**) and Tukey's multiple comparison test (Panels **E-H** and **L**). One-way ANOVA using Bonferroni's multiple comparison test (Panel **M**). Significant differences are indicated with asterisks (*, $p < 0.05$; **, $p < 0.01$; ***, $p < 0.001$; ****, $p < 0.0001$).

The online version of this article includes the following figure supplement(s) for figure 6:

**Figure supplement 1.** Inhibition of lipid peroxidation improves Mtb control by B6.Sst1S macrophages.

treated with TNF prior to infection or not. After co-culture with the immune T cells for 1 or 2 days, Mtb loads were significantly reduced in T cell co-cultures with the resistant B6 macrophages. The susceptible B6.Sst1S BMDMs did not respond to the same T cells either in the presence or absence of exogenous TNF (*Figure 6I-J* and *Figure 6—figure supplement 1F*). The B6.Sst1S BMDM responsiveness to T cells was improved by inhibitors of lipid peroxidation (Fer1) and Myc (F4): the bacterial loads were significantly reduced 48 hr after co-culture with the immune T cells (*Figure 6K*, *Figure 6—figure supplement 1G*). This T-cell-mediated Mtb control was specific for T cells isolated from BCG-immunized mice (*Figure 6—figure supplement 1H*). The IFNAR1 blockade improved the ability of TNF-stimulated B6.Sst1S macrophages to control Mtb with and without T help (*Figure 6L*, *Figure 6—figure supplement 1I*). It also restored their responsiveness to soluble IFN-γ that was inhibited by pre-stimulation with TNF (*Figure 6M*). These data demonstrate that during prolonged TNF stimulation of B6.Sst1S macrophages, the Myc-driven lipid peroxidation and subsequent IFN-I hyperactivity compromise both the cell autonomous and T-cell-mediated Mtb control.

## Loss of Mtb control in pulmonary TB lesions is associated with the accumulation of lipid peroxidation products and stress escalation in intralesional macrophages

We wanted to determine whether the aberrantly activated macrophages accumulate within pulmonary TB lesions in vivo. We used a mouse model of pulmonary TB where the lung lesions develop after hematogenous spread from the primary site of infection and progress exclusively in the lungs, despite systemic immunity and control of infection in other organs. Microscopic pulmonary lesions develop in the lungs of both B6 and B6.Sst1S mice, but advanced multibacillary TB lesions develop exclusively in the B6.Sst1S (*Yabaji et al., 2025a*; *Yabaji et al., 2025b*).

Based on Mtb loads, TB lesions were classified in two categories: the Mtb-controlling paucibacillary lesions and multibacillary lesions, in which the control of Mtb growth was compromised (*Figure 7— figure supplement 1A and B*). The 4-HNE adduct levels dramatically increased in multibacillary lesions (*Figure 7A* and *Figure 7—figure supplement 2A*). The majority of the 4-HNE + cells were CD11b + myeloid cells (*Figure 7B* and *Figure 7—figure supplement 2B*).

To characterize macrophages in TB lesions and identify pathways associated with the loss of Mtb control, we performed spatial transcriptomics analysis of intralesional macrophages (Iba1+) using the Nanostring GeoMX Digital Spatial Profiler (DSP) system (*Merritt et al., 2020*; *Figure 7—figure supplement 3A-C*).

Comparing the Iba1 + macrophage gene expression profiles in the multibacillary vs paucibacillary lesions, we identified 192 upregulated and 376 downregulated genes at a two and above fold change (*Supplementary file 7*). Pathway analysis demonstrated a highly significant upregulation of Hypoxia, TNF, and IL6/STAT3 Signaling, Glycolysis, Complement, and Coagulation pathways in the multibacillary lesions consistent with the escalation of hypoxia, inflammation, and macrophage activation in the advanced lesions. Mechanistically, top transcription factors associated with genes upregulated in the multibacillary lesions were NFKB1, JUN, STAT1, STAT3, and SP1 (*Figure 7—figure supplement 4A-C*).

To specifically interrogate the interferon pathways in paucibacillary vs multibacillary lesions, we compiled a list of 430 interferon type I and type II inducible genes from public databases that were also included in Nanostring Whole Transcriptome Analysis (WTA) probes. Among these, 70 genes were differentially regulated between the Iba1 + macrophages in multi- vs paucibacillary lesions (*Figure 7C*). Among the upregulated genes were metalloprotease *Mmp12, IL6,* and *Socs3*, chemokines *Cxcl10, Ccl2, Ccl3, Ccl4, Ccl19*, cell stress and senescence marker p21 (*Cdkn1a*), and many known IFN-I target genes. The most upregulated pathways in Iba1 + cells within multibacillary lesions were interferon, TNF, and IL6/STAT3 signaling, and top transcription factors associated with upregulation of these pathways were NFKB1, STAT3, STAT1, and IRF1 (*Figure 7—figure supplement 4D and E*).

Among the IFN-inducible genes upregulated in paucibacillary lesions were *Ifi44l*, a recently described negative regulator of IFN-I that enhances control of Mtb in human macrophages (*DeDiego et al., 2019*; *Jiang et al., 2021*) and *Ciita*, a regulator of MHC class II inducible by Ifnγ, but not IFN-I (*Supplementary file 8*). Thus, the loss of local Mtb control in advanced pulmonary TB lesions was associated with the accumulation of aberrantly activated macrophages that contained lipid peroxidation products, with upregulation of the IFN-I pathway and downregulation of Ifnγ-inducible genes.

To detect the Ifnβ-expressing cells within TB lesions, we introduced the *Ifnb1*-YFP reporter described previously (*Scheu et al., 2008*) in the B6.Sst1S background. The B6.Sst1S,*Ifnb1*-YFP reporter mice were infected with virulent Mtb (*smyc'*:: mCherry) constitutively expressing the red fluorescent protein reporter (*Lavin and Tan, 2022*). We validated the Ifnβ reporter in vitro using co-staining of TNF-stimulated B6.Sst1S, *Ifnb1*-YFP BMDMs with YFP-specific antibodies (*Figure 7— figure supplement 5A*). We demonstrated the accumulation of Ifnβ-expressing cells in pulmonary TB lesions of B6.Sst1S, *Ifnb1*-YFP mice using both the YFP reporter and in situ hybridization with the *Ifnb1* probe (*Figure 7—figure supplement 5B*). The majority of the YFP + cells in TB lesions were Iba1 + macrophages a fraction of which were iNOS positive (*Figure 7D*, *Figure 7—figure supplement 6A*). The mice with paucibacillary lesions had 62% and multibacillary lesions had 80% activated macrophages expressing YFP. These findings clearly demonstrated the production of Ifnβ by activated M1-like macrophages.

Next, we assessed the expression of stress markers phospho-cJun and Chac1 in total, IFN-I producing, and activated macrophages in TB lesions. These markers were primarily expressed by activated macrophages (Iba1+) expressing iNOS and/or Ifnβ (YFP+)(*Figure 7E*, *Figure 7—figure supplement 6B-F*). We also documented the upregulation of PKR in the multibacillary lesions, which is consistent with the upregulation of the upstream IFN-I and downstream Integrated Stress Response (ISR) pathways, as described in our previous studies (*Bhattacharya et al., 2021*; *Figure 7—figure supplement 7A*).

Thus, progression from the Mtb-controlling paucibacillary to non-controlling multibacillary TB lesions in the lungs of TB-susceptible mice was mechanistically linked with a pathological state of macrophage activation characterized by escalating stress (as evidenced by the upregulation of phospho-cJun, PKR, and Chac1), the upregulation of Ifnβ and the IFN-I pathway hyperactivity, with a concurrent reduction of Ifnγ responses. In our in vitro experiments, these stressed macrophages were

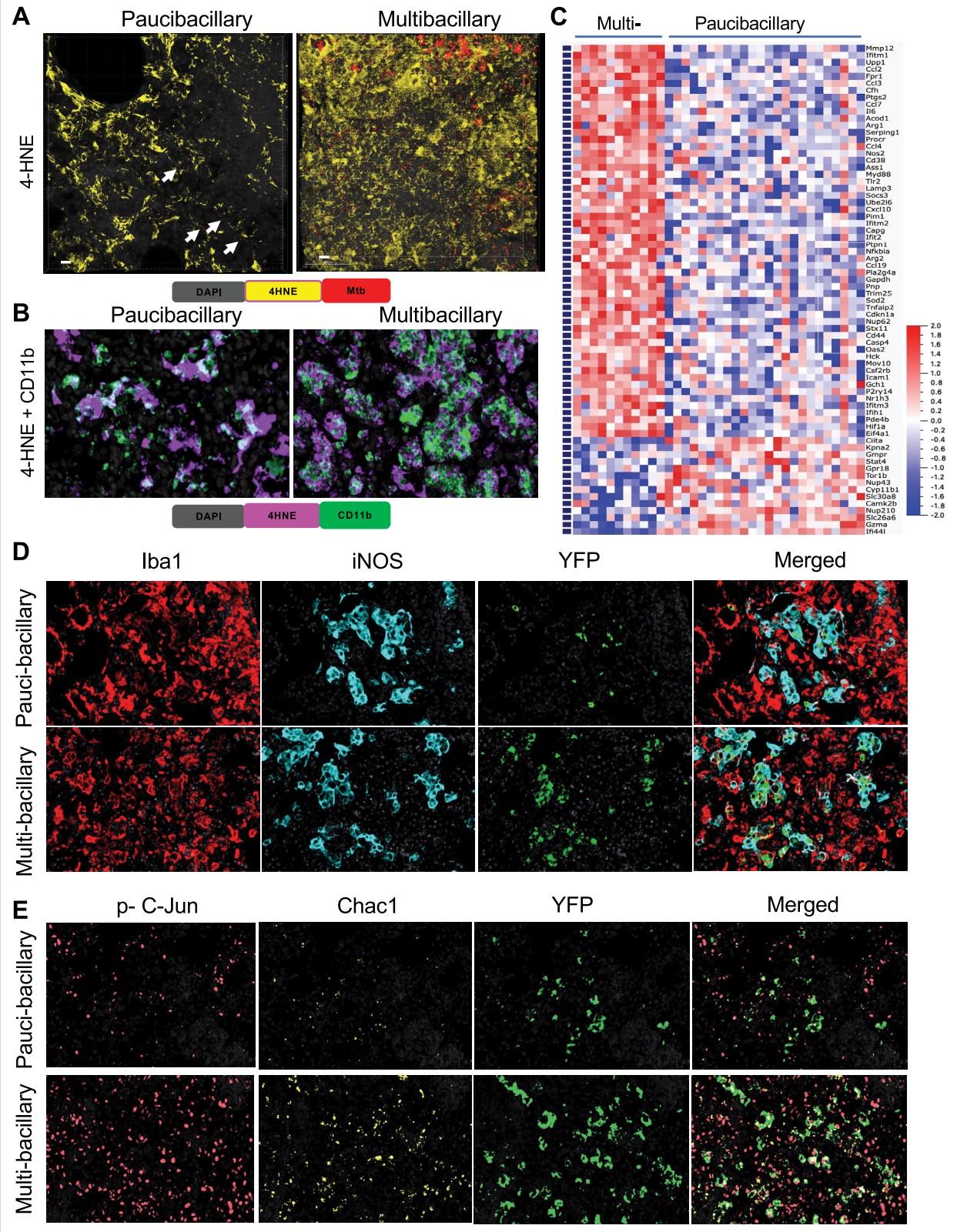

**Figure 7.** Accumulation of lipid peroxidation products and stress escalation in macrophages during pulmonary TB progression. (**A**) Representative 3D confocal images of paucibacillary (n=16) and multibacillary (n=16) pulmonary TB lesions of B6.Sst1S,*Ifnb1* -YFP reporter mice stained with anti-4-HNE antibody (yellow). Cells expressing YFP are green, Mtb reporter Mtb (smyc':: mCherry) is red. Arrows indicate Mtb reporter strain expressing mCherry. The mice were infected for 20 weeks. (**B**) Representative fluorescent multiplexed immunohistochemistry (fmIHC) images of pauci-bacillary and multi-

*Figure 7 continued on next page*

*Figure 7 continued*

bacillary PTB lesion in B6.Sst1S mice at high magnification (×600). 4-HNE adducts (magenta), CD11b (green), and DAPI (gray). White areas showing 4-HNE adducts and CD11b co-localization. The mice were infected for 20 weeks. (**C**) Heatmap of interferon-inducible genes differentially expressed in Iba1 + cells within multibacillary vs paucibacillary lesions (fold change 1.5 and above). Pooled gene list of IFN type I and II regulated genes was assembled using public databases (as described in Materials and methods). The mice were infected for 14 weeks. (**D**) Representative fmIHC images of IFN-I producing (YFP positive) myeloid cells in pauci-bacillary (n=8) and multi-bacillary (n=8) lesion of B6.Sst1S,*Ifnb1*-YFP reporter mice. The different markers are shown as Iba1 (red), iNOS (teal), and YFP (green) at ×400 magnification. The mice were infected for 20 weeks. (**E**) Representative fmIHC images of IFN-I producing (YFP positive) cells accumulating stress markers in pauci-bacillary (n=6) and multi-bacillary (n=9) lesion of B6.Sst1S,*Ifnb1*-YFP reporter mice. The different markers are shown as phospho-c-Jun (peach), Chac-1 (yellow), and YFP (green) at ×400 original magnification. The mice were infected for 20 weeks.

The online version of this article includes the following figure supplement(s) for figure 7:

**Figure supplement 1.** Pauci- and multi-bacillary pulmonary lesions of Mtb-infected B6.Sst1S mice.

**Figure supplement 2.** Accumulation of 4-HNE and Ifnβ producing cells in Mtb-infected B6.Sst1S mouse lung lesions.

**Figure supplement 3.** Representative images of GeoMX Region of Interests (ROIs).

**Figure supplement 4.** Analysis of spatial transcriptomics data from Iba1 + cells in pauci- and multi-bacillary lesions in Mtb-infected B6.Sst1S mouse lungs.

**Figure supplement 5.** Accumulation of 4-HNE and Ifnβ producing cells in Mtb-infected B6.Sst1S mouse lung lesions.

**Figure supplement 6.** Fluorescent multiplexed immunohistochemistry (fmIHC) images representing increased expression of stress markers in macrophages within multi-bacillary pulmonary TB lesions of B6.Sst1S mice.

**Figure supplement 7.** Expression of stress markers in pauci- and multi-bacillary lesions of Mtb-infected B6.Sst1S mouse lungs.

unresponsive to T cell help (*Figure 6J*). Consequently, the administration of BCG vaccine to Mtb-infected B6.Sst1S mice did not increase their survival (*Figure 7—figure supplement 7B*).

## Myc upregulated genes are enriched in TB patients who fail treatment

Next, we tested whether the upregulation of Myc pathway is associated with pulmonary TB progression in human TB patients. We used blood samples obtained from 41 individuals recently (<90 days) diagnosed with TB that were enrolled in the Regional Prospective Observational Research for Tuberculosis (RePORT)-India consortium. Patients were infected with drug-susceptible Mtb and treated with rifampicin, isoniazid, ethambutol, and pyrazinamide, per Technical and Operational Guidelines for TB Control by the Ministry of Health and Family Welfare, Government of India, 2016. Blood samples were collected before or within a week of the antibiotic treatment commencement. Patients were monitored for two years during and post-treatment. Individuals who failed the antibiotic treatment (n=21) were identified by positive sputum culture or clinical diagnosis of symptoms at any time after 4 full months of treatment and symptoms determined to not be from another cause. Cured TB patients (controls) remained culture-negative and symptom-negative for the 2-year observation period.

Several oncogene signatures identified previously (*Bild et al., 2006*) were analyzed with the TBSignatureProfiler (*Johnson et al., 2021*) to determine their ability to differentiate between treatment failures and controls based on bootstrapped AUC scores. Myc_up ranked within the top 3 signatures with an AUC of 0.74 and p-value of 0.008 (*Figure 8A*). Boxplots of ssGSEA scores were created to determine whether the myc signatures were differentially enriched between treatment failures and controls. Myc_upregulated genes were enriched in the treatment failure group relative to the treatment control group (*Figure 8B*, *Supplementary file 9*). The ROC curve used to generate the AUC scores for myc_up is depicted in *Figure 8C* and a heatmap of the genes used to generate the ssGSEA scores is depicted in *Figure 8D*, with ssGSEA scores of each patient sample. These data indicate that the upregulation of Myc pathway in peripheral blood cells of TB patients was associated with poor prognosis and TB persistence even in patients infected with antibiotic-sensitive Mtb. Although pathological evaluation was not included in this study, treatment failures in immunocompetent patients are often associated with massive fibro-necrotic pulmonary TB lesions.

## Discussion

This study revealed a mechanistic connection between susceptibility to TB conferred by the *sst1* locus, hyperactivity of the IFN-I pathway, and unresolving stress in activated macrophages. We have shown

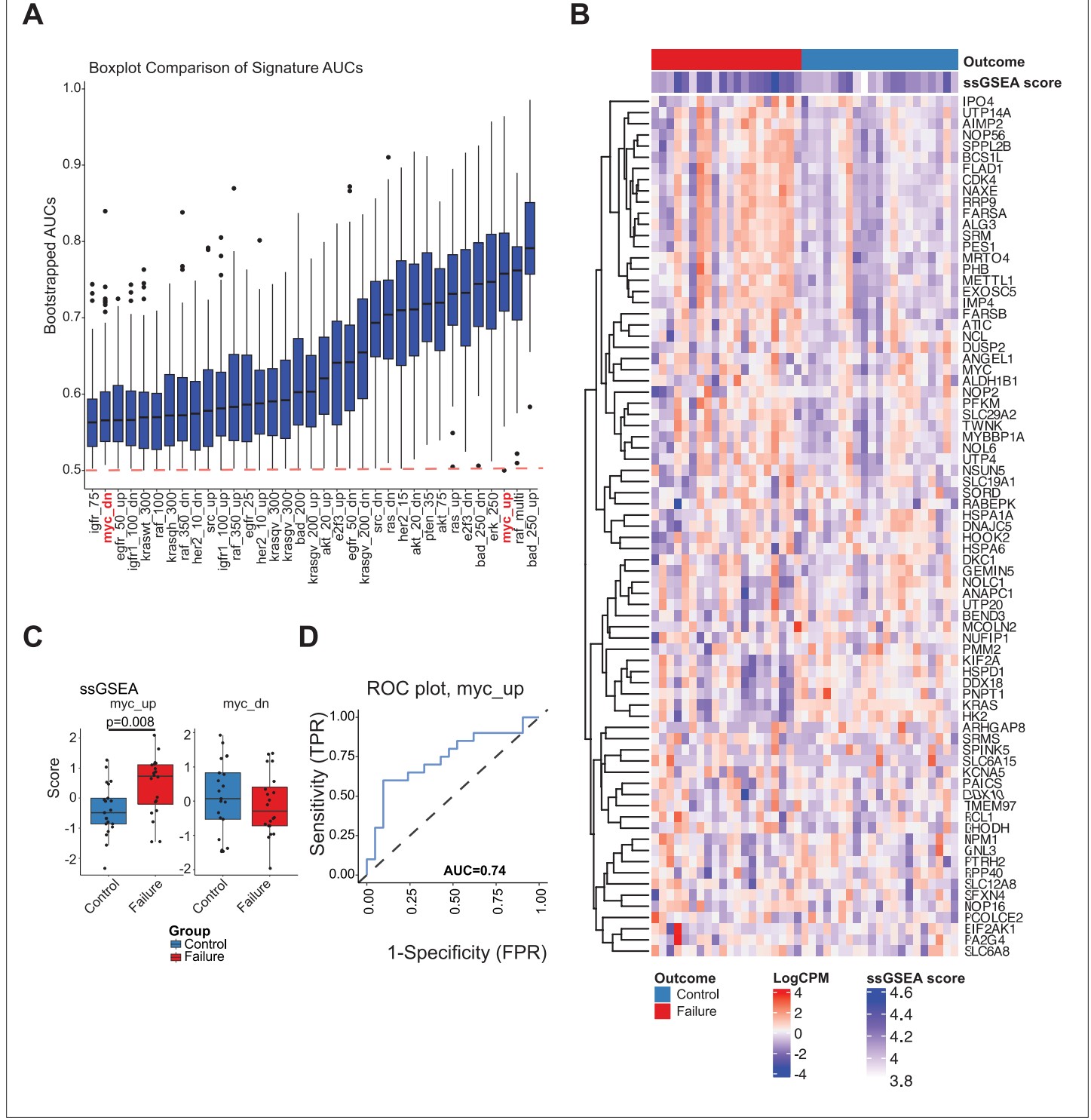

**Figure 8.** Myc upregulated gene signature in peripheral blood of TB patients is associated with treatment failures. (**A**) PBMC gene expression profiling of TB patients prior to TB treatment. Boxplot of bootstrapped ssGSEA enrichment AUC scores from several oncogene signatures ranked from lowest to highest area under curve (AUC) score (Y-axis), with *Myc*_up and *Myc*_dn gene sets (X-axis) highlighted in red. (**B and C**) Boxplots of myc upregulated (*Myc*_up) and downregulated (*Myc*_dn) gene signatures in successful (control) or failed (failed) TB therapy groups, with the receiver-operating characteristic (ROC) curve of *Myc*_up depicted in C. (**D**) Heatmap of all genes utilized in ssGSEA enrichment of the myc_up signature is depicted with individual ssGSEA scores for each patient sample. All plots were generated with the TBSignatureProfiler in R, and p-values were determined by a Student's t-test.

that the aberrant activation of the *sst1*-susceptible macrophages and persistent stress in response to TNF form a vicious cycle shaped by disbalance of anabolic (Myc), homeostatic (antioxidant defense and proteostasis), and immune pathways (TNF and type I IFN).

Previous studies demonstrated that the IFN-I pathway hyperactivity underlies the Mtb susceptibility mediated by the *sst1* locus (*Moreira-Teixeira et al., 2018*; *O'Garra et al., 2013*). Indeed, the IFN-I pathway inactivation in B6.Sst1S mice increased their resistance to Mtb infection, but it did not restore it to the wildtype B6 level (*Ji et al., 2019*). Thus, the IFN-I pathway hyperactivity was insufficient to explain the *sst1*-mediated susceptibility.

Exploring additional mechanisms controlled by the *sst1* locus in activated macrophages, we revealed deficient activation of a fraction of the antioxidant defense genes, including genes involved in glutathione and thioredoxin antioxidant systems, NADPH regeneration, and ferroptosis resistance, such as ferritin light and heavy chains *Ftl* and *Fth*, subunits of glutamate–cysteine ligase subunits *Gclm*, glutathione peroxidases *Gpx1* and *Gpx4*, and stearoyl-Coenzyme A desaturase 2 *Scd2*. These experimental findings are consistent with an unbiased computational analysis that considered genes co-regulated with the *sst1*-encoded *Sp110* and *Sp140* genes in mouse macrophage datasets and suggested their primary association with antioxidant response pathways.

Blockade of IFNAR1 did not restore the AOD gene expression. In contrast, ROS scavengers, iron chelators, and inhibitors of lipid peroxidation prevented the *Ifnb1* superinduction, suggesting that the dysregulated oxidative stress response of B6.Sst1S macrophages was upstream of the IFN-I pathway hyperactivity. Boosting antioxidant defense in B6.Sst1S mice by knockout of the *Bach1* gene, a negative regulator of Nrf2, not only reduced lung inflammatory damage, but also decreased the expression of the IFN-I pathway genes (*Amaral et al., 2024*).

Taken together, these data are consistent with our previous observations that the initial *Ifnb1* superinduction in the *sst1*-susceptible macrophages was driven by the synergy of TNF/NFκB and a JNK pathway activated by oxidative stress (*Bhattacharya et al., 2021*). Here, we found that the prolonged TNF stimulation of the *sst1* mutant macrophages led to the IFN-I-dependent accumulation of toxic low-molecular-weight lipid peroxidation products MDA and 4-HNE adducts. The *Ifnb1* superinduction also resulted in the downstream activation of the integrated stress response markers Trb3 and Chac1. Thus, the IFN-I pathway did not initiate, but amplified the AOD dysregulation.

Among pathways upregulated in TNF-stimulated B6.Sst1S macrophages was the Myc pathway. In actively growing cells, Myc promotes ribosome biogenesis, cap-dependent protein translation, and also suppresses expression of ferritins, most likely to provide labile iron for anabolic metabolism (*Torti and Torti, 2002*; *Wu et al., 1999*). Myc hyperactivity is associated with cell senescence and maladaptive activity in the mTOR pathway (*Alic and Partridge, 2015*; *Hofmann et al., 2015*). In macrophages, Myc promotes the development of myeloid-derived suppressor cells and alternatively activated macrophages (*Kumar et al., 2016*; *Pello et al., 2012*). Here, we have shown that Myc hyperactivity was responsible for the initiation of the aberrant macrophage activation trajectory leading to increased intracellular labile iron pool (Fe$^{+2}$) and lipid peroxidation. Myc inhibition also prevented the *Ifnb1* superinduction and activation of the integrated stress response markers *Trb3* and *Chac1*.

The above findings allowed us to reconstruct the regulatory cascade driving the aberrant macrophage activation (*Figure 9*). During prolonged TNF stimulation of the B6.Sst1S macrophages, Myc hyperactivity and the impairment of AOD lead to a coordinated downregulation of the Ferritin heavy and light chains and lipid peroxidation inhibitor genes. In the absence of terminators, the intracellular peroxidation of polyunsaturated fatty acids (PUFA) is catalyzed by ferrous iron (Fe$^{+2}$) via the Fenton reaction and proceeds in an autocatalytic manner damaging cell membranes containing tightly packed PUFA (*Mortensen et al., 2023*). This autocatalytic lipid peroxidation fuels persistent oxidative stress and, likely, sustains the activity of JNK. Downstream, *Ifnb1* superinduction in B6.Sst1S macrophages leads to the upregulation of interferon-inducible genes, including PKR. The subsequent PKR-dependent ISR activation leads to the upregulation of Chac1 (*Bhattacharya et al., 2021*) - a glutathione degrading enzyme gamma-glutamylcyclotransferase 1 (*Crawford et al., 2015*) that further compromises the AOD. This stepwise escalation eventually locks the susceptible macrophages in a state of unresolving stress, which is maintained by continuous stimulation with TNF and boosted by IFN-I signaling. Thus, during prolonged stimulation of the B6.Sst1S macrophages with TNF, the autocatalytic lipid peroxidation and IFN-I hyperactivity form a positive feedback loop sustaining the unresolving oxidative stress. Unlike ferroptosis, however, this did not immediately result in massive

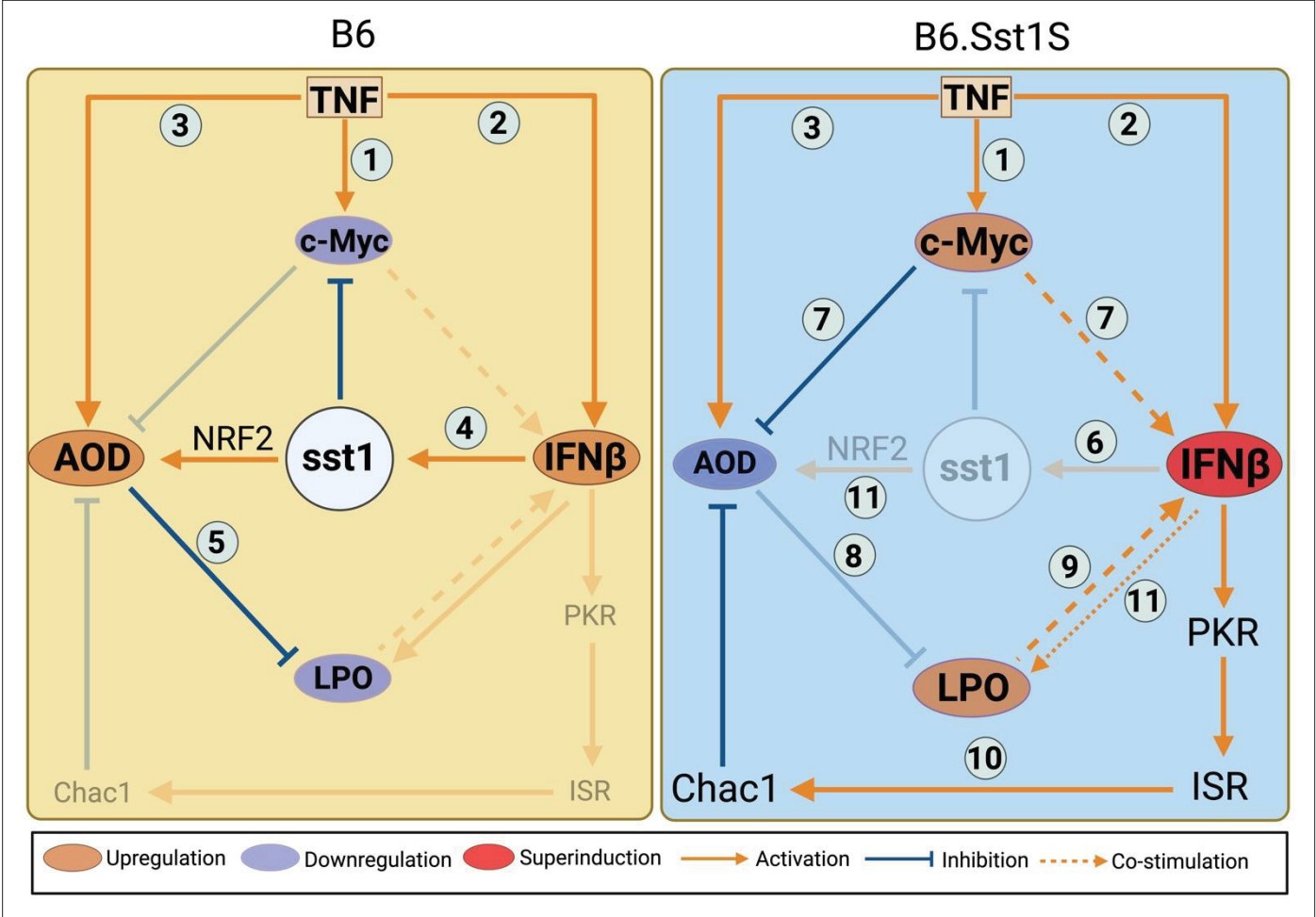

**Figure 9.** Schematic representation of B6 and B6.Sst1S macrophage responses to TNF. Common for B6 and B6.Sst1S: 1. TNF activates c-Myc expression. 2. TNF induces moderate *Ifnb1* expression. 3. TNF stimulation upregulates Nrf2. B6-specific: 4. Ifnβ induces the sst1-encoded SP110 and SP140 nuclear proteins that co-activate Nrf2 and suppress c-Myc. 5. Nrf2 activates antioxidant defense (AOD) that inhibits lipid peroxidation (LPO). B6.Sst1S-specific: 6. *Sp110* and *Sp140* are not expressed. 7. Myc is upregulated, inhibits ferritin expression, and coactivates *Ifnb1* transcription. 8. Deficient AOD activation coupled with increased labile iron pool promotes accumulation of LPO products. 9. LPO further co-stimulates *Ifnb1* superinduction. 10. IFN-I hyperactivity activates ISR and induces Chac1 that further inhibits the AOD and increases LPO. 11. Alternative mechanisms of IFN-mediated dysregulation of AOD defense and iron homeostasis.

cell death but sustained a state of the aberrant macrophage activation that reduced their ability to control Mtb both in a cell-autonomous and a T-cell-dependent manner. In contrast, in the wild type B6 macrophages, Myc was inhibited more readily, and ferritin proteins were upregulated to sequester free iron in parallel with the upregulation of Nrf2. Their combined effects increased stress resilience of activated macrophages. Recent in vivo studies also demonstrated that ferritins and antioxidant pathways were upregulated in inflammatory macrophages isolated from Mtb-infected wild type B6 mice (*Pisu et al., 2020*).

At the molecular level, the detailed characterization of the *sst1*-mediated regulation of macrophage activation provides proximal phenotypes for deeper understanding of the functions of the *sst1*-encoded Sp110 and Sp140 proteins. The *Sp110* and *Sp140* genes and proteins are upregulated exclusively in the wild type cells 8–12 hr after TNF stimulation, that is the time interval during which the divergence between the wild type and mutant macrophages occurs (*Bhattacharya et al., 2021*). Their transcription and protein stability are regulated by interferons and stress kinases (*Fraschilla and Jeffrey, 2020a*). Their activities can modulate (i) responses to inflammatory stimuli via CARD domain (*Fraschilla and Jeffrey, 2020b*), (ii) metabolism, via the activated nuclear receptor interaction domain LXXLL, (iii) regulation of chromatin silencing (*Mehta et al., 2017*), transcriptional elongation

(*Amatullah et al., 2022*) via chromatin interacting PHD, BRD, and SAND domains and indirect regulation of the *Ifnb1* mRNA stability (*Witt et al., 2024*). Potentially, they may interact with intracellular nucleic acids via the DNA interaction domain SAND (*Bottomley et al., 2001*; *Carles and Fletcher, 2010*).

Our data suggest that one of them might be involved in regulation of the Nrf2 protein biosynthesis either at the levels of *Nfe2l1/2* mRNA splicing (*Goldstein et al., 2016*) or cap-independent translation (*Li et al., 2010*; *Shay et al., 2012*; *Figure 2*). Recent experimental evidence suggests that a specialized translation machinery is involved in context-dependent regulation of protein biosynthesis (*Shi and Barna, 2015*). Ribosome modifications, such as protein composition and rRNA methylation, were shown to be involved in control of selective IRES-dependent translation in myeloid cells (*Basu et al., 2011*; *Mazumder et al., 2003*) and regulation of inflammation (*Basu et al., 2014*; *Poddar et al., 2013*; *Poddar et al., 2016*). Recently, enrichment of Sp100 and Sp110 proteins in ribosomes was observed during B6 macrophage activation with LPS (*Susanto et al., 2023*). These findings suggest that the Sp110 protein may be involved in fine-tuning of protein biosynthesis in activated macrophages favoring the IRES-dependent Nrf2 translation.

More generally, the Sp110 and Sp140 proteins may serve as context-dependent regulators (possibly, both as sensors and tunable controllers) of macrophage activation and stress resilience via a coordinated upregulation of AOD and downregulation of Myc and IFN-I pathways. Hypothetically, downregulating this mechanism may also play physiological roles in eliminating infected (*Rothchild et al., 2019*) or transformed cells (*Gorbunova et al., 2012*; *Leonova et al., 2013*). By coordinately increasing the free iron pool for non-enzymatic free radical generation and simultaneously decreasing the buffering capacity of intracellular antioxidants, this activation state promotes generation and unopposed spread of free radicals. However, the inappropriate activation of its built-in amplification mechanisms, such as IFN-I, may drive immunopathologies in the face of chronic stimulation, as occurs with Mtb infection.

In the mouse model of chronic TB infection, lipid peroxidation products did not appear to significantly affect the long-term Mtb survival: the mycobacterial cell wall and multiple mycobacterial antioxidant pathways exist to allow for the survival of, at least, a fraction of mycobacterial population in highly oxidative environments (*Pacl et al., 2018*; *Piddington et al., 2001*). The accumulation of stressed macrophages and LPO products within TB lesions, however, gradually degraded local immunity that allowed for Mtb replication and provided a fodder for growing mycobacteria (*Mahamed et al., 2017*). Thus, in the mouse model that recapitulates key pathomorphological features of pulmonary TB in human patients, the aberrant macrophage activation state was responsible for both the inflammatory tissue damage and the failure of local immunity.

The above mechanistic dissection of the aberrant macrophage activation drivers provides therapeutic targets for interrupting their vicious cycle in vivo. For example, monocytes recruited to inflammatory lesions may be particularly vulnerable to the dysregulation of Myc. Prior to terminal differentiation within an inflammatory milieu, these immature cells undergo several cycles of replication. Cell growth requires labile iron that is provided by Myc via downregulation of ferritins (*Stockwell et al., 2017*; *Torti and Torti, 2002*). Recently, we found that lung epithelial cells embedded within TB lesions express CSF1 – a macrophage growth factor known to stimulate Myc expression (*Yabaji et al., 2025b*). Thus, the coincidence of TNF and CSF1 – Myc stimulation may initiate and propagate the aberrant activation state in monocytes recruited to chronic pulmonary TB lesions. Similarly, growth factors produced locally within inflamed tissues may sustain Myc hyperactivity and the aberrant activation state of monocytes/macrophages in other chronic inflammatory pathologies.

Further understanding of the specific roles of the SP100 family proteins, as well as other signals and checkpoints that regulate the pathological macrophage activation state transitions, will allow for their rational therapeutic modulation - to boost host defenses and mitigate the collateral tissue damage.

## Resource availability

### Lead contact

Further information and requests for resources and reagents should be directed to and will be fulfilled by the lead contact, Igor Kramnik (ikramnik@bu.edu).

## Materials availability

Mouse strains (B6J.C3-*Sst1*$^{C3HeB/FeJ}$Krmn, B6.Sst1S,*Ifnb1*-YFP) and all unique/stable reagents generated in this study are available from the Lead Contact with a completed Materials Transfer Agreement. Mouse strain (B6J.C3-*Sst1*$^{C3HeB/FeJ}$Krmn is available from https://www.mmrrc.org (Stock No: 043908-UNC)).

## Inclusion and diversity

We support inclusive, diverse, and equitable conduct of research.

# Materials and methods

## Reagents

Recombinant mouse TNF (Cat# 315–01 A) and IL-3 (Cat# 213–13) were procured from Peprotech. The mouse monoclonal antibody to mouse TNF (MAb; clone XT22), isotype control, and mouse monoclonal antibody to mouse Ifnβ (Clone: MAR1-5A3) were obtained from Thermo Fisher Scientific. BHA (Cat# B1253) and Deferoxamine mesylate (Cat#D9533) were sourced from Sigma Aldrich. PLX3397 (Cat# S7818), Myc inhibitor (10058-F4) (Cat# S7153), and ferrostatin-1 (Cat# S7243) were purchased from Selleckchem. D-JNK1 (Cat# HY-P0069), GW2580 (Cat# HY-10917), and BLZ945 (Cat# HY-12768) were acquired from Med Chem Express. Fetal bovine serum (FBS) for cell culture medium was sourced from HyClone. Middlebrook 7H9 and 7H10 mycobacterial growth media were purchased from BD and prepared following the manufacturer's instructions. A 50 µg/mL hygromycin solution was utilized for reporter strains of Mtb Erdman (SSB-GFP, smyc'::mCherry).

## Experimental animals

C57BL/6 J inbred mice were obtained from the Jackson Laboratory (Bar Harbor, Maine, USA). The B6J.C3-*Sst1*$^{C3HeB/FeJ}$ Krmn congenic mice were created by transferring the *sst1* susceptible allele from C3HeB/FeJ mouse strain on the B6 (C57BL/6 J) genetic background using twelve backcrosses (referred to as B6.Sst1S in the text). The B6.Sst1S,*Ifnb1*-YFP mice were produced by breeding B6.Sst1S mice with a reporter mouse containing a Yellow Fluorescent Protein (YFP) reporter gene inserted after the *Ifnb1* gene promoter (*Scheu et al., 2008*). The YFP serves as a substitute for *Ifnb1* expression. All animal procedures were approved by the Institutional Animal Care and Use Committee (IACUC) of Boston University (PROTO201800218). Mice were euthanized by $CO_2$ asphyxiation in accordance with IACUC-approved protocols.

## BMDMs culture and treatment

The isolation of mouse bone marrow and the cultivation of bone marrow-derived macrophages (BMDMs) from C57BL/6 J and B6.Sst1S were conducted following the procedures outlined in a prior study (*Yabaji et al., 2022*). BMDMs were plated in tissue culture plates, followed by treatment with TNF and incubation for 16 hr at 37 °C with 5% CO2. Various inhibitors were applied to the cells until the point of harvest.

## Infection of BMDM with *M. tuberculosis*

For infection experiments, *M. tuberculosis* H37Rv (Mtb) was cultured in 7H9 liquid media for 3 days and subsequently harvested. The bacteria were then diluted in media containing 10% L929 cell culture medium (LCCM) without antibiotics to achieve the desired Multiplicity of Infection (MOI). Following this, 100 µL of Mtb-containing media with the specified MOI was added to BMDMs cultivated in a 96-well plate format that had been pre-treated with TNF or inhibitors. The plates underwent centrifugation at 500 x *g* for 5 min, followed by an incubation period of 1 hr at 37 °C. To eliminate any extracellular bacteria, cells were treated with Amikacin at 200 µg/µL for 45 min. Subsequently, cells were washed and maintained with inhibitors, as applicable, in DMEM/F12 containing 10% FBS medium without antibiotics at 37 °C in 5% CO2 for the duration of each experiment. Media changes and inhibitor replacements were carried out every 48 hr. The confirmation of MOIs was achieved by counting colonies on 7H10 agar plates. All procedures involving live *M. tuberculosis* were conducted within Biosafety Level 3 containment, in accordance with Boston University Institutional Biosafety Committee (IBC) approved protocol #25–875, and adhering to regulations from Environmental Health and Safety

at the National Emerging Infectious Disease Laboratories, the Boston Public Health Commission, and the Centers for Disease Control and Prevention.

## Cytotoxicity and mycobacterial growth assays

The cytotoxicity, cell loss, and Mtb growth assays were conducted in accordance with the procedures outlined by *Yabaji et al., 2022*. In brief, BMDMs were subjected to inhibitor treatment or Mtb infection as previously described for specified time points. Upon harvesting, samples were treated with Live-or-Dye Fixable Viability Stain (Biotium) at a 1:1000 dilution in 1 X PBS/1% FBS for 30 min. Following staining, samples were carefully washed to prevent any loss of dead cells from the plate and subsequently treated with 4% paraformaldehyde (PFA) for 30 min. After rinsing off the fixative, samples were replaced with 1 X PBS. Following decontamination, the sample plates were safely removed from containment. Utilizing the Operetta CLS HCA System (PerkinElmer), both infected and uninfected cells were quantified. The intracellular bacterial load was assessed through quantitative PCR (qPCR) employing a specific set of Mtb and *M. bovis*-BCG primer/probes, with BCG spikes included as an internal control, as previously described (*Yabaji et al., 2022*). The percentage of cell numbers was calculated based on the number of cells at Day 0 (immediately after Mtb infection). The fold change of Mtb was calculated after normalization using Mtb load at Day 0 after infection and washes. We quantified the percentage of replicating Mtb using a replication reporter strain, Mtb Erdman (SSB-GFP, *smyc'*::mCherry). Replicating Mtb was identified by counting the number of SSB-GFP puncta-positive Mtb, while the total number of Mtb per field was determined by counting mCherry-expressing bacteria (red).

## RNA isolation and quantitative PCR

The extraction of total RNA was carried out utilizing the RNeasy Plus mini kit (QIAGEN). Subsequently, cDNA synthesis was conducted using the Invitrogen SuperScript III First-Strand Synthesis SuperMix (Cat#18080400). Real-time PCR was executed with the GoTaq qPCR Mastermix (Promega) employing the CFX-96 real-time PCR System (Bio-Rad). Oligonucleotide primers were designed using Integrated DNA Technologies. Either 18 S or β-actin expression served as an internal control, and the quantification of fold induction was computed using the △△Ct method.

## Analysis of RNA sequencing data

Raw sequence reads were mapped to the reference mouse genome build 38 (GRCm38) by STAR (*Dobin et al., 2013*). The read count per gene for analysis was generated by featureCounts (*Liao et al., 2014*). Read counts were normalized to the number of fragments per kilobase of transcript per million mapped reads (FPKM) using the DESeq2 Bioconductor R package (*Love et al., 2014*). Pathway analysis was performed using the GSEA method implemented in the camera function from the limma R package (*Wu and Smyth, 2012*). Databases KEGG, MSigDB Hallmark, Reactome were used for the GSEA analysis. Transcripts expressed in either 3 conditions with FPKM >1 were included in the pathway analyses. To infer mice macrophage gene regulatory network, we used ARCHS4 collected list of RNE-seq data for mice macrophage cells from Gene Expression Omnibus (GEO; *Lachmann et al., 2018*). The total number of analyzed experimental datasets was 1960. This gene expression matrix was utilized as input for the GENIE3 algorithm, which calculated the most likely co-expression partners for transcription factors. The list of transcription factors was derived from the Animal TFDB 3.0 database (*Hu et al., 2019*). The Virtual Inference of Protein Activity by Enriched Regulon Analysis (VIPER) algorithm was used to estimate the prioritized list of transcription factors (*Alvarez et al., 2016*). The software Cytoscape (version 3.4.0) was used for network visualization (*Shannon et al., 2003*). The i-cisTarget web service was used to identify transcription factor binding motifs that were over-represented on a gene list (*Imrichová et al., 2015*). Statistical analysis and data processing were performed with R version 3.6.1 (https://www.r-project.org/) and RStudio version 1.1.383 (https://www.rstudio.com). Raw data were deposited in NCBI's Gene Expression Omnibus (GEO, accession number-GSE164698).

## Generation and sequencing of single-cell RNA libraries

Single-cell suspensions of mouse bone-marrow-derived macrophages were loaded on a 10 X genomics chip G at concentrations varying between 1,240,000–1,575,000 cells/mL aiming for a targeted cell

recovery of 5000 cells per sample and processed using the 10 X Genomics 3'v3.1 Dual index kit according to the manufacturer protocol (10 x Genomics, CA, USA). Following the 10 X Genomics Chromium controller run, the Gel-beads in emulsion (GEMs) were transferred to strip tubes and subsequently put into a thermal cycler using the following parameters: 45 min at 53 °C, 5 min at 85 °C, hold at 4 °C. The samples were then stored in a –20 °C freezer overnight. The next day, samples were thawed at room temperature before adding the recovery agent to break the GEMs. Subsequently, the cDNA was purified and amplified, and the gene expression libraries were generated and purified. The size distribution and molarity of these libraries were assessed using the Bioanalyzer High Sensitivity DNA Assay (Agilent Technologies, Lexington, MA, USA). The libraries were then pooled at 5 nM and sequenced on an Illumina NextSeq 2000 instrument at a 600pM input and 2% PhiX spike-in using a P3 100 cycle flow cell (Illumina, San Diego, CA, USA) resulting in 36,000–58,000 reads per cell.

## Processing of scRNA-seq data

The 10 x Genomics Cell Ranger pipeline v6.0.1 was used for demultiplexing, alignment, identification of cells, and counting of unique molecular indices (UMIs). The Cell Ranger mkfastq pipeline was used to demultiplex raw base call (BCL) files generated by Illumina sequencers into FASTQ files. The Cell Ranger count pipeline was used to perform alignment and create UMI count matrices using reference genome mm10 (Ensembl 84) and parameters – expect-cells = 5000. Droplets with at least 500 UMIs underwent further quality control with the SCTK-QC pipeline (*Hong et al., 2022*). The median number of UMIs was 9927, the median number of genes detected was 2637, the median percentage of mitochondrial reads was 5.43%, and the median contamination estimated by decontX was 0.17 across cells from all samples (*Yang et al., 2020*). Cells with less than 500 counts, less than 500 genes detected, or more than 25% mitochondrial counts were excluded, leaving a total of 14,658 cells for the downstream analysis.

## Clustering of single-cell data

The Seurat package was used to cluster the cells into subpopulations (*Stuart et al., 2019*). The 2000 most variable genes were selected using the FindVariableFeatures function, after removing features with fewer than three counts in three cells. Cells were embedded in two dimensions with UMAP using the RunUMAP function. The clustering was performed at resolution 0.8, resulting in 15 clusters. Markers for each cluster were identified with the FindAllMarkers function using the Wilcoxon Rank Sum test and default parameters: log fold-change threshold = 0.25 and min.pct=0.1. The trajectory analysis was performed using the slingshot package; the pseudotime trajectory was computed using the slingshot function and selecting the PCA (considering 50 PCA) and unweighted cluster labels as inputs (*Street et al., 2018*). The pathway analysis was performed using the VAM package in combination with MSigDB's Hallmark geneset collection. The gene sets were scored using the vamForSeurat function, and the top pathways were found by running the FindAllMarkers analysis on the CDF assay. The following package versions were used: Seurat v4.0.4, VAM v1.0.0, Slingshot v2.3.0, SingleCellTK v2.4.1, Celda v1.10.0.

## Western blot analysis

Equal amounts of protein extracts were separated by SDS-PAGE and transferred to a PVDF membrane (Millipore). Following a 2 hr blocking period in 5% skim milk in TBS-T buffer [20 mM Tris-HCl (pH 7.5), 150 mM NaCl, and 0.1% Tween20], the membranes were incubated overnight with the primary antibody at 4 °C. Subsequently, bands were acquired using the chemiluminescence (ECL) kit (Thermo Fisher Scientific) and GE ImageQuant LAS-4000 Multi-Mode imager. A loading control, either β-actin or β-tubulin, was assessed on the same membrane. Secondary antibodies used included HRP-conjugated goat anti-mouse and anti-rabbit antibodies. Densiometric analysis was performed by quantification of area using Image J.

## Half-life determination of Nrf2 protein

The B6 and B6.Sst1S BMDM were stimulated with TNF (10 ng/ml) for 6 hr. Cycloheximide (2 uM) was added to the cultures and cells were harvested after 15, 30, 45, 60, 90, and 120 min. Western blotting was carried out for Nrf2 and β-tubulin as described. The densitometry analysis was performed using

ImageJ software (NIH, USA). The linear regression curve was plotted using GraphPad Prism software, and the half-life of Nrf2 protein was derived.

## Electrophoretic mobility shift assay

The nuclear extracts were prepared using a nuclear extraction kit (Signosis Inc) according to the instructions provided. The EMSA was carried out using the EMSA assay kit (Signosis Inc) following the instructions provided. Briefly, 5 µg of nuclear extract was incubated with biotin-conjugated Nrf2 anti-oxidant response element (ARE) probe in 1 X binding buffer for 30 min. The reaction mixture was electrophoresed in 6.5% non-denaturing polyacrylamide gel and then blotted onto a nylon membrane. The protein-bound probe and free probe were immobilized on the membrane by UV-light-mediated cross-linking. After blocking the membrane, the probes were detected using streptavidin-HRP mediated chemiluminescence method. The images were captured using ImageQuant LAS 4000.

## Quantification of intracellular labile iron

The method used to measure Labile Intracellular Iron (LIP) involved the Calcein acetoxymethyl (AM) ester (Invitrogen) quenching technique with some modifications, as described in *Amaral et al., 2019*; *Picard et al., 1998*; *Thomas et al., 1999*. BMDMs were plated in 96-well plates and washed with 1×DPBS before being lysed with 0.05% saponin for 10 min. Next, the cell lysates were incubated with 125 nM calcein AM at 37 °C for 30 min. A negative control was established by incubating the lysates with the iron chelator deferoxamine (Sigma-Aldrich) at room temperature for 30 min, while FeSO4-treated cells served as the positive control. The fluorescence of Calcein AM was measured using a fluorescence microplate reader, and the differential MFI of calcein AM between untreated and deferoxamine-treated samples was calculated to determine intracellular labile iron and represented as fold change.

## MDA assay

The amount of MDA in the BMDMs was measured using a Lipid Peroxidation (MDA) Assay Kit (Abcam, ab118970) according to the manufacturer's protocols. Briefly, $2 \times 10^6$ cells were lysed in Lysis Solution containing butylated hydroxytoluene (BHT). The insoluble fraction was removed by centrifugation, and the supernatant was collected, protein concentrations were estimated and used for analysis. The equal amount of supernatants (based on protein concentration) was mixed with thiobarbituric acid (TBA) solution reconstituted in glacial acetic acid and then incubated at 95 °C for 60 min. The supernatants containing MDA-TBA adduct were added into a 96-well microplate for analysis. A microplate reader was used to measure the absorbance at OD 532 nm.

## Lipid peroxidation by C11-Bodipy 581/591

Briefly, the cells were washed three times with 1 X PBS and treated with 1 µM C11-Bodipy 581/591 dye (Invitrogen) suspended in 1 X PBS for 30 min in the dark at 37 °C. The LPOs were measured fluorometrically using Spectramax M5 microplate reader (Molecular Devices).

## Measurement of total antioxidant capacity

The total antioxidant capacity was quantified using the Antioxidant assay kit (Cayman chemical) according to the instructions provided. A standard curve was generated by using various concentrations of Trolox (0.068–0.495 mM). 10 µg of the cell lysate was used to determine the Trolox equivalent antioxidant capacity. The percentage of induced antioxidant capacity was calculated using the formula (Trolox equivalent$_{TNF\ stimulated}$-Trolox equivalent$_{unstimulated}$)/ Trolox equivalent $_{unstimulated}$ X 100.

## Immunofluorescence imaging

BMDMs were cultured on coverslips and subjected to treatment with or without TNF, followed by processing for inhibitor treatment or Mtb infection. Subsequently, cells were fixed with 4% paraformaldehyde (PFA) for 10 min (non-infected) or 30 min (Mtb infected) at room temperature and then blocked for 60 mins with 1% BSA containing 22.52 mg/mL glycine in PBST (PBS + 0.1% Tween 20). After the blocking step, cells were incubated overnight with primary antibodies (4-HNE, Nrf2, or Nrf-1 specific), washed, and then incubated with Alexa Fluor 594-conjugated Goat anti-Rabbit IgG (H+L) secondary Antibody (Invitrogen) in 1% BSA in the dark for 1 hr. The cells were mounted using

ProlongTM Gold antifade reagent (Thermo Fisher Scientific), and images were captured using an SP5 confocal microscope. All images were processed using ImageJ software.

## Detection of lipid peroxidation by Click-iT linoleamide alkyne (LAA) method

To investigate the lipid peroxidation, the Click-iT lipid peroxidation imaging kit (C10446, Thermo Fisher) was used. The BMDMs were cultured on 12 mm coverslips in 24-well plates. Following a 6 hr of TNF treatment, 50 µM Click-iT linoleamide alkyne (LAA) was added to the cell culture and incubated at 37 °C for 1 hr under 5% CO2. The cells were washed, fixed, permeabilized, blocked, and stained with Alexa fluor 488 azide according to the manufacturer's instructions. Subsequently, the coverslips were mounted onto the microscope slide with ProLongTM Diamond Antifade mountant with DAPI (P36962, Thermo Fisher). The images were captured using the Leica SP5 confocal microscope, and further, the signal quantification was carried out using ImageJ software (Version 1.53 k, NIH).

## Quantification of oxidative stress

B6 and B6.Sst1S cells were seeded in a 96-well plate and stimulated with TNF (10 ng/mL) or left untreated for 6, 24, and 36 hr. For ROS inhibition, cells were treated with BHA (100 µM) 2 hr post TNF stimulation. At each time point, CellROX Oxidative Stress Reagent (Cat#C10444, Invitrogen) at 5 µM final concentration was added to the cells and incubated for 30 min at 37 °C, protected from light. After incubation, cells were washed twice with warm PBS, and fluorescence was analyzed using a fluorescence microscope or flow cytometer. ROS levels were quantified by comparing the fluorescence intensity across treatment groups and time points.

## Analysis of small RNAs in macrophages

Small RNA libraries were prepared from size fractionation of 10 µg of total macrophage RNA on a denaturing polyacrylamide gel to purify 18-35nt small RNAs, and converted into Illumina sequencing libraries with the NEBNext Small RNA Library kit (NEB). Libraries were sequenced on the Illumina NextSeq-550 in the Boston University Microarray and Sequencing Core. We applied long RNAs and small RNAs RNAseq fastq files to a transposon and small RNA genomics analysis pipeline previously developed for mosquitoes (*Dayama et al., 2022*; *Ma et al., 2021*) with the mouse transposon consensus sequences loaded instead. Mouse transposon consensus sequences were downloaded from RepBase (*Bao et al., 2015*), and RNA expression levels were normalized internally to each library's depth by the Reads Per Million counting method.

## Infection of mice with *Mycobacterium tuberculosis* and collection of organs

The subcutaneous infection of mice was conducted following the procedure outlined in *Yabaji et al., 2025b*. In brief, mice were anesthetized using a ketamine-xylazine solution administered intraperitoneally. Each mouse was subcutaneously injected in the hock, specifically in the lateral tarsal region just above the ankle (by placing the animal in the restrainer), with 50 µl of 1 X PBS containing 10^6 CFU of Mtb H37Rv or Mtb Erdman (SSB-GFP, smyc'::mCherry). At the designated time points, the mice were anesthetized, lung perfusion was performed using 1 X PBS/heparin (10 U/ml), and organs were collected.

Female mice were used unless otherwise specified, and mice were randomly assigned to experimental groups. No animals were excluded after enrollment. Investigators were blinded to group assignments during outcome assessment. Sample size was determined based on prior studies; no formal power analysis was conducted.

## Confocal immunofluorescence microscopy of tissue sections

Immunofluorescence of thick lung sections (50–100 µm) was conducted following the detailed procedures outlined earlier (*Yabaji et al., 2025b*). Briefly, lung slices were prepared by embedding formalin-fixed lung lobes in 4% agarose and cutting 50 µm sections using a Leica VT1200S vibratome. Sections were permeabilized in 2% Triton X-100 for 1 day at room temperature, followed by washing and blocking with 3% BSA in PBS and 0.1% Triton X-100 for 1 hr. Primary antibodies were applied overnight at room temperature, followed by washing and incubation with secondary antibodies for

2 hr. Samples were stained with Hoechst solution for nuclei detection, cleared using RapiClear 1.47 solution for 2 days, and mounted with ProLong Gold Antifade Mountant. Imaging was performed using a Leica SP5 spectral confocal microscope. Primary rabbit anti-4-HNE, anti-Iba1, and anti-iNOS polyclonal antibodies were detected using goat anti-rabbit antibodies labeled with Alexa Fluor 647. Custom-designed HCR RNA-FISH probe sets targeting mouse *Ifnb1* mRNA, along with amplifiers and buffers (v3 chemistry), were obtained from Molecular Instruments (www.molecularinstruments.com). Hybridization and amplification were performed according to the manufacturer's instructions, and samples were imaged using a confocal microscope.

## Mycobacterial staining of lung sections

Paraffin-embedded 5 μm sections were stained using New Fuchsin method (Poly Scientific R and D Corp., cat no. K093) and counterstained with methylene blue following the manufacturer's instructions.

## Tissue inactivation, processing, and histopathologic interpretation

Tissue samples were submersion fixed for 48 hr in 10% neutral buffered formalin, processed in a Tissue-Tek VIP-5 automated vacuum infiltration processor (Sakura Finetek, Torrance, CA, USA), followed by paraffin embedding with a HistoCore Arcadia paraffin embedding machine (Leica, Wetzlar, Germany) to generate formalin-fixed, paraffin-embedded (FFPE) blocks, which were sectioned to 5 μm, transferred to positively charged slides, deparaffinized in xylene, and dehydrated in graded ethanol. A subset of slides from each sample was stained with hematoxylin and eosin (H&E), and consensus qualitative histopathology analysis was conducted by a board-certified veterinary pathologist (N.A.C.) to characterize the overall heterogeneity and severity of lesions.

## Chromogenic monoplex immunohistochemistry

A rabbit-specific HRP/DAB detection kit was employed (Abcam catalog #ab64261, Cambridge, United Kingdom). In brief, slides were deparaffinized and rehydrated, endogenous peroxidases were blocked with hydrogen peroxide, antigen retrieval was performed with a citrate buffer for 40 min at 90 °C using a NxGen Decloaking chamber (Biocare Medical, Pacheco, California), non-specific binding was blocked using a kit protein block, the primary antibody was applied at a 1:200 dilution, which cross-reacts with mycobacterium species (Biocare Medical catalog#CP140A,C) and was incubated for 1 hr at room temperature, followed by an anti-rabbit antibody, DAB chromogen, and hematoxylin counterstain. Uninfected mouse lung was examined in parallel under identical conditions with no immunolabeling observed serving as a negative control.

## Multiplex fluorescent immunohistochemistry (mfIHC)

A Ventana Discovery Ultra (Roche, Basel, Switzerland) tissue autostainer was used for brightfield and multiplex fluorescent immunohistochemistry (fmIHC). In brief, tyramide signaling amplification (TSA) was used in an iterative approach to covalently bind Opal fluorophores (Akoya Bioscience, Marlborough, MA) to tyrosine residues in tissue sections, with subsequent heat stripping of primary-secondary antibody complexes until all antibodies were developed. Before multiplex-IHC was performed, each antibody was individually optimized using a single-plex-IHC assay using an appropriate positive control tissue. Optimizations were performed to determine ideal primary antibody dilution, sequential order of antibody development, assignment of each primary antibody to an Opal fluorophore, and fluorophore dilution. Once an optimal protocol was established, 5 μm tissue sections were cut from FFPE lung arrays. All Opal TSA-conjugated fluorophore reactions took place for 20 min. Fluorescent slides were counterstained with spectral DAPI (Akoya Biosciences) for 16 min before being mounted with ProLong gold antifade (Thermo Fisher, Waltham, MA). Antibodies utilized in 4-plex 5 color (DAPI counterstained) analysis included: Iba1, iNOS, YFP, Chac1, and P-c-Jun. All rabbit antibodies were developed with a secondary goat anti-rabbit HRP-polymer antibody (Vector Laboratories, Burlingame, CA) for 20 min at 37° C and all mouse-derived antibodies were developed with a secondary goat anti-mouse HRP-polymer antibody (Vector Laboratories).

## Brightfield immunohistochemistry

Antigen retrieval was conducted using a Tris-based buffer-Cell Conditioning 1 (CC1)-Catalog # 950–124 (Roche). The MHCII primary was of mouse origin, so a goat anti-mouse HRP-polymer antibody (Vector

Laboratories) was utilized. Brightfield slides utilized A ChromoMap DAB (3,3'-Diaminobenzidine) Kit-Catalog #760–159 (Roche) to form a brown precipitate at the site of primary-secondary antibody complexes containing HRP. Slides were counterstained with hematoxylin and mounted.

## Multispectral whole slide microscopy

Fluorescent and chromogen-labeled slides were imaged at 40 X using a Vectra Polaris Quantitative Pathology Imaging System (Akoya Biosciences). For fluorescently labeled slides, exposures for all Opal dyes were set based upon regions of interest with strong signal intensities to minimize exposure times and maximize the specificity of signal detected. A brightfield acquisition protocol at 40 X was used for chromogenically labeled slides.

## Digitalization and linear unmixing of multiplex fluorescent immunohistochemistry

Whole slide fluorescent images were segmented into QPTIFFs, uploaded into Inform software version 2.4.9 (Akoya Biosciences), unmixed using spectral libraries affiliated with each respective opal fluorophore including removal of autofluorescence, then fused together as a single whole slide image in HALO (Indica Labs, Inc, Corrales, NM).

## Quantitative analysis of immunohistochemistry

View settings were adjusted to allow for optimal visibility of immunomarkers and to reduce background signal by setting threshold gates to minimum signal intensities. After optimizing view settings, annotations around the entire tissue were created to define analysis area using the flood tool, and artifacts were excluded using the exclusion pen tool. To analyze lesions independently, a tissue classifier that uses a machine-learning approach was utilized to identify lesions. The outputs from this classifier were annotations for each lesion, which were then isolated into independent layers, which were analyzed separately from other layers. For quantifying myeloid markers in multiplexes, an algorithm called the HALO (v3.4.2986.151, Indica Labs, Albuquerque, NM, USA) Area Quantification (AQ) module was created and finetuned to quantify the immunoreactivity for all targets. Thresholds were set to define positive staining based on a real-time tuning feature. AQ outputted the percentage of total area displaying immunoreactivity across the annotated whole slide scan in micrometers squared ($\mu m^2$). AQ output the total percentage positive for each phenotype. AQ and HP algorithms were run across all layers for each individual lesion and exported as a.csv file.

## Spatial transcriptomics

To perform spatial transcriptomics analysis, we used the Nanostring GeoMX Digital Spatial Profiler (DSP) system (Nanostring, Seattle, WA) (*Danaher et al., 2022*; *Merritt et al., 2020*). To identify pathways dominating early vs. late (advanced) lesions, we selected lungs from 2 mice with paucibacillary early lesions and 2 mice with advanced multibacillary lesions with necrotic areas. Slides were stained with fluorescent CD45-, pan-keratin-specific antibodies, and Iba1-specific antibodies (sc-32725, Santa Cruz Biotechnology, CA, USA) and DAPI. Diseased regions of interest (ROI) for expression profiling of Iba1 + cells were selected to focus on myeloid-rich areas avoiding areas of micronecrosis and tertiary lymphoid tissue (*Figure 7—figure supplement 3*). Eight ROI each of normal lung, early and late lesions (respectively) were studied. The profiling used the Mouse Whole Transcriptome Atlas (WTA) panel which targets ~21,000 + transcripts for mouse protein coding genes plus ERCC-negative controls to profile the whole transcriptome, excluding uninformative high expressing targets such as ribosomal subunits. A subsequent study focused on gene expression within macrophages identified by Iba1 labeling in lung sections showing uncontrolled (advanced multibacillary) lesions in comparison to controlled (paucibacillary stage). Samples from each ROI were packaged into a library for sequencing (NextSeq550, Illumina) following the procedure recommended by Nanostring. After sequencing, the data analysis workflow began with QC evaluation of each ROI based on thresholds for number of raw and aligned reads, sequencing saturation, negative control probe means, and number of nuclei and surface area. Background correction is performed using subtraction of the mean of negative probe counts. Q3 normalization (recommended by Nanostring) results in scaling of all ROIs to the same value for their 3rd quartile value. The count matrix was imported into the Qlucore Genomics Explorer software package (Qlucore, Stockholm, Sweden) and log2 transformed for further analysis. Statistical

analysis was then performed to identify differentially expressed genes (typically unpaired t-test with Benjamini-Hochberg control for FDR rate at q<.05). Lists of differentially expressed genes (DEGs) from comparisons of distinct stages were further analyzed for enrichment of pathways or predicted transcription factors using the Enrichr online tool (*Chen et al., 2013*; *Liberzon et al., 2015*; *Xie et al., 2021*). To specifically explore interferon-related gene expression in Iba1 + macrophages, we used: (1) a list of genes unique to type I genes derived by pooling four common type I and type II interferon gene list respectively and identifying the non-overlapping (unique) genes for each type (see *Supplementary file 8*, *Figure 7—figure supplement 4*); (2) using the GSEA analysis tool of the Qlucore software to evaluate enrichment of the MSigDB Hallmark collection of pathways (which include Interferon-alpha and Interferon-gamma, and other pathways of interest [*Liberzon et al., 2015*]). Raw data were deposited in NCBI's Gene Expression Omnibus (GEO, accession number- GSE292392).

## Human blood transcriptome analysis

### Ethics approval

Ethics approval for the study was obtained from the Institutional Ethics Committee of the participating institutions, and written informed consent was obtained prior to enrollment. This study utilized data from four longitudinal observational studies collected at five clinical sites within the Regional Prospective Observational Research for Tuberculosis (RePORT)-India consortium: Byramjee Jeejeebhoy Government Medical College (BJMC), Pune; the Jawaharlal Institute of Postgraduate Medical Education and Research (JIPMER, Puducherry); National Institute for Research in Tuberculosis (NIRT), Chennai; Prof. M. Viswanathan Diabetes Research Centre (MVDRC), Chennai; and the Christian Medical College (CMC), Vellore (*Ayiraveetil et al., 2020*; *Christopher et al., 2021*; *Gupte et al., 2016*; *Kornfeld et al., 2016*).

## Blood sample collection and processing

Participant demographics including age, sex, body mass index (BMI), diabetes status, alcohol use, and tobacco use were recorded and matched for Blood collection. Blood samples were collected from 41 individuals and recently (<90 days) diagnosed with TB, within one week of treatment commencement from five clinical sites within the Regional Prospective Observational Research for Tuberculosis (RePORT)-India consortium. Individuals for this study were newly diagnosed with TB (within 90 days; sputum smear-positive or Xpert MTB/RIF assay positive (Cepheid, Sunnyvale, CA, USA)), at least 15 years of age, multi-drug-resistant negative, and had received less than one week of treatment. Sociodemographic data such as age, sex, behavioral characteristics (smoking, alcohol use (using the Alcohol Use Disorders Identification Test [AUDIT-C])), tobacco use, and body mass index (BMI) were obtained through a questionnaire and the medical history of the participants. There were no significant differences in risk factors for TB or treatment failure, including sex, risky alcohol use, tobacco use, BMI, or diabetes. Patients were monitored for two years during and post-treatment with Rifampicin, isoniazid, ethambutol, and pyrazinamide, per India national guidelines Division CT. Technical and Operational Guidelines for TB Control in India 2016. Central TB Division, Directorate General of Health Service, Ministry of Health and Family Welfare; Government of India, New Delhi. 2016 (*Bush et al., 1998*). Individuals who failed treatment (n=21) were identified by positive sputum culture or clinical diagnosis of symptoms at any time after 4 full months of treatment and symptoms determined to not be from another cause. Treatment controls were determined by those who remained culture-negative and symptom-negative for the two-year observation period.

Blood samples from each person were collected and stored in PAXgene tubes (Cat #762165, BD Biosciences, San Jose, CA, USA) at –80 °C until processing using the PAXgene Blood RNA kit (Cat #762164, QIAGEN, Hilden, Germany). PAXgene tubes were sent to MedGenome (Bangalore, India) for processing. Library preparation and sequencing were performed at the GenoTypic Technology Pvt. Ltd. Genomics facility in Bangalore, India, with the SureSelect Strand-Specific mRNA Library Prep kit (Cat #5190–6411, Agilent, Santa Clara, USA). One µg of RNA was used for mRNA enrichment by poly(A) selection and fragmented using RNAseq Fragmentation Mix containing divalent cations at 94°C for 4 min. Enriched mRNA underwent fragmentation through exposure to RNASeq Fragmentation Mix with divalent cations at 94 °C for 4 min. Subsequently, single-strand cDNA was synthesized in the presence of Actinomycin D (Gibco, Life Technologies, Carlsbad, CA, USA) and purified utilizing HighPrep magnetic beads (Magbio Genomics Inc, USA). Following this step, double-stranded cDNA

was synthesized, and the ends were repaired before undergoing further purification. Adenylation of the 3'-ends of cDNA preceded the ligation of Illumina Universal sequencing adaptors, which were then purified and amplified with 10 PCR cycles. The final cDNA sequencing libraries were purified and quantified using Qubit, while the fragment size distribution was assessed using Agilent TapeStation. Subsequently, the libraries were combined in equimolar proportions, and the resulting multiplexed library pools were sequenced utilizing the Illumina NextSeq 500 platform for 75 bp single-end reads.

## RNA-sequencing data processing of human samples

### QC and alignment

Raw sequencing FASTQ files were assessed for data quality using FastQC[S. A. FastQC: a quality control tool for high-throughput sequence data. Babraham Bioinformatics, Babraham Institute. 2010]. Trimmomatic was used to trim the reads (SLIDINGWINDOW:4:20 LEADING:3 TRAILING:3 MINLEN:36; *Bolger et al., 2014*). Rsubread (*Liao et al., 2013*) was used to align reads to the human genome hg39 and to determine expression counts for each gene. Genes missing (or with 0 counts) in more than 20% of the samples were excluded before batch correction, as well as one outlier identified with Principal Components Analysis (PCA) before batch correction and could not be rectified using batch correction (unpublished data).

## Batch correction and normalization

The RNA samples were processed sequentially in two batches. Batch effects created by combining the two batches were removed using ComBat-Seq (*Zhang et al., 2020*). The ComBat-Seq adjusted counts were normalized using a $\log_2$-counts per million (logCPM) adjustment, and the logCPM values were used for downstream analysis.

## Differential expression

Differential expression was performed on batch-corrected counts/logcpm using limma (*Ritchie et al., 2015*) and signature genes were selected from the limma results to obtain logFC, p-values, and p-adjusted values (using a Benjamini-Hochberg false-discovery rate).

## Oncogene signatures

Normal Human mammary epithelial cells (HMEC) were transfected with adenoviruses containing either the gene of interest or GFP as described in *Bild et al., 2006*; *McQuerry et al., 2019*. Differentially expressed genes between transfected cells and controls with a Benjamini-Hochberg cutoff of < 0.05 were used in for the signatures of each transfected oncogene.

## TBSignatureProfiler platform

Heatmaps, boxplots, receiver-operating characteristic (ROC) curve, and area under the ROC curve (AUC) were calculated and depicted using the functions within the TBSignatureProfiler (*Johnson et al., 2021*). Bootstrapping was used to iteratively calculate AUC values using leave-one-out cross-validation to obtain mean AUCs and 95% confidence intervals (CI) for 100 repeats for each signature. We generated mean AUC values for the predictive performance of collected oncogene-specific signatures (*Bild et al., 2006*; *McQuerry et al., 2019*) in terms of their ability to distinguish between treatment-failure individuals and control samples in our cohort. The single sample Gene Set Enrichment Analysis (ssGSEA; *Barbie et al., 2009*) was used to generate enrichment scores for each signature.

## Statistical analysis

Statistical analyses were performed using GraphPad Prism 9 software (RRID:SCR_002798). Differences among groups involving two or more variables were assessed using two-way analysis of variance (ANOVA) with adjustments for multiple post hoc comparisons. For comparisons across multiple groups based on a single variable, one-way ANOVA with post hoc testing was applied. Two-tailed paired or unpaired t-tests were used for comparisons between two groups after verifying data normality. For non-parametric datasets, the Wilcoxon Rank Sum test (Mann-Whitney U test equivalent) was employed. Sample sizes were chosen based on prior studies using the same infection model; no formal power calculation was performed. Animals were randomly assigned to experimental groups.

Statistical significance was defined as p<0.05. Significance levels are indicated as follows: *, p<0.05; **, p < 0.01; ***, p < 0.001; and ****, p < 0.0001.

## Acknowledgements

The authors would like to acknowledge expert support of Boston University Avedisian and Chobanian School of Medicine Sequencing and Single Cell Sequencing Cores. This work was supported by the National Institutes of Health grant R01HL126066 (to IK), R01HL133190 (to IK and WRB), National Institutes of Health grant R01CA244660 and EU grant no. 101136926 MULTIR" (to BNK), and National Institutes of Health grant R01GM114864 (to AAG). NIH grants R01-AG052465 and R01-GM135215 to NCL. National Library of Medicine grant R01LM013154 to JDC, NIH grant R01: 5R01GM127430-07 to WEJ, Health Research Board, Ireland, grant number ERATRANSCAN-2022-001 (to VZ), Health Research Board, Ireland, grant number EPPerMed-2024-1 (to VZ).

## Additional information

### Competing interests

Lester Kobzik: employee of Cellecta, Inc. The other authors declare that no competing interests exist.

### Funding

| Funder | Grant reference number | Author |
| --- | --- | --- |
| National Heart Lung and Blood Institute | R01HL126066 | Igor Kramnik |
| National Heart Lung and Blood Institute | R01HL133190 | Igor Kramnik William R Bishai |
| National Cancer Institute | R01CA244660 | Boris N Kholodenko |
| Horizon Europe | 10.3030/101136926 | Boris N Kholodenko |
| National Institute of General Medical Sciences | R01GM114864 | Alexander A Gimelbrant |
| National Institute on Aging | R01-AG052465 | Nelson C Lau |
| National Institute of General Medical Sciences | R01-GM135215 | Nelson C Lau |
| National Library of Medicine | R01LM013154 | Joshua D Campbell |
| National Institute of General Medical Sciences | R01: 5R01GM127430-07 | W Evan Johnson |
| Health Research Board, Ireland | ERATRANSCAN-2022-001 | Vadim Zhernovkov |
| Health Research Board, Ireland | EPPerMed-2024-1 | Vadim Zhernovkov |
| Boston University | S10OD030269 | Nicholas A Crossland |
| Boston University | S10OD026983 | Nicholas A Crossland |

The funders had no role in study design, data collection and interpretation, or the decision to submit the work for publication.

### Author contributions

Shivraj M Yabaji, Conceptualization, Data curation, Formal analysis, Validation, Investigation, Visualization, Methodology, Writing – original draft; Vadim Zhernovkov, Oleksii S Rukhlenko, Formal analysis, Validation, Methodology; Prasanna Babu Araveti, Formal analysis, Validation, Investigation, Methodology, Writing – original draft; Suruchi Lata, Investigation, Methodology; Salam Al Abdullatif, Arthur Vanvalkenburg, Qicheng Ma, Formal analysis, Validation; Yuriy O Alekseyev, Methodology; Gargi

Dayama, Formal analysis, Validation, Investigation; Nelson C Lau, Alexander A Gimelbrant, Resources, Formal analysis, Funding acquisition, Validation, Methodology; W Evan Johnson, Conceptualization, Resources, Funding acquisition, Methodology, Writing – review and editing; William R Bishai, Conceptualization, Funding acquisition, Writing – original draft; Nicholas A Crossland, Joshua D Campbell, Resources, Methodology; Boris N Kholodenko, Resources, Formal analysis, Funding acquisition, Validation; Lester Kobzik, Conceptualization, Resources, Formal analysis, Funding acquisition, Validation, Methodology, Writing – original draft, Writing – review and editing; Igor Kramnik, Conceptualization, Resources, Supervision, Funding acquisition, Writing – original draft, Project administration, Writing – review and editing

### Author ORCIDs
Shivraj M Yabaji ⬡ https://orcid.org/0000-0002-5793-1661
Vadim Zhernovkov ⬡ https://orcid.org/0000-0001-8903-1142
Prasanna Babu Araveti ⬡ https://orcid.org/0000-0002-5441-3826
Suruchi Lata ⬡ https://orcid.org/0000-0001-5548-9315
Oleksii S Rukhlenko ⬡ https://orcid.org/0000-0003-1863-4987
Arthur Vanvalkenburg ⬡ https://orcid.org/0009-0007-1793-9160
Yuriy O Alekseyev ⬡ https://orcid.org/0000-0001-6105-8861
Nelson C Lau ⬡ https://orcid.org/0000-0001-9907-1404
W Evan Johnson ⬡ https://orcid.org/0000-0002-6247-6595
William R Bishai ⬡ https://orcid.org/0000-0002-8734-4118
Nicholas A Crossland ⬡ https://orcid.org/0000-0003-3873-9188
Joshua D Campbell ⬡ https://orcid.org/0000-0003-0780-8662
Boris N Kholodenko ⬡ https://orcid.org/0000-0002-9483-4975
Alexander A Gimelbrant ⬡ https://orcid.org/0000-0001-6986-0285
Lester Kobzik ⬡ https://orcid.org/0000-0003-4328-937X
Igor Kramnik ⬡ https://orcid.org/0000-0001-6511-9246

### Ethics
Human subjects: Ethics approval for the study was obtained from the Institutional Ethics Committee of the participating institutions, and written informed consent was obtained prior to enrollment. This study utilized data from four longitudinal observational studies collected at five clinical sites within the Regional Prospective Observational Research for Tuberculosis (RePORT)-India consortium: Byramjee Jeejeebhoy Government Medical College (BJMC), Pune; the Jawaharlal Institute of Postgraduate Medical Education and Research (JIPMER, Puducherry); National Institute for Research in Tuberculosis (NIRT), Chennai; Prof. M. Viswanathan Diabetes Research Centre (MVDRC), Chennai; and the Christian Medical College (CMC), Vellore(Ayiraveetil et al, 2020; Christopher et al, 2021; Gupte et al, 2016; Kornfeld et al, 2016).

All animal procedures were approved by the Institutional Animal Care and Use Committee (IACUC) of Boston University (PROTO201800218). Mice were euthanized by CO2 asphyxiation in accordance with IACUC-approved protocols.

Reviewer #1 (Public review): https://doi.org/10.7554/eLife.106814.2.sa1
Author response https://doi.org/10.7554/eLife.106814.2.sa2

---

# Additional files

### Supplementary files
Supplementary file 1. Cell cycle analysis of B6 and B6.Sst1S BMDMS 24 h after TNF stimulation using scRNA-seq.

Supplementary file 2. Cell cycle analysis of B6 and B6.Sst1S specific BMDM subpopulations 24 h after TNF stimulation using scRNA-seq.

Supplementary file 3. Gene set enrichment analysis of differentially activated pathways in B6 and B6.Sst1S BMDMs 12 h after TNF stimulation.

Supplementary file 4. Transcription factor binding sites analysis of differentially expressed genes in B6 and B6.Sst1S BMDMs 12 h after TNF stimulation.

Supplementary file 5. The list of identified transcription factors associated with differences between activated genes in response to TNF stimulation in B6 and B6.Sst1S BMDMs.

Supplementary file 6. Master regulator analysis of the transcription factors associated with differences between activated genes in response to TNF stimulation in B6 and B6.Sst1S BMDMs.

Supplementary file 7. Lists of differentially expressed genes in Iba1 + cells from pauci- and multi-bacillary lesions of Mtb infected B6.Sst1S mouse lungs.

Supplementary file 8. Expression of IFN pathway genes in Iba1 +cells from pauci- and multi-bacillary lesions of Mtb infected B6.Sst1S mouse lungs.

Supplementary file 9. Upregulated Myc pathway genes differentially expressed in peripheral blood cells of human TB patients.

MDAR checklist

## Data availability

The RNA-seq and spatial transcriptomics datasets generated in this study have been deposited in the Gene Expression Omnibus (GEO) under accession numbers GSE164698 and GSE292392.

The following datasets were generated:

| Author(s) | Year | Dataset title | Dataset URL | Database and Identifier |
|---|---|---|---|---|
| Kramnik I, Zhernovkov V, Gimelbrant A | 2023 | Control of macrophage response to TNF by the sst1 locus | https://www.ncbi.nlm.nih.gov/geo/query/acc.cgi?acc=GSE164698 | NCBI Gene Expression Omnibus, GSE164698 |
| Kobzik L, Kramnik I | 2025 | Spatial Transcriptomics of Controlled vs Uncontrolled Tuberculosis in a Mouse Model | https://www.ncbi.nlm.nih.gov/geo/query/acc.cgi?acc=GSE292392 | NCBI Gene Expression Omnibus, GSE292392 |

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

# Appendix 1

**Appendix 1—key resources table**

| Reagent type (species) or resource | Designation | Source or reference | Identifiers | Additional information |
|---|---|---|---|---|
| Antibody | Rabbit polyclonal anti-iNOS antibody | Abcam | Cat# ab15323, RRID:AB_301857 | IHC (1:100) |
| Antibody | Rabbit monoclonal anti-Iba1/AIF-1 (E4O4W) XP antibody | Cell Signaling Technology | Cat# 17198, RRID:AB_2820254 | IHC (1:500) |
| Antibody | Rabbit polyclonal anti-GFP antibody | Invitrogen | Cat# A11122, RRID:AB_221569 | IHC (1:500) IF (1:500) |
| Antibody | Mouse monoclonal anti-IFNAR1 antibody (clone MAR1-5A3), functional grade | Thermo Fisher Scientific | Cat# 16594585, RRID:AB_1210688 | IFN I inhibition (1:200) |
| Antibody | Mouse IgG1 κ isotype control antibody (clone P3.6.2.8.1) | Thermo Fisher Scientific | Cat# 14471482, RRID:AB_470111 | Isotype C Ab (1:200) |
| Antibody | Mouse monoclonal anti-TNFα antibody (clone XT22) | Thermo Fisher Scientific | Cat# MM350D, RRID:AB_223528 | Inhibition of TNF signaling (10 µg/mL) |
| Antibody | Rabbit polyclonal anti-KEAP1 antibody (reactive to human, mouse, rat) | Proteintech Group Inc | Cat# 10503–2-AP, RRID:AB_2132625 | WB (1:5000) |
| Antibody | Rabbit polyclonal anti-NRF2/NFE2L2 antibody (reactive to human, mouse, rat) | Proteintech Group Inc | Cat# 16396–1-AP, RRID:AB_2782956 | WB (1:5000) |
| Antibody | Rabbit polyclonal anti-βTrCP antibody | Proteintech Group Inc | Cat# 28393–1-AP, RRID:AB_2935467 | WB (1:2000) |
| Antibody | Rabbit polyclonal anti-BACH1 antibody | Invitrogen | Cat# PA5-117013, RRID:AB_2901643 | WB (1:1000) |
| Antibody | Rabbit polyclonal anti-Histone H3 antibody | Cell Signaling Technology | Cat# 9715 s, RRID:AB_331563 | WB (1:5000) |
| Antibody | Mouse monoclonal anti-β-tubulin antibody | Santa Cruz Biotechnology | Cat# sc-55529, RRID:AB_2210962 | WB (1:1000) |
| Antibody | Mouse monoclonal anti-β-Actin antibody(Human/Mouse/Rat) | R&D Systems | Cat# MAB8929; RRID:AB_3076436 | WB (1:5000) |
| Antibody | Rabbit polyclonal anti-phospho-cJun antibody | Cell Signaling Technology | Cat# 9261 s, RRID:AB_2130162 | WB (1:1000) |
| Antibody | Rabbit monoclonal anti-c-Myc antibody [Y69] | Abcam | Cat# ab32072, RRID:AB_731658 | WB (1:1000) |
| Antibody | Rabbit monoclonal anti-Ferritin Heavy Chain antibody [EPR18878] | Abcam | Cat# ab183781, RRID:AB_2940987 | WB (1:2000) |
| Antibody | Rabbit monoclonal anti-Ferritin Light Chain antibody [EPR5260] | Abcam | Cat# ab109373, RRID:AB_1086271 | WB (1:5000) |
| Antibody | Rabbit polyclonal anti-4-Hydroxynonenal antibody | Abcam | Cat# ab46545, RRID:AB_722490 | IHC (1:50) IF (1:100) |
| Antibody | Mouse monoclonal anti-NRF1 antibody | Santa Cruz Biotechnology | Cat# sc-515360 | WB (1:500) |
| Antibody | Rabbit polyclonal anti-GPX1 antibody | Proteintech Group Inc | Cat# 29329–1-AP, RRID:AB_2918283 | WB (1:1000) |

*Appendix 1 Continued on next page*

*Appendix 1 Continued*

| Reagent type (species) or resource | Designation | Source or reference | Identifiers | Additional information |
|---|---|---|---|---|
| Antibody | Mouse monoclonal anti-GPX4 antibody | Proteintech Group Inc | Cat# 67763–1-Ig, RRID:AB_2909469 | WB (1:1000) |
| Antibody | Horse anti-mouse IgG, HRP-linked antibody (polyclonal) | Cell Signaling Technology | Cat# 7076 s, RRID:AB_330924 | WB (1:2000) |
| Antibody | Goat anti-rabbit IgG, HRP-linked antibody (polyclonal) | Cell Signaling Technology | Cat# 7074 s, RRID:AB_2099233 | WB (1:2000) |
| Antibody | Goat polyclonal anti-rabbit IgG (H+L), ImmPRESS HRP polymer detection kit | Vector Laboratories | Cat# MP-7451, RRID:AB_2631198 | |
| Antibody | Goat polyclonal anti-mouse IgG (H+L), ImmPRESS HRP polymer detection kit | Vector Laboratories | Cat# MP-7452, RRID:AB_2744550 | |
| Antibody | Goat polyclonal anti-rabbit IgG (H+L), Alexa Fluor 546, cross-adsorbed | Invitrogen | Cat# A-11010, RRID:AB_2534077 | IF (1:200) |
| Antibody | Goat polyclonal anti-rabbit IgG (H+L), Alexa Fluor Plus 647, highly cross-adsorbed | Invitrogen | Cat# A-32733, RRID:AB_2633282 | IF (1:200) |
| Strain, strain background (*Mycobacterium tuberculosis*) | *Mycobacterium tuberculosis* H37Rv (TMC 102) | ATCC | Cat# 27294 | |
| Strain, strain background (*Mycobacterium bovis* BCG) | *Mycobacterium bovis* BCG, TMC 1019 [BCG Japanese] | ATCC | Cat# 35737 | |
| Strain, strain background (*Mycobacterium tuberculosis*) | Erdman(SSB-GFP, *smyc'*::mCherry) | *Lavin and Tan, 2022* | N/A | A gift from Shumin Tan |
| Chemical compound, drug, peptides | ChromoMap DAB Kit | Roche | Cat#760–159 | |
| Chemical compound, drug, peptides | HRP/DAB detection kit | Abcam | Cat# ab64261 | |
| Chemical compound, drug, peptides | Tris based buffer-Cell Conditioning 1 (CC1) | Roche | Cat#950–124 | |
| Chemical compound, drug, peptides | Pexidartinib (PLX3397) | Selleckchem | Cat# S7818 | |
| Chemical compound, drug, peptides | Sotuletinib (BLZ945) | MedChemExpress | Cat# HY-12768 | |
| Chemical compound, drug, peptides | GW-2580 | MedCHemExpress | Cat#HY-10917 | |
| Chemical compound, drug, peptides | 10058-F4 | Selleckchem | Cat# S7153 | |

*Appendix 1 Continued on next page*

*Appendix 1 Continued*

| Reagent type (species) or resource | Designation | Source or reference | Identifiers | Additional information |
|---|---|---|---|---|
| Chemical compound, drug, peptides | Ferrostatin-1 | Selleckchem | Cat#S7243 | |
| Chemical compound, drug, peptides | D-JNK-1 | MedChemExpress | Cat# HY-P0069 | |
| Chemical compound, drug, peptides | Deferoxamine mesylate (DFOM) | Sigma Aldrich | Cat#D9533 | |
| Chemical compound, drug, peptides | Butylated hydroxyanisole (BHA) | Sigma Aldrich | Cat# B1253 | |
| Chemical compound, drug, peptides | Hygromycin B | Roche | Cat# 10843555001 | 50 µg/mL |
| Chemical compound, drug, peptides | Murine IFN-gamma | Peprotech | Cat# 315–05 | |
| Chemical compound, drug, peptides | Murine Interleukin –3 | Peprotech | Cat# 213–13 | |
| Chemical compound, drug, peptides | Murine Interleukin-4 | Peprotech | Cat# 214–14 | |
| Chemical compound, drug, peptides | Murine TNF-alpha | Peprotech | Cat# 315–01 A | |
| Chemical compound, drug, peptides | Middlebrook 7H9 Broth | BD Biosciences | Cat# 271310 | |
| Chemical compound, drug, peptides | Middlebrook 7H10 Agar | BD Biosciences | Cat# 262710 | |
| Chemical compound, drug, peptides | Cycloheximide | Cell Signaling Technology | Cat#2112 | |
| Commercial assay or kit | Live-or-Dye 594/614 Fixable Viability Staining Kits | Biotium | Cat# 32006 | Dilution 1:1000 |
| Commercial assay or kit | TaqMan Environmental Master Mix 2.0 | Fisher Scientific | Cat#4396838–5 mL | |
| Commercial assay or kit | Lipid Peroxidation (MDA) Assay Kit (Colorimetric/Fluorometric) | Abcam | Cat# ab118970 | |
| Commercial assay or kit | BODIPY 581/591 C11 (Lipid Peroxidation Sensor) | Thermo Fisher Scientific | Cat# D3861 | |
| Commercial assay or kit | CellROX Green Reagent | Thermo Fisher Scientific | Cat# C10444 | |
| Commercial assay or kit | Click-iT Lipid Peroxidation Imaging Kit | Thermo Fisher Scientific | Cat# C10446 | |
| Commercial assay or kit | Nuclear Extraction Kit | Signosis | Cat# SK-0001 | |

*Appendix 1 Continued on next page*

*Appendix 1 Continued*

| Reagent type (species) or resource | Designation | Source or reference | Identifiers | Additional information |
|---|---|---|---|---|
| Commercial assay or kit | NRF2(ARE) EMSA Kit | Signosis | Cat# GS-0031 | |
| Commercial assay or kit | HCR *Ifnb1* probe set | Molecular Instruments | N/A | Detection of *Ifnb1* transcripts |
| Commercial assay or kit | HCR Buffers | Molecular instruments | N/A | |
| Commercial assay or kit | Antioxidant Assay Kit | Cayman chemical | Cat# 709001 | |
| Commercial assay or kit | RNeasy plus mini kit | Qiagen | Cat#74136 | |
| Commercial assay or kit | Invitrogen SuperScript III First-Strand Synthesis SuperMix | Invitrogen | Cat#18080400 | |
| Commercial assay or kit | GoTaq qPCR Mastermix | Promega | Cat#A6002 | |
| Commercial assay or kit | PAXgene Blood RNA kit | Qiagen, Hilden, Germany | Cat #762164 | |
| Commercial assay or kit | SureSelect Strand-Specific mRNA Library Prep kit | Agilent, Santa Clara, USA | Cat #5190–6,411 | |
| Commercial assay or kit | HCR RNA-FISH probe set targeting *Ifnb1* mRNA (custom design) | Molecular Instruments | N/A | |
| Commercial assay or kit | HCR RNA-FISH amplifier and buffers (used with *Ifnb1* probe set) | Molecular Instruments | N/A | |
| Strain, strain background (*Mus musculus*) | Mouse: C57BL/6 J, adult male and female | The Jackson Laboratory | Stock No.: 000664, RRID:IMSR_JAX:000664 | https://www.jax.org/strain/000664 |
| Strain, strain background (*Mus musculus*) | Mouse: B6J.C3-*Sst1*[C3HeB/FeJ]Krmn, adult male and female | *Pichugin et al., 2009* | Stock No: 043908-UNC https://www.mmrrc.org | Available at https://www.mmrrc.org |
| Strain, strain background (*Mus musculus*) | Mouse: (C3XB6.Sst1S) F1, adult male and female | This study | N/A | |
| Strain, strain background (*Mus musculus*) | Mouse: B6.Sst1S;*Ifnb1*-YFP, adult male and female | *Scheu et al., 2008*; *Yabaji et al., 2025b* | N/A | YFP-based detection of *Ifnb1* expression |
| Sequence-based reagent | Mtb specific_F | This paper | PCR primers | GGAAATGTCACGTCCATTCATTC |
| Sequence-based reagent | Mtb specific_R | This paper | PCR primers | GCGTTGTTCAGCTCGGTA |
| Sequence-based reagent | Mtb specific probe | This paper | PCR probe | 56-FAM/AGCTTGGTCAGGGACTGCTTCC/36-TAMSp/ |
| Sequence-based reagent | BCG specific_F | This paper | PCR primers | GTGGTGGAGCGGATTTGA |
| Sequence-based reagent | BCG specific_R | This paper | PCR primers | CAACCGGACGGTGATCC |
| Sequence-based reagent | BCG specific probe | This paper | PCR probe | /5Cy5/TTCTGGTCG/TAO/ACGATTGGCACATCC/3IAbRQSp/ |

*Appendix 1 Continued on next page*

*Appendix 1 Continued*

| Reagent type (species) or resource | Designation | Source or reference | Identifiers | Additional information |
|---|---|---|---|---|
| Sequence-based reagent | *Fth*_F | This paper | PCR primers | TGTATGCCTCCTACGTCTATCT |
| Sequence-based reagent | *Fth*_R | This paper | PCR primers | CCTCATGAGATTGGTGGAGAAA |
| Sequence-based reagent | *Ftl*_F | This paper | PCR primers | AGGAGGTGAAACTCATCAAGAA |
| Sequence-based reagent | *Ftl*_R | This paper | PCR primers | TGAGGCGCTCAAAGAGATAC |
| Sequence-based reagent | *Myc*_F | This paper | PCR primers | TCTCCACTCACCAGCACAACTACG |
| Sequence-based reagent | *Myc*_R | This paper | PCR primers | ATCTGCTTCAGGACCCT |
| Sequence-based reagent | *Hmox-1*_F | This paper | PCR primers | CCTTCCCGAACATCGACAGCC |
| Sequence-based reagent | *Hmox-1*_R | This paper | PCR primers | GCAGCTCCTCAAACAGCTCAA |
| Sequence-based reagent | *Nqo1*_F | This paper | PCR primers | CCTCGCTGGAAAAAGAAGTG |
| Sequence-based reagent | *Nqo1*_R | This paper | PCR primers | GGAGAGGATGCTGCTGAAAG |
| Sequence-based reagent | *Nfe2l2*_F | This paper | PCR primers | CCTCGCTGGAAAAAGAAGTG |
| Sequence-based reagent | *Nfe2l2*_R | This paper | PCR primers | GGAGAGGATGCTGCGGAAAG |
| Sequence-based reagent | *Gpx1*_F | This paper | PCR primers | CACCAGGAGAATGGCAAGAA |
| Sequence-based reagent | *Gpx1*_R | This paper | PCR primers | CATTCCGCAGGAAGGTAAAGA |
| Sequence-based reagent | *Ciita*_F | This paper | PCR primers | CTTCAAGCAGCCTCAGTATC |
| Sequence-based reagent | *Ciita*_R | This paper | PCR primers | ATGTGTCCTCTGTCTCATTTAC |
| Sequence-based reagent | *Ifnb1*_F | This paper | PCR primers | ATGAGTGGTGGTTGCAGGC |
| Sequence-based reagent | *Ifnb1*_R | This paper | PCR primers | TGACCTTTCAAATGCAGTAGATTC |
| Sequence-based reagent | *Rsad2*_F | This paper | PCR primers | AAGCTGAGGAGGTGGTGCAG |
| Sequence-based reagent | *Rsad2*_R | This paper | PCR primers | GAAAACCTTCCAGCGCACAG |
| Sequence-based reagent | *Trib3*_F | This paper | PCR primers | GCAAAGCGGCTGATGTCTG |
| Sequence-based reagent | *Trib3*_R | This paper | PCR primers | AGAGTCGTGGAATGGGTATCTG |
| Sequence-based reagent | *Chac1*_F | This paper | PCR primers | CCTGCTACCCTGCTCTTACCT |
| Sequence-based reagent | *Chac1*_R | This paper | PCR primers | GAGCTTGGCTCCTCAGGTC |

*Appendix 1 Continued on next page*

*Appendix 1 Continued*

| Reagent type (species) or resource | Designation | Source or reference | Identifiers | Additional information |
|---|---|---|---|---|
| Sequence-based reagent | *b-actin*_F | This paper | PCR primers | GTGGGCCGCTCTAGGCACCA |
| Sequence-based reagent | *b-actin*_R | This paper | PCR primers | CGGTTGGCCTTAGGGTTCAGGG |
| Sequence-based reagent | *18 S*_F | This paper | PCR primers | TCAAGAACGAAAGTCGGAGGT |
| Sequence-based reagent | *18 S*_R | This paper | PCR primers | CGGGTCATGGGAATAACG |
| Software, algorithm | Graphpad Prism 9.5.1 (528) | Graphpad | https://www.graphpad.com/, RRID:SCR_002798 | |
| Software, algorithm | Microsoft office | Microsoft | https://www.office.com/?auth=2 | |
| Software, algorithm | Halo HighPlex FL v4.2.3 | Indica Labs Inc. | https://indicalab.com/halo/ | |
| Software, algorithm | EndnoteX9 | Clarivate Analytics | https://endnote.com/downloads | |
| Software, algorithm | Imaris Viewer | Oxford Instruments | https://imaris.oxinst.com/microscopy-imaging-software-free-trial?source%20=viewer | |
| Software, algorithm | ImageJ | National Institutes of Health (NIH) | https://imagej.nih.gov/ij/, SCR_003070 | Image analysis |
| Software, algorithm | STAR | STAR | RRID:SCR_004463 | |
| Software, algorithm | featureCounts | featureCounts | RRID:SCR_012919 | |
| Software, algorithm | DESeq2 | DESeq2 | RRID:SCR_015687 | |
| Software, algorithm | limma | limma | RRID:SCR_010943 | |
| Software, algorithm | GSEA | GSEA | RRID:SCR_003199 | |
| Software, algorithm | Seurat | Seurat | RRID:SCR_007322 | |
| Software, algorithm | Cytoscape | Cytoscape | RRID:SCR_003032 | |
| Software, algorithm | Trimmomatic | Trimmomatic | RRID:SCR_011848 | |
| Software, algorithm | GEO | GEO | RRID:SCR_005012 | |
| Software, algorithm | GENIE3 | GENIE3 | RRID:SCR_000217 | |
| Software, algorithm | RStudio | RStudio | RRID:SCR_000432 | |
| Software, algorithm | Enrichr | Enrichr | RRID:SCR_001575 | |

*Appendix 1 Continued on next page*

*Appendix 1 Continued*

| Reagent type (species) or resource | Designation | Source or reference | Identifiers | Additional information |
|---|---|---|---|---|
| Other | Operetta CLS HCA System | Operetta | https://www.perkinelmer.com/in/lab-solutions/product/operetta-cls-system-hh16000020 | |
| Other | Vibratome | Leica VT1200 S | https://www.leicabiosystems.com/us/research/vibratomes/leica-vt1200/ | |
| Other | SP5 Confocal Microscope | Leica | N/A | |
| Other | LAS-4000 | FujiFilm | N/A | |
| Other | Automate in vivo manual gravity perfusion system for mice double 140 mL – IV 4140 | Braintree Scientific, Inc | Cat# IV 4140 | |
| Other | Rapiclear 1.47 | SunJin Lab Co. | Cat# NC1660944 | |
| Other | ProLong Gold Antifade Mountant | Invitrogen | Cat# P36934 | |
| Other | Hoechst 33342 | Fisher Scientific | Cat# H3570 | 10 µg/mL |
| Other | Paraformaldehyde Solution 4% in PBS | Fisher Scientific | Cat# J19943-K2 | |
| Other | L-Glutamine | Corning | Cat# 25–005 CI | |
| Other | Penicillin Streptomycin solution | Corning | Cat# 30–002 CI | |
| Other | HEPES buffer | Corning | Cat# 25–060 CI | |
| Other | L929 Cell Conditioned Media (LCCM) | This paper | N/A | |
| Other | Lymphoprep (1.077 A) | STEMCELL | Cat#07801 | |
| Other | Poly Ethylene Glycol (PEG), Bioultra-8000 | Sigma | Cat#89510 | |
| Other | 5 M NaCl | Invitrogen | Cat#AM9759 | |
| Other | Tris Hydrochloride, 1 M solutions (pH 8.0) | Fisher Scientific | Cat#77-86-1 | |
| Other | Ultrapure 0.5 M EDTA pH 8.0 | Invitrogen | Cat#15575–038 | |
| Other | Ambion Nuclease-free Water | Invitrogen | Cat#AM9932 | |
| Other | SpeedBead Magnetic Carboxylate Modified Particles | GE Healthcare | Cat#65152105050250 | |
| Other | DynaMag-96 side | Life Technologies | Cat#12331D | |
| Other | Glycine | Sigma | Cat#50046 | |
| Other | NaOH Solution | Sigma | Cat#72068 | |
| Other | Proteinase K | Ambion | Cat#AM2546 | |
| Other | Middlebrook 7H9 Broth | BD Biosciences | Cat# 271310 | |
| Other | Middlebrook 7H10 Agar | BD Biosciences | Cat# 262710 | |

