## [Editor Report · eLife Assessment]

Yabaji et al. reports a **fundamental** study highlighting the mechanistic connection for susceptibility to TB infection via the sst1 locus, this was shown to involve increased IFN and Myc production causing the down-regulation of anti-oxidant defence genes and chronic lipidation. Ultimately, lipid peroxidation may underlie infectivity and macrophage dysfunction. Overall, the data presented are **compelling**, supported by a well designed multi-omics approach and the findings will be of broad interest to researchers investigating the molecular mechanisms of TB infection.

[Editors' note: this paper was reviewed by Review Commons.]

---

## [Referee Report · Reviewer #1 (Public review)]

Summary:

In this report, Yabaji et al describe studies designed to address the mechanism behind the TB susceptibility gene sst1. This locus is known to affect expression of IFN and synergizes with Myc to potentiate infectivity. Using a variety of molecular expression and imaging techniques, the authors demonstrate that mice harboring an sst1 transgene (compared to B6 controls) are highly susceptible to TB infection via a mechanism involving loss of antioxidant defense systems, the down regulation of key antioxidant genes and ferritin controlling intracellular iron levels. The combination of increased iron plus decreased antioxidant defense systems in turn increases lipid peroxidation and downstream sequelae. Inhibition of peroxidation diminishes infectivity increases ferritin levels. Furthermore, the authors demonstrate that Myc activation potentiates this process and that down regulation of NRF2 antioxidant defenses accompany potentiated infectivity. Increased peroxidation products (4-HNE) may activate the ASK1/JNK system leading to IFNb superinduction and diminished macrophage viability thereby diminishing ability to withstand TB infection. Extending these findings, additional mouse models plus some work in humans supports the peroxidation hypothesis. Overall, the work is significant for it introduces a molecular basis for TB infectivity and presents a potential novel therapeutic opportunity.

Strengths:

(1) Strengths of this study include a multi-omic analysis of infectivity combining gene expression analysis with biochemical and cell biological evaluation.

(2) Novel identification of an iron-catalyzed lipid peroxidation based mechanism for why the sst1 locus is linked to TB infection.

(3) Parallels to human biology are included via analysis of Myc upregulation in peripheral blood from patients.

(4) Appropriate statistical analysis

Weaknesses:

(1) Lipid peroxidation is a broad phenotype process and the authors honed in on 4-HNE dependent processes as a likely mechanism because they can measure 4-HNE conjugated proteins. However, lipid peroxidation is a complex phenomenon and the work presented herein is largely descriptive.

(2) The authors continually refer to increased 4HNE while they do not measure this 9 carbon lipid, they actually measure 4-HNE conjugated proteins immunochemically.

(3) The authors do not distinguish between increased protein-HNE adducts and increased membrane peroxidation (or both) as mechanistically linked to infectivity.

---

## [Author Response]

General Statements

We are grateful for constructive reviewers’ comments and criticisms and have thoroughly addressed all major and minor comments in the revised manuscript.

Summary of new data.

We have performed the following additional experiments to support our concept:

(1) The kinetcs of ROS production in B6 and B6.Sst1S macrophages after TNF stimulation (Fig. 3I and J, Suppl. Fig. 3G);

(2) Time course of stress kinase activation (Fig.3K) that clearly demonstrated the persistent stress kinase (phospho-ASK1 and phospho-cJUN) activation exclusively in. the B6.Sst1S macrophages;

(3) New Fig.4 C-E panels include comparisons of the B6 and B6.Sst1S macrophage responses to TNF and effects of IFNAR1 blockade in both backgrounds.

(4) We performed new experiments demonstrating that the synthesis of lipid peroxidation products (LPO) occurs in TNF-stimulated macrophages earlier than the IFNβ super-induction (Suppl.Fig.4A and B).

(5) We demonstrated that the IFNAR1 blockade 12, 24 and 32 h after TNF stimulation still reduced the accumulation of LPO product (4-HNE) in TNF-stimulated B6.Sst1S BMDMs (Suppl.Fig.4 E-G).

(6) We added comparison of cMyc expression between the wild type B6 and B6.Sst1S BMDMs during TNF stimulation for 6-24 h (Fig.5I-J).

(7) New data comparing 4-HNE levels in Mtb-infected B6 wild type and B6.Sst1S macrophages and quantification of replicating Mtb was added (Fig.6B, Suppl.Fig.7C and D).

(8) In vivo data described in Fig.7 was thoroughly revised and new data was included. We demonstrated increased 4-HNE loads in multibacillary lesions (Fig.7A, Suppl. Fig.9A) and the 4-HNE accumulation in CD11b+ myeloid cells (Fig.7B and Suppl.Fig.9B). We demonstrated that the Ifnb – expressing cells are activated iNOS+ macrophages (Fig.7D and Suppl.Fig.13A). Using new fluorescent multiplex IHC, we have shown that stress markers phopho-cJun and Chac1 in TB lesions are expressed by Ifnb- and iNOS-expressing macrophages (Fig.7E and Suppl.Fig.13D-F).

(9) We performed additional experiment to demonstrate that naïve (non-BCG vaccinated) lymphocytes did not improve Mtb control by Mtb-infected macrophages in agreement with previously published data (Suppl.Fig.7H).

Summary of updates

Following reviewers requests we updated figures to include isotype control antibodies, effects of inhibitors on non-stimulated cells, positive and negative controls for labile iron pool, additional images of 4-HNE and live/dead cell staining.

Isotype control for IFNAR1 blockade were included in Fig.3M, Fig.4C -E, Fig.6L-M Suppl.Fig.4F-G, 7I.

Positive and negative controls for labile iron pool measurements were added to Fig.3E, Fig.5D, Suppl.Fig.3B

Cell death staining images were added Suppl.Fig.3H

Co-staining of 4-HNE with tubulin was added to Suppl.Fig.3A.

High magnification images for Figure 7 were added in Suppl.Fig.8 to demonstrate paucibacillary and multibacillary image classification.

Single-channel color images for individual markers were provided in Fig.7E and Suppl.Fig.13B-F.

Inhibitor effects on non-stimulated cells were included in Fig.5 D-H, Suppl.Fig.6A and B. Titration of CSF1R inhibitors for non-toxic concentration determination are included in Suppl.Fig.6D.

In addition, we updated the figure legends in the revised manuscript to include more details about the experiments. We also clarified our conclusions in the Discussion. Responses to every major and minor comment of the reviewers are provided below.

**Point-by-point description of the revisions**

**Reviewer #1 (Evidence, reproducibility and clarity):**
SummaryThe study by Yabaji et al. examines macrophage phenotypes B6.Sst1S mice, a mouse strain with increased susceptibility to *M. tuberculosis* infection that develops necrotic lung lesions. Extending previous work, the authors specifically focus on delineating the molecular mechanisms driving aberrant oxidative stress in TNF-activated B6.Sst1S macrophages that has been associated with impaired control of *M. tuberculosis*. The authors use scRNAseq of bone marrow-derived macrophages to further characterize distinctions between B6.Sst1S and control macrophages and ascribe distinct trajectories upon TNF stimulation. Combined with results using inhibitory antibodies and small molecule inhibitors in in vitro experimentation, the authors propose that TNF-induced protracted c-Myc expression in B6.Sst1S macrophages disables the cellular defense against oxidative stress, which promotes intracellular accumulation of lipid peroxidation products, fueled at least in part by overexpression of type I IFNs by these cells. Using lung tissue sections from *M. tuberculosis*-infected B6.Sst1S mice, the authors suggest that the presence of a greater number of cells with lipid peroxidation products in lung lesions with high counts of stained *M. tuberculosis* are indicative of progressive loss of host control due to the TNF-induced dysregulation of macrophage responses to oxidative stress. In patients with active tuberculosis disease, the authors suggest that peripheral blood gene expression indicative of increased Myc activity was associated with treatment failure.Major commentsThe authors describe differences in protein expression, phosphorylation or binding when referring to Fig 2A-C, 2G, 3D, 5B, 5C. However, such differences are not easily apparent or very subtle and, in some cases, confounded by differences in resting cells (e.g. pASK1 Fig 3L; c-Myc Fig 5B) as well as analyses across separate gels/blots (e.g. Fig 3K, Fig 5B). Quantitative analyses across different independent experiments with adequate statistical analyses are required to strengthen the associated conclusions.

We updated our Western blots as follows:

(1) Densitometery of normalized bands is included above each lane (Fig.2A-C; Fig.3C-D and 3K; Fig.4A-B; Fig.5B,C,I,J). New data in Fig.3K is added to highlight differences between B6 and B6.Sst1S at individual timepoints after TNF stimulation. In Fig.5I we added new data comparing Myc levels in B6 and B6.Sst1S with and without JNK inhibitor and updated the results accordingly. New Fig.3K clearly demonstrates the persistent activation of p-cJun and pAsk1 at 24 and 36h of TNF stimulation. In Fig.5B we clearly demonstrate that Myc levels were higher in B6.Sst1S after 12 h of TNF stimulation. At 6h, however, the basal differences in Myc levels are consistently higher in B6.Sst1S and the induction by TNF is 1.6-fold similar in both backgrounds. We noted this in the text.

(2) A representative experiment is shown in individual panels and the corresponding figure legend contains information on number of biological repeats. Each Western blot was repeated 2 – 4 times.

The representative images of fluorescence microscopy in Fig 3H, 4H, 5H, S3C, S3I, S5A, S6A seem to suggest that under some conditions the fluorescence signal is located just around the nucleus rather than absent or diminished from the cytoplasm. It is unclear whether this reflects selective translocation of targets across the cell, morphological changes of macrophages in culture in response to the various treatments, or variations in focal point at which images were acquired. Control images (e.g. cellular actin, DIC) should be included for clarification. If cell morphology changes depending on treatments, how was this accounted for in the quantitative analyses? In addition, negative controls validating specificity of fluorescence signals would be warranted.

Our conclusion of higher LPO production is based on several parameters: 4-HNE staining, measurements of MDA in cell lysates and oxidized lipids using BODIPY C11. Taken together they demonstrate significant and reproducible increase in LPO accumulation in TNFstimulated B6.Sst1S macrophages. This excludes imaging artefact related to unequal 4-HNE distribution noted by the reviewer. In fact, we also noted that the 4-HNE was spread within cell body of B6.Sst1S macrophages and confirmed it using co-staining with tubulin, as suggested by the reviewer (new Suppl.Fig.3A). Since low molecular weight LPO products, such as MDA and 4-HNE, traverse cell membranes, it is unlikely that they will be strictly localized to a specific membrane bound compartment. However, we agree that at lower concentrations, there might be some restricted localization, explaining a visible perinuclear ring of 4-HNE staining in B6 macrophages. This phenomenon may be explained just by thicker cytoplasm surrounding nucleus in activated macrophages spread on adherent plastic surface or by proximity to specific organelles involved in generation or clearance of LPO products and definitively warrants further investigation.

We also included images of non-stimulated cells in Fig.3H, Suppl.Fig.3A and 3E. We used multiple fields for imaging and quantified fluorescence signals (Suppl. Fig.3D and 3F, Suppl.Fig.4G, Suppl.Fig.6A and B).

We used negative controls without primary antibodies for the initial staining optimization, but did not include it in every experiment.

To interpret the evaluation on the hierarchy of molecular mechanisms in B6.Sst1S macrophages, comparative analyses with B6 control cells should be included (e.g. Fig 4C-I, Fig 5, Fig 6B, E-M, S6C, S6E-F). This will provide weight to the conclusions that the dysregulated processes are specifically associated with the susceptibility of B6.Sst1S macrophages.

Understanding the *sst1*-mediated effects on macrophage activation is the focus of our previously published studies Bhattacharya et al., JCI, 2021 and this manuscript. The data comparing B6 and B6.Sst1S macrophage are presented in Fig.1, Fig.2, Fig.3, Fig.4, Fig.5A-C, I and J, Fig.6A-C, 6J and corresponding supplemental figures 1, 2, 3, 4A and B, Suppl.Fig.5, Suppl.Fig.6C, Suppl.Fig.7A-D,7F.

Once we identified the aberrantly activated pathways in the B6.Sst1S, we used specific inhibitors to correct the aberrant response in B6.Sst1S.

All experiments using inhibitory antibodies require comparison to the effect of a matched isotype control in the same experiment (e.g. Fig 3J, 4F, G, I; 6L, 6M, S3G, S6F).

Isotype control for IFNAR1 blockade were included in Fig.3M, Fig.4C-E, Fig.6L-M Suppl.Fig.4F-G, 7I.

Experiments using inhibitors require inclusion of an inhibitor-only control to assess inhibitor effects on unstimulated cells (e.g. Fig 4I, 5D-I)

Inhibitor effects on non-stimulated cells were included in Fig.5 D-H, Suppl.Fig.6A and B.

Fig 3K and Fig 5J appear to contain the same images for p-c-Jun and b-tubulin blots.

Fig.3K and 5J partially overlapped but had different focus – 3K has been updated to reflect the time course of stress kinase activation. Fig.5J is updated (currently Fig.5I and J) to display B6 and B6.Sst1S macrophage data including cMyc and p-cJun levels.

Data of TNF-treated cells in Fig 3I appear to be replotted in Fig 3J.

Currently these data is presented in Fig.3L and 3M and has been updated to include comparison of B6 and B6.Sst1S cells (Fig.3L) and effects of inhibitors in Fig.3M.

It is stated that lungs from 2 mice with paucibacillary and 2 mice with multi-bacillary lesions were analyses. There is contradicting information on whether these tissues were collected at the same time post infection (week 14?) or whether the pauci-bacillary lesions were in lungs collected at earlier time points post infection (see Fig S8A). If the former, how do the authors conclude that multi-bacillary lesions are a progression from paucibacillary lesions and indicative of loss of *M. tuberculosis* control, especially if only one lesion type is observed in an individual host? If the latter, comparison between lesions will likely be dominated by temporal differences in the immune response to infection.In either case, it is relevant to consider density, location, and cellular composition of lesions (see also comments on GeoMx spatial profiling). Is the macrophage number/density per tissue area comparable between pauci-bacillary and multi-bacillary lesions?

We did not collect lungs at the same time point. As described in greater detail in our preprints (Yabaji et al., https://doi.org/10.1101/2025.02.28.640830 and https://doi.org/10.1101/2023.10.17.562695) pulmonary TB lesions in our model of slow TB progression are heterogeneous between the animals at the same timepoint, as observed in human TB patients and other chronic TB animal models. Therefore, we perform analyses of individual TB lesions that are classified by a certified veterinary pathologist in a blinded manner based on their morphology (H&E) and acid fast staining of the bacteria, as depicted in Suppl.Fig.8. Currently it is impossible to monitor progression of individual lesions in mice. However, in mice TB is progressive disease and no healing and recovery from the disease have been observed in our studies or reported in literature. Therefore, we assumed that paucibacillary lesions preceded the multibacillary ones, and not vice versa, thus reflecting the disease progression. In our opinion, this conclusion most likely reflects the natural course of the disease. However, we edited the text : instead of disease progression we refer to paucibacillary and multibacillary lesions.

Does 4HNE staining align with macrophages and if so, is it elevated compared to control mice and driven by TNF in the susceptible vs more resistant mice?

We performed additional staining and analyses to demonstrate the 4-HNE accumulation in CD11b+ myeloid cells of macrophage morphology. Non-necrotic lesions contain negligible proportion of neutrophils (Fig.7B, Suppl.Fig.9B). B6 mice do not develop advanced multibacillary TB lesions containing 4-HNE+ cells. Also, 4-HNE staining was localized to TB lesions and was not found in uninvolved lung areas of the infected mice, as shown in Suppl.Fig.9A (left panel).

It is well established that TNF plays a central role in the formation and maintenance of TB granulomas in humans and in all animal models. Therefore, TNF neutralization would lead to rapid TB progression, rapid Mtb growth and lesions destruction in both B6 and B6.Sst1S genetic backgrounds.

Pathway analysis of spatial transcriptomic data (Suppl.Fig.11) identified TNF signaling via NFkB among dominant pathways upregulated in multibacillary lesions, suggesting that the 4-HNE accumulation paralleled increased TNF signaling. In addition, in vivo other cytokines, including IFN-I, could activate macrophages and stimulate production of reactive oxygen and nitrogen species and lead to the accumulation of LPO products as shown in this manuscript.

It would be relevant to state how many independent lesions per host were sampled in both the multiplex IHC as well as the GeoMx data. Can the authors show the selected regions of interest in the tissue overview and in the analyses to appreciate within-host and across-host heterogeneity of lesions. The nature of the spatial transcriptomics platform used is such that the data are derived from tissue areas that contain more than just Iba1+ macrophages. At later stages of infection, the cellular composition of such macrophage-rich areas will be different when compared to lesions earlier in the infection process. Hence, gene expression profiles and differences between tissue regions cannot be attributed to macrophages in this tissue region but are more likely a reflection of a mix of cellular composition and per-cell gene expression.

We used Iba1 staining to identify macrophages in TB lesions and programmed GeoMx instrument to collect spatial transcriptomics probes from Iba1+ cells within ROIs. Also, we selected regions of interest (ROI) avoiding necrotic areas (depicted in Suppl.Fig.10). We agree that Iba1+ macrophage population is heterogenous – some Iba1+ cells are activated iNOS+ macrophages, other are iNOS-negative (Fig.7C and D, and Suppl.Fig.13A). Multibacillary lesions contain larger areas occupied by activated (iNOS+) macrophages (Fig.7D),

(Suppl.Fig.13B and 13F). Although the GeoMx spatial transcriptomic platform does not provide single cell resolution, it allowed us to compare populations of Iba1+ cells in paucibacillary and multibacillary TB lesions and to identify a shift in their overall activation pattern.

It is stated that loss of control of *M. tuberculosis* in multibacillary lesions was associated with "downregulation of IFNg-inducible genes". If the authors base this on the tissue expression of individual genes, this requires further investigation to support such conclusion (also see comment on GeoMx above). Furthermore, how might this conclusion be compatible with significantly elevated iNOS+ cells (Fig 7D) in multibacillary lesions?

We demonstrated that *Ciita* gene expression is specifically induced by IFN-gamma and is suppressed by IFN-I (Fig.6M). The expression of Ciita in paucibacillary lesions suggest the presence of the IFN-gamma activated cells and its disappearance in the multibacillary lesion is consistent with massive activation of IFN-I pathway (Fig.7C).

It is appreciated that the human blood signature analyses contain Myc-signatures but the association with treatment failure is not very strong based on the data in Fig 13B and C (Suppl.Fig.15B and C now). The authors indicate that they have no information on disease severity, but it should perhaps not be assumed that treatment failure is indicative of poor host control of the infection. Perhaps independent analyses in separate cohort/data set can add strength and provide -additional insights (e.g. PMID: 35841871; PMID: 32451443, PMID: 17205474, PMID: 22872737). In addition, the human data analyses could be strengthened by extension to additional signatures such as IFN, TNF, oxidative stress. Details of the human study design are not very clear and are lacking patient demographics, site of disease, time of blood collection relative to treatment onset, approving ethics committees.

X axis of Suppl.Fig.15A represent pre-defined molecular signature gene sets (MSigDB) in Gene Set Enrichment Analysis (GSEA) database (https://www.gseamsigdb.org/gsea/msigdb). On Y axis is area under curve (AUC) score for each gene set. The Myc upregulated gene set myc_up was identified among top gene sets associated with treatment failure using unbiased ssGSEA algorithm. The upregulation of Myc pathway in the blood transcriptome associated with TB treatment failure most likely reflects greater proportion of immature cells in peripheral blood, possibly due to increased myelopoiesis.

Pathway analysis of the differentially expressed genes revealed that treatment failures were associated with the following pathways relevant to this study: NF-kB Signaling, Flt3 Signaling in Hematopoietic Progenitor Cells (indicative of common myeloid progenitor cell proliferation), SAPK/JNK Signaling and Senescence (indicative of oxidative stress). The upregulation of these pathways in human patients with poor TB treatment outcomes correlates with our findings in TB susceptible mice. The detailed analysis of differentially regulated pathways in human TB patients is beyond the scope of this study and is presented in another manuscript entitled “ Tuberculosis risk signatures and differential gene expression predict individuals who fail treatment” by Arthur VanValkenburg et al., submitted for publication.

Blood collection for PBMC gene expression profiling of TB patients was prior to TB treatment or within a first week of treatment commencement. Boxplot of bootstrapped ssGSEA enrichment AUC scores from several oncogene signatures ranked from lowest to highest AUC score, with myc_up and myc_dn genes highlighted in red.

We agree with the reviewer that not every gene in the myc_up gene set correlates with the treatment outcome. But the association of the gene set is statistically significant, as presented in Suppl.Fig.15B – C.

We updated the details of the study, including study sites and the ethics committee approval statement and references describing these cohorts.

Other commentsIt is excellent that the authors provide individual data points. Choosing a colour other than black would increase clarity when black bars are used.

We followed this useful suggestion and selected consistent color codes for B6 and B6.Sst1S groups to enhance clarity throughout the revised manuscript.

Error bars are inconsistently depicted as either bi-directional or just unidirectional.

We used bi-directional error bars in the revised manuscript.

Fig 1E, G, H - please include a scale to clarify what the heat map is representing.

We have included the expression key in Fig.1E,G and H and Suppl.Fig.1C and D in the revised version.

Fig 2K, Fig S10A gene information cannot be deciphered.

We increased the font in previous Fig.2K and moved to supplement to keep larger fonts (current Suppl.Fig.2G).

Fig S4A,B please add error bars.

These data are presented as Suppl.Fig.5 in the revised version. We performed one experiment to test the hypothesis. Because the data indicated no clear increase in transposon small RNAs in the sst1S macrophages, we did not pursue this hypothesis further, and therefore, the error bars were not included. However, we decided to include these negative data because it rejects a very attractive and plausible hypothesis.

Please use gene names as per convention (e.g. Ifnb1) to distinguish gene expression from protein expression in figures and text.

We addressed the comment in the revised manuscript.

Fig S8B. Contrary to the description of results, there seems to be minimal overlap between the signal for YFP and the Ifnb1 probe. Is the Ifnb1 reporter mouse a legacy reporter? If so, it is worth stating this and including such considerations in the data interpretation.

The YFP reporter expresses YFP protein under the control of the *Ifnb1* promoter. The YFP protein accumulates within the cells and while Ifnb protein is rapidly secreted and does not accumulate in the producing cells in appreciable amounts. So YFP is not a lineage tracing reporter, but its accumulation marks the *Ifnb1* promoter activity in cells, although the YFP protein half-life is longer than that of the *Ifnb1* mRNA that is rapidly degraded (Witt et al., BioRxiv, 2024; doi:10.1101/2024.08.28.61018). Therefore, there is no precise spatiotemporal coincidence of these readouts.

Please clarify what is meant by "normal interstitium" ? If the tissue is from uninfected mice, please state clearly.

In this context we refer to the uninvolved lung areas of the infected lungs. In every sample we compare uninvolved lung areas and TB lesions of the same animal. Also, we performed staining of lung of non-infected mice as additional controls.

If macrophage cultures underwent media changes every 48h, how was loss of liberated Mtb taken into account especially if differences in cell density/survival were noted? The assessment of *M. tuberculosis* load by qPCR is not well described. In particular, the method of normalization applied within the experiments (not within the qPCR) here remains unclear, even with reference to the authors' prior publication.

Our lab has many years of experience working with macrophage monolayers infected with virulent Mtb and uses optimized protocols to avoid cell losses and related artifacts. Recently we published a detailed protocol for this methodology in STAR Protocols (Yabaji et al., 2022; PMID 35310069). In brief, it includes preparation of single cell suspensions of Mtb by filtration to remove clumps, use of low multiplicity of infection, preparation of healthy confluent monolayers and use of nutrient rich culture medium and medium change every 2 days. We also rigorously control for cell loss using whole well imaging and quantification of cell numbers and live/dead staining.

Please add citation for the limma package.

The references has been added (Ritchie et al, NAR 2015; PMID 25605792).

The description of methodology relating to the "oncogene signatures" is unclear.

This signature was described in Bild etal, Nature, 2006 and McQuerry JA, et al, 2019 “Pathway activity profiling of growth factor receptor network and stemness pathways differentiates metaplastic breast cancer histological subtypes”. BMC Cancer 19: 881 and is cited in Methods section Oncogene signatures

Please clearly state time points post infection for mouse analyses.

We collected lung samples from Mtb infected mice 12 – 20 weeks post infection. The lesions were heterogeneous and were individually classified using criteria described above.

Reference is made to "a list of genes unique to type I [interferon] genes [....]" (p29). Can the authors indicate the source of the information used for compiling this list?

The lists were compiled from Reactome, EMBL's European Bioinformatics Institute and GSEA databases. The links for all datasets are provided in Suppl.Table 8 “Expression of IFN pathway genes in Iba1+ cells from pauci- and multi-bacillary lesions of Mtb infected B6.Sst1S mouse lungs” in the “Pool IFN I & II gene sets” worksheet.

The discussion at present is very long, contains repetition of results and meanders on occasion.

Thank you for this suggestion, We critically revised the text for brevity and clarity.

**Reviewer #1 (Significance):**
Strengths and limitationsStrengths: multi-pronged analysis approaches for delineating molecular mechanisms of macrophage responses that might underpin susceptibility to *M. tuberculosis* infection; integration of mouse tissues and human blood samplesWeaknesses: not all conclusions supported by data presented; some concerns related to experimental design and controls; links between findings in human cohort and the mechanistic insights gained in mouse macrophage model uncertain

The revised manuscript addresses every major and minor comment of the reviewers, including isotype controls and naïve T cells, to provide additional support for our conclusions. Our study revealed causal links between Myc hyperactivity with the deficiency of anti-oxidant defense and type I interferon pathway hyperactivity. We have shown that Myc hyperactivity in TNF-stimulated macrophages compromises antioxidant defense leading to autocatalytic lipid peroxidation and interferon-beta superinduction that in turn amplifies lipid peroxidation, thus, forming a vicious cycle of destructive chronic inflammation. This mechanism offers a plausible mechanistic explanation of for the association of Myc hyperactivity with poorer treatment outcomes in TB patients and provide a novel target for host-directed TB therapy.

AdvanceThe study has the potential to advance molecular understanding of the TNF-driven state of oxidative stress previously observed in B6.Sst1S macrophages and possible implications for host control of *M. tuberculosis* in vivo.AudienceExperts seeking understanding of host factors mediating *M. tuberculosis* control, or failure thereof, with appreciation for the utility of the featured mouse model in assessing TB diseases progression and severe manifestation. Interest is likely extended to audience more broadly interested in TNF-driven macrophage (dys)function in infectious, inflammatory, and autoimmune pathologies.Reviewer expertiseIn preparing this review, I am drawing on my expertise in assessing macrophage responses and host defense mechanisms in bacterial infections (incl. virulent *M. tuberculosis*) through in vitro and in vivo studies. This includes but is not limited to macrophage infection and stimulation assays, microscopy, intra-macrophage replication of *M. tuberculosis*, analyses of lung tissues using multi-plex IHC and spatial transcriptomics (e.g. GeoMx). I am familiar with the interpretation of RNAseq analyses in human and mouse cells/tissues, but can provide only limited assessment of appropriateness of algorithms and analysis frameworks.
**Reviewer #2 (Evidence, reproducibility and clarity):**
Yabaji et al. investigated the effects of BMDMs stimulated with TNF from both WT and B6.Sst1S mice, which have previously been identified to contain the sst1 locus conferring susceptibility to *Mycobacterium tuberculosis*. They identified that B6.Sst1S macrophages show a superinduction of IFNß, which might be caused by increased c-Myc expression, expanding on the mechanistic insights made by the same group (Bhattacharya et al. 2021). Furthermore, prolonged TNF stimulation led to oxidative stress, which WT BMDMs could compensate for by the activation of the antioxidant defense via NRF2. On the other hand, B6.Sst1S BMDMs lack the expression of SP110 and SP140, co-activators of NRF2, and were therefore subjected to maintained oxidative stress. Yabaji et al. could link those findings to in vivo studies by correlating the presence of stressed and aberrantly activated macrophages within granulomas to the failure of Mtb control, as well as the progression towards necrosis. As the knowledge regarding Mtb progression and necrosis of granulomas is not yet well understood, findings that might help provide novel therapy options for TB are crucial. Overall, the manuscript has interesting findings with regard to macrophage responses in *Mycobacteria tuberculosis* infection.However, in its current form there are several shortcomings, both with respect to the precision of the experiments and conclusions drawn.In particular (a) important controls are often missing, e.g. T-cells form non-immune mice in Fig. 6J, in F, effectivity of BCG in B6 mice in 6N; (b) single experiments are shown throughout the manuscript, in particular western blots and histology without proper quantification and statistics, this is absolutely not acceptable; (c) very few repetitions are shown in in vitro experiments, where there is no evidence for limitation in resources (usually not more than 3), it is not clear what "independent experiment means" - i.e. the robustness of the findings is questionable; (d) data are often normalized multiple times, e.g. in the case of qPCR, and the methods of normalization are not clear (what house-keeping gene exactly?);Moreover, experiments regarding IFN I signaling (e.g. short term TNF treatment of BMDMs to analyze LPO, making sure that the reporter mouse for IFNß works in vivo) and c-Myc (e.g. the increase after M-CSF addition might impact on other analysis as well and the experiments should be adjusted to control for this effect; MYC expression in the human samples) should be carefully repeated and evaluated to draw correct conclusions.In addition, we would like to strongly encourage the authors to more precisely outline the experimental set-ups and figure legends, so that the reader can easily understand and follow them. In other words: The legends are - in part very - incomplete. In addition, the authors should be mindful of gene names vs. protein names and italicize where appropriate.

We appreciate a very thorough evaluation of our manuscript by this reviewer. Their insightful comments helped us improve the manuscript. As outlined below in point-by-point responses (1) we added important controls including isotype control antibodies in IFNAR blocking experiments and non-vaccinated T cells in T cell – macrophage interactions experiments; updated figure legends to indicate number of repeated experiment where a representative experiment is shown, numbers of mouse lungs and individual lesions, methods of data normalization, where it was missing. We also explained our in vitro experimental design and how we analyzed and excluded effects of media change and fresh CSF1 addition, by using a rest period before TNF stimulation and Mtb infection. The data shown in Suppl. Fig. 6C (previously Suppl. Fig. 5B) demonstrate that Myc levels induced by CSF1 return to the basal level at 12 h after media change. Our detailed in vitro protocol that contains these details has been published (Yabaji et al., STAR Protocols, 2022). We added new data demonstrating the ROS and LPO production at 6h of TNF stimulation, while the *Ifnb1* mRNA super-induction occurred at 16 – 18 h, and edited the text to highlight these dynamics. The upregulation of Myc pathway in human samples does not necessarily mean the upregulation of Myc itself, it could be due to the dysregulation of downstream pathways. The upregulation of Myc pathway in the blood transcriptome associated with TB treatment failure most likely reflects greater proportion of immature cells in peripheral blood, possibly due to increased myelopoiesis. The detailed analysis of this cell populations in human patients is suggested by our findings but it is beyond the scope of this study.

The reviewer’s comments also suggested that a summary of our findings was necessary. The main focus of our study was to untangle connections between oxidative stress and *Ifnb1* superinduction. It revealed that Myc hyperactivity caused partial deficiency of antioxidant defense leading to type I interferon pathway hyperactivity that in turn amplifies lipid peroxidation, thus establishing a vicious cycle driving inflammatory tissue damage.

Our laboratory worked on mechanisms of TB granuloma necrosis over more than two decades using genetic, molecular and immunological analyses in vitro and in vivo. It provided mechanistic basis for independent studies in other laboratories using our mouse model and further expanding our findings, thus supporting the reproducibility and robustness of our results and our lab’s expertise.

Specific comments to the experiments and data:- Fig. 1E: Evaluation of differences in up- and downregulation between B6 and B6.Sst1S cells should highlight where these cells are within the heatmap, as it is only labelled with the clusters, or it should be depicted differently (in particular for cluster 1 and 2). Furthermore, a more simple labelling of the pathways would increase the readability of the data.

For our scRNAseq data presentation, we used formats accepted by computational community. To clarify Fig.1E, we added labels above B6 and B6.Sst1S-specific clusters.

- Fig. 2D, E: The staining legend is missing. For the quantification it is not clear what % total means. Is this based on the intensity or area? What do the dots represent in the bar chart? Is one data point pooled from several pictures? If not, the experiments need to be repeated, as three pictures might not be representative for evaluation.- Fig. 2E: Statistics comparing B6/ B6,SsT1S with TNF (different) is required: Absence of induction is not a proof for a difference!

We included staining with NRF2-specific antibodies and performed area quantification per field using ImageJ to calculate the NRF2 total signal intensity per field. Each dot in the graph represents the average intensity of 3 fields in a representative experiment. The experiment was repeated 3 times. We included pairwise comparison of TNF-stimulated B6 and B6.Sst1S macrophages and updated the figure legend.

- Fig. 3E: Positive and negative control need to be depicted in the figure (see legend).

We have added the positive and negative controls for the determination of labile iron pool to the data in Fig. 3E and related Suppl. Fig. 3B and to Fig. 5D that also demonstrates labile iron determination.

- Fig. 3I: A quantification by flow cytometry or total cell counts are important, as 6% cell death in cell culture is a very modest observation. Otherwise, confocal images of the quantification would be a good addition to judge the specificity of the viability staining.

To validate the specificity of the viability staining method, we have provided fluorescent images as Suppl.Fig.3H. The main point of this experiment was to demonstrate a modest, but reproducible, increase in cell death in the *sst1*-mutant macrophages that suggested an IFNdependent oxidative damage. In our study, we did not focus on mechanisms of cell death, but on a state of chronic oxidative stress in the *sst1* mutant live cells during TNF stimulation.

- Fig. 3I, J: What does one dot represent?

We performed this assay in 96 well format and each dot represent the % cell death in an individual well.

- Fig. 3K,L: For the B6 BMDMs it seems that p-cJun is highly increased at 12h in (L), while it is not in (K). On the other hand, for the B6.Sst1S BMDMs it peaks at 24h in (K), while in (L) it seems to at 12h. According to the data in (L) it seems that p-cJun is rather earlier and stronger activated in B6 BMDMs and has a weakened but prolonged activation in the B6.Sst1S BMDMs, which would not fit with your statement in the text that B6.Sst1S BMDMs show an upregulation.These experiments need repetitions and quantification and statistiscs.Fig. 3L: ASK1 seems to be higher at 12h for the B6 BMDMs and similar for both lines at 24h, which is not fitting to the statement in the text. ("Also, the ASK1 - JNK - cJun stress kinase axis was upregulated in B6.Sst1S macrophages, as compared to B6, after 12 - 36 h of TNF stimulation")

These experiments were repeated, and new data were added to highlight differences in ASK1 and c-Jun phosphorylation between B6 and B6.Sst1S at individual timepoints after TNF stimulation (presented in new Fig.3K). It demonstrated that after TNF stimulation the activation of stress kinases ASK1 and c-Jun initially increased in both genetic backgrounds. However, their upregulation was maintained exclusively in the *sst1*-susceptible macrophages from 24 to 36 h of TNF stimulation, while in the resistant macrophages their upregulation was transient. Thus, during prolonged TNF stimulation, B6.Sst1S macrophages experience stress that cannot be resolved, as evidenced by this kinetic analysis. The quantification of the band intensity was added to Western blot images above individual lanes.

Reviewer 2 pointed to missing isotype control antibodies in Fig.3 and Fig.4:- Figure 3J: the isotype control for the IFNAR antibody is missing- Figure 4E: It seems the isotype control itself has already an effect in the reduction of IFNb.- Fig. 4H: It seems that the Isotype control antibody had an effect to increase 4-HNE (compared to TNF stimulated only).

We always include isotype control antibodies in our experiments because antibodies are known to modulate macrophage activation via binding to Fc receptor. To address the reviewer’s comments, we updated all panels that present the effects of IFNAR1 blockade with isotypematched non-specific control antibodies in the revised manuscript. Specifically, we included isotype control in Fig. 3M (previously Fig.3J), (Fig.4I, Suppl.4E-G, Fig.6L-M), Suppl.Fig.7I (previously Suppl.Fig.6F).

- Fig.4A - C: "IFNAR1 blockade, however, did not increase either the NRF2 and FTL protein levels, or the Fth, Ftl and Gpx1 mRNA levels above those treated with isotype control antibodies"

Maybe not above the isotype but it is higher than the TNF alone stimulation at least for NRF2 at 8h and for Ftl at both time points. Why does the isotype already cause stimulation/induction of the cells? !These experiments need repetitions and quantification and statistics!

To determine specific effects of IFNAR blockade we compared effects of non-specific isotype control and IFNAR1-specific antibodies. In our experiments, the isotype control antibody modestly increased of Nrf2 and Ftl protein levels and the *Fth* and *Ftl* mRNA levels, but their effects were similar to the effect of IFNAR-specific antibody. The non-IFN -specific effects of antibodies, although are of potential biological significance, are modest in our model and their analysis is beyond the scope of this study.

- Fig.4H Was the AB added also at 12h post stimulation? Figure legend should be adjusted.

The IFNAR1 blocking antibodies and isotype control antibodies were added at 2 h after TNF stimulation in Fig.4H and 4I, as described in the corresponding figure legend. The data demonstrating effects of IFNAR blockade after 12, 24,and 33h of TNF stimulation are presented in Suppl.Fig.4 E-G.

- Figure 4I: How was the data measured here, i.e. what is depicted? The isotype control is missing. It seems a two-way ANOVA was used, yet it is stated differently. The figure legend should be revised, as Dunnett's multiple comparison would only check for significances compared to the control.

The microscopy images and bar graphs were updated to include isotype control and presented in Suppl. Fig.4E - G of the revised version. We also revised the statistical analysis to include correction for multiple comparisons.

- Figure 4C and subsequent: How exactly was the experiment done (house-keeping gene)?

We included the details in the figure legends of revised version. We quantified the gene expression by DDCt method using b-actin (for Fig. 4C-E) and 18S (For Fig. 4F and G) as internal controls.

- Figure 4D,E: Information on cells used is missing. Why the change in stimulation time? Did it not work after 12h? Then the experiments in A-C should be repeated for 16h.

The updated Fig. 4D and E present comparison of B6 and B6.Sst1S BMDMs clearly demonstrating significant difference between these macrophages in *Ifnb1* mRNA expression 16 h after TNF stimulation, in agreement with our previous publication(Bhattacharya, et al., 2021). There we studied the time course of responses of B6 and B6.Sst1S macrophages to TNF at 2h intervals and demonstrated the divergence between their activation trajectories starting at 12 h of TNF stimulation Therefore, to reveal the underlying mechanisms we focus our analyses on this critical timepoint, i.e. as close to the divergence as possible. However, the difference between the strains in *Ifnb1* mRNA expression achieved significance only by 16h of TNF stimulation. That is why we have used this timepoint for the *Ifnb1* and *Rsad2* analyses. It clearly shows that the superinduction was not driven by the positive feedback via IFNAR, as has been shown by the Ivashkiv lab for B6 wild type macrophages previously PMID 21220349.

- Figure 4E: It would be helpful to see if these transcripts are actually translated into protein levels, e.g. perform an ELISA. Authors state that IFNAR blockages does not alter the expression but you statistic says otherwise.- The data for Ifnb expression (or better protein level) should be provided for B6 BMDMs as well.

We have previously reported the differences in Ifnb protein secretion (He et al., Plos Pathogens, 2013 and Bhattacharya et al., JCI 2021). We use mRNA quantification by qRT-PCR as a more sensitive and direct measurement of the *sst1*-mediated phenotype. The revised Fig.4D and E include responses of B6 in addition to the B6.Sst1S to demonstrate that the IFNAR blockade does not reduce the *Ifnb1* mRNA levels in TNF-stimulated B6.Sst1S mutant to the B6 wild type levels. A slight reduction can be explained by a known positive feedback loop in the IFN-I pathway (see above). In this experiment we emphasized that the effect of the *sst1* locus is substantially greater, as compared to the effect of the IFNAR blockade (Fig.4D), and updated the text accordingly.

- Fig. 4F: To what does the fold induction refer to? If it is again to unstimulated cells, then why is the induction now so much higher than in (E) where it was only 50x (now to 100x).- Figure 4G: Again to what is the fold induction referring to? It seems your Fer-1 treatment only contains 2 data points. This needs to be fixed.

Yes, the fold induction was calculated by normalizing mRNA levels to untreated control incubated for the same time. Regarding the variation in *Ifnb1* mRNA levels - a two-fold variation is not unusual in these experiments that may result in the *Ifnb1* mRNA superinduction ranging from 50 -200-fold at this timepoint (16h). The graph in Fig.4G was modified to make all datapoints more visible.

- "These data suggest that type I IFN signaling does not initiate LPO in our model but maintains and amplifies it during prolonged TNF stimulation that, eventually, may lead to cell death". Data for a short term TNF stimulation are not shown, however, so it might impact also on the initiation of LPO.- The overall conclusion drawn from Fig. 3 and 4 is not really clear with regard that IFN does not initiate LPO. Where is that shown? Data on earlier stimulation time points should be added to make this clear.

We demonstrated ROS production (new Suppl.Fig.3G) and the rate of LPO biosynthesis (new Suppl.Fig.4E-F) at 6 h post TNF stimulation, while the *Ifnb1* superinduction occurs between 12-18 h post TNF stimulation. This temporal separation supports our conclusion that IFN-β superinduction does not initiate LPO. We clarified it in the text:

“Thus, *Ifnb1* super-induction and IFN-I pathway hyperactivity in B6.Sst1S macrophages follow the initial LPO production, and maintain and amplify it during prolonged TNF stimulation”. (Previously: These data suggest that type I IFN signaling does not initiate LPO in our model). We also edited the conclusion in this section to explain the hierarchy of the *sst1*-regulated AOD and IFN-I pathways better:

“Taken together, the above experiments allowed us to reject the hypothesis that IFN-I hyperactivity caused the *sst1*-dependent AOD dysregulation. In contrast, they established that the hyperactivity of the IFN-I pathway in TNF-stimulated B6.Sst1S macrophages was itself driven by the initial dysregulation of AOD and iron-mediated lipid peroxidation. During prolonged TNF stimulation, however, the IFN-I pathway was upregulated, possibly via ROS/LPOdependent JNK activation, and acted as a potent amplifier of lipid peroxidation”.

We believe that these additional data and explanation strengthen our conclusions drawn from Figures 3 and 4.

- "A select set of mouse LTR-containing endogenous retroviruses (ERV's) (Jayewickreme et al, 2021), and non-retroviral LINE L1 elements were expressed at a basal level before and after TNF stimulation, but their levels in the B6.Sst1S BMDMs were similar to or lower than those seen in B6". This sentence should be revised as the differences between B6 and B6.Sst1S BMDMs seem small and are not there after 48h anymore. Are these mild changes really caused by the mutation or could they result from different housing conditions and/or slowly diverging genetically lines. How many mice were used for the analysis? Is there already heterogeneity between mice from the same line?

We agree with the reviewer that the data presented in Suppl.Fig.4 (Suppl.Fig.5 in the revised version) indicated no increase in single- and double-stranded transposon RNAs in the B6.Sst1S macrophages. The purpose of these experiment was to test the hypothesis that increased transposon expression might be responsible for triggering the superinduction of type I interferon response in TNF-stimulated B6.Sst1S macrophages. In collaboration with a transposon expert Dr. Nelson Lau (co-author of this manuscript) we demonstrated that transposon expression was not increased above the B6 level and, thus, rejected this attractive hypothesis. We explained the purpose of this experiment in the text and adequately described our findings as “the levels in the B6.Sst1S BMDMs were similar to or lower than those seen in B6”…and concluded that ” the above analyses allowed us to exclude the overexpression of persistent viral or transposon RNAs as a primary mechanism of the IFN-I pathway hyperactivity” in the *sst1*-mutant macrophages.

- Fig. 5A: Indeed, it even seems that Myc is upregulated for the mutant BMDMs. Yet, there are only 2 data points for B6 12h.These experiments need repetitions and quantification and statistics.

We observed these differences in c-Myc mRNA levels by independent methods: RNAseq and qRT-PCR. The qRT-PCR experiments were repeated 3 times. A representative experiment in Fig.5A shows 3 data points for each condition. We reformatted the panel to make all data points clearly visible.

- Fig. 5B: Why would the protein level decrease in the controls over 6h of additional cultivation? Is this caused by fresh M-CSF? In this case maybe cells should be left to settle for one day before stimulating them to properly compare c-Myc induction. Comment on two c-Myc bands is needed. At 12h only the upper one seems increased for TNF stimulated mutant BMDMs compared to B6 BMDMs.

We agree with the reviewer’s point that cells need to be rested after media change that contains fresh CSF-1. Indeed, in Suppl.Fig.6C, we show that after media change containing 10% L929 supernatant (a source of CSF1) there is an increase in c-Myc protein levels that takes approximately 12 hours to return to baseline.

Our protocol includes resting period of 18-24 h after medium change before TNF stimulation.

We updated Methods to highlight this detail. Thus, the increase in c-Myc levels we observe at 12 h of TNF stimulation (Fig.5B) is induced by TNF, not the addition of growth factors, as further discussed in the text.

The two c-Myc bands observed in Fig.5B,I and J, are similar to patterns reported in previous studies that used the same commercial antibodies (PMIDs: 24395249, 24137534, 25351955). Whether they correspond to different c-Myc isoforms or post-translational modifications is unknown.

- Fig. 5A,B: It seems that not all the RNA is translated into protein, as c-Myc at 12h in the mutant BMDMs seems to be lower than at 6h, while the gene expression implicates it vice versa.

In addition to Fig.5B, the time course of Myc protein expression up to 24 h is presented in new panels Fig. 5I-5J. It demonstrates the gradual decrease of Myc protein levels. The observed dissociation between the mRNA and protein levels in the *sst1*-mutant BMDMs at 12 and 24 h is most likely due to translation inhibition as a result of the development of the integrated stress response, ISR (as shown in our previous publication by Bhattacharya et al., JCI, 2021). Translation of Myc is known to be particularly sensitive to the ISR (PMID18551192, PMID25079319, PMID28490664). Perhaps, the IFN-driven ISR may serve as a backup mechanism for Myc downregulation. We are planning to investigate these regulatory mechanisms in greater detail in the future.

- Fig. 5J: Indeed, the inhibitor seems to cause the downregulation of the proteins. Explanation?

This experiment was repeated twice and the average normalized densitometry values are presented in the updated Fig.5J. The main question addressed in this experiment was whether hyperactivity of JNK in TNF-stimulated *sst1* mutant macrophages contributed to Myc upregulation, as had been previously shown in cancer. Comparing effects of JNK inhibition on phospho-cJun and c-Myc protein levels in TNF stimulated B6.Sst1S macrophages (updated Fig.5J), we rejected the hypotghesis that JNK activity might have a major role in c-Myc upregulation in *sst1* mutant macrophages.

- "TNF stimulation tended to reduce the LPO accumulation in the B6 macrophages and to increase it in the B6.Sst1S ones" However, this is not apparent in Sup. Fig. 6B. Here it seems that there might be a significant increase.

Suppl.Fig.6B (currently Suppl.Fig.7B) shows the 4-HNE accumulation at day 3 post infection. The data obtained after 5 days of Mtb infection are shown in Fig.6A. We clarified this in the text: “By day 5 post infection, TNF stimulation induced significant LPO accumulation only in the B6.Sst1S macrophages (Fig.6A)”.

- Fig. 6B: Mtb and 4-HNE should be shown in two different channels in order to really assign each staining correctly.What time point is this? Are the mycobacteria cleared at MOI1, since it looks that there are fewer than that? How does this look like for the B6 BMDMs? Are there even less mycobacteria?

We included B6 infection data to the updated Fig.6B and added Suppl.Fig.7C and 7D that address this reviewer’s comment. The data represent day 5 of Mtb infection as indicated in the updated Fig.6B and Suppl.Fig.7C and 7D legends. New Suppl.Fig.7D shows quantification of replicating Mtb using Mtb replication reporter stain expressing single strand DNA binding protein GFP fusion, as described in Methods. We observed fewer Mtb and a lower percentage of replicating Mtb in B6 macrophages, but we did not observe a complete Mtb elimination in either background.

We used red fluorescence for both Mtb::mCherry and 4-HNE staining to clearly visualize the SSB-GFP puncta in replicating Mtb DNA. In the revised manuscript, we have included the relevant channels in Suppl. Fig.7C and D to demonstrate clearly distinct patterns of Mtb::mCherry and 4-HNE signals. We did not aim to quantify the 4-HNE signal intensity in this experiment. For the 4-HNE quantification we use Mtb that expressed no reporter proteins (Fig.6A-B and Suppl.Fig.7A-B).

- Fig 6E: In the context of survival a viability staining needs to be included, as well as the data from day 0. Then it needs to be analyzed whether cell numbers remain the same from D0 or if there is a change.

We updated Fig.6 legend to indicate that the cell number percentages were calculated based on the number of cells at Day 0 (immediately after Mtb infection). We routinely use fixable cell death staining to enumerate cell death to exclude artifacts due to cell loss. Brief protocol containing this information is included in Methods section. The detailed protocol including normalization using BCG spike has been published – Yabaji et al, STAR Protocols, 2022. Here we did not present dead cell percentage as it remained low and we did not observe damage to macrophage monolayers. The fold change of Mtb was calculated after normalization using Mtb load at Day 0 after infection and washes.

"The 3D imaging demonstrated that YFP-positive cells were restricted to the lesions, but did not strictly co-localize with intracellular Mtb, i.e. the Ifnb promoter activity was triggered by inflammatory stimuli, but not by the direct recognition of intracellular bacteria. We validated the IFNb reporter findings using in situ hybridization with the Ifnb probe, as well as anti-GFP antibody staining (Suppl.Fig.8B - E)." The colocalization is not present within the tissue sections. It seems that the reporter line does not show the same staining pattern in vivo as the IFNß probe or the anti GFP antibody staining. The reporter line has to be tested for the specificity of the staining. Furthermore, to state that it was restricted to the lesions, an uninvolved tissue area needs to be depicted.

The Ifnb secreting cells are notoriously difficult to detect in vivo using direct staining of the protein. Therefore, lineage tracing of reporter expression are used as surrogates. The Ifnb reporter used in our study has been developed by the Locksley laboratory (Scheu et al., PNAS, 2008, PMID: 19088190) and has been validated in many independent studies. The reporter mice express the YFP protein under the control of the *Ifnb1* promoter. The YFP protein accumulates within the cells, while Ifnb protein is rapidly secreted and does not accumulate in the producing cells in appreciable amounts. Also, the kinetics of YFP protein degradation is much slower as compared to the endogenous *Ifnb1* mRNA that was detected using in situ hybridization. Thus, there is no precise spatiotemporal coincidence of these readouts in Ifnb expressing cells in vivo. However, this methodology more closely reflect the Ifnb expressing cells in vivo, as compared to a Cre-lox mediated lineage tracing approach. In the revised manuscript we demonstrate that both YFP and mRNA signals partially overlap (Suppl.Fig.12B). In Suppl.Fig.12B. we also included a new panel showing no YFP expression in the uninvolved area of the reporter mice infected with Mtb. The YFP expression by activated macrophages is demonstrated by co-staining with Iba1- and iNOS-specific antibodies (new Fig.7D and Suppl.Fig.13A). Our specificity control also included TB lesions in mice that do not carry the YFP reporter and did not express the YFP signal, as reported elsewhere (Yabaji et al., BioRxiv, https://doi.org/10.1101/2023.10.17.562695).

- Are paucibacillary and multibacillary lesions different within the same animal or does one animal have one lesion phenotype? If that is the case, what is causing the differences between mice? Bacterial counts for the mice are required.

The heterogeneity of pulmonary TB lesions has been widely acknowledged in clinic and highlighted in recent experimental studies. In our model of chronic pulmonary TB (described in detail in Yabaji et al., https://doi.org/10.1101/2025.02.28.640830 and https://doi.org/10.1101/2023.10.17.562695) the development of pulmonary TB lesions is not synchronized, i.e. the lesions are heterogeneous between the animals and within individual animals at the same timepoint. Therefore, we performed a lesion stratification where individual lesions were classified by a certified veterinary pathologist in a blinded manner based on their morphology (H&E) and acid fast staining of the bacteria, as depicted in Suppl.Fig.8.

- "Among the IFN-inducible genes upregulated in paucibacillary lesions were Ifi44l, a recently described negative regulator of IFN-I that enhances control of Mtb in human macrophages (DeDiego et al, 2019; Jiang et al, 2021) and Ciita, a regulator of MHC class II inducible by IFNy, but not IFN-I (Suppl.Table 8 and Suppl.Fig.10 D-E)." Why is Sup. Fig. 10 D, E referred to? The figure legend is also not clear, e.g. what means "upregulated in a subset of IFN-inducible genes"? Input for the hallmarks needs to be defined.

These data is now presented in Suppl.Fig.11 and following the reviewer’s comment, we moved reference to panels 11D – E up to previous paragraph in the main text, where it naturally belongs . We also edited the figure legend to refer to the list of IFN-inducible genes compiled from the literature that is discussed in the text. We appreciate the reviewer’s suggestion that helped us improve the text clarity. The inputs for the Hallmark pathway analysis are presented in Suppl.Tables 7 and 8, as described in the text.

- Fig. 7C: Single channel pictures are required as it is hard to see the differences in staining with so many markers. Why is there no iNOS expression in the bottom row? What does the rectangle indicate on the bottom right? As black is chosen for DAPI, it is not visible at all. In case the signal is needed a visible a color should be chosen.

We thoroughly revised this figure to address the reviewer’s concern about the lack of clarity. We provide individual channels for each marker in Fig.7D – E and Suppl.Fig.13F. We have to use DAPI in these presentation in gray scale to better visualize other markers.

- "In the advanced lesions these markers were primarily expressed by activated macrophages (Iba1+) expressing iNOS and/or Ifny (YFP+)(Fig.7D)" Iba1 is needed in the quantification. Based on the images, iNOS seems to be highly produced in Iba1 negative cells. Which cells do produce it then? Flow cytometry data for this quantification are required. This would allow you to specifically check which cells express the markers and allow for a more precise analysis of double positive cells.

Currently these data demonstrating the co-localization of stress markers phospho-c-Jun and Chac1 with YFP are presented in Fig.7E (images) and Suppl.Fig.13D (quantification). The co-localization of stress markers phospho-cJun and Chac1 with iNOS is presented in Suppl.Fig.13F (images) and Suppl.Fig.13E (quantification). We agree that some iNOS+ cells are Iba1-negative (Fig.7D). We manually quantified percentages of Iba1+iNOS+ double positive cells and demonstrated that they represent the majority of the iNOS+ population(Suppl.Fig.13A). Regarding the required FACS analysis, we focus on spatial approaches because of the heterogeneity of the lesions that would be lost if lungs are dissociated for FACS. We are working on spatial transcriptomics at a single cell resolution that preserves spatial organization of TB lesions to address the reviewer’s comment and will present our results in the future.

- Results part 6: In general, can you please state for each experiment at what time point mice were analyzed? You should include an additional macrophage staining (e.g. MerTK, F4/80), as alveolar macrophages are not staining well for Iba1 and you might therefore miss them in your IF microscopy. It would be very nice if you could perform flow cytometry to really check on the macrophages during infection and distinguish subsets (e.g. alveolar macrophages, interstitial macrophages, monocytes).

We have included the details of time post infection in figure legends for Fig.7, Suppl.Figures 8, 9, 12B, 13, 14A of the revised manuscript. We have performed staining with CD11b, CD206 and CD163 to differentiate the recruited and lung resident macrophages and determined that in chronic pulmonary TB lesions in our model the vast majority of macrophages are recruited CD11b+, but not resident (CD206+ and CD163+) macrophages. These data is presented in another manuscript (Yabaji et al., BioRxiv https://doi.org/10.1101/2023.10.17.562695).

- Spatial sequencing: The manuscript would highly profit from more data on that. It would be very interesting to check for the DEGs and show differential spatial distribution. Expression of marker genes should be inferred to further define macrophage subsets (e.g. alveolar macrophages, interstitial macrophages, recruited macrophages) and see if these subsets behave differently within the same lesion but also between the lesions. Additional bioinformatic approaches might allow you to investigate cell-cell interactions. There is a lot of potential with such a dataset, especially from TB lesions, that would elevate your findings and prove interesting to the TB field.- "Thus, progression from the Mtb-controlling paucibacillary to non-controlling multibacillary TB lesions in the lungs of TB susceptible mice was mechanistically linked with a pathological state of macrophage activation characterized by escalating stress (as evidenced by the upregulation phospho-cJUN, PKR and Chac1), the upregulation of IFNβ and the IFN-I pathway hyperactivity, with a concurrent reduction of IFNγ responses." To really show the upregulation within macrophages and their activation, a more detailed IF microscopy with the inclusion of additional macrophage markers needs to be provided. Flow cytometry would enable analysis for the differences between alveolar and interstitial macrophages, as well as for monocytes. As however, it seems that the majority of iNOS, as well as the stress associated markers are not produced by Iba1+ cells. Analyzing granulocytes and T lymphocytes should be considered.

We appreciate the reviewer’s suggestion. Indeed, our model provides an excellent opportunity to investigate macrophage heterogeneity and cell interactions within chronic TB lesions. We are working on spatial transcriptomics at a single cell resolution that would address the reviewer’s comment and will present our results in the future.

In agreement with classical literature the overwhelming majority of myeloid cells in chronic pulmonary TB lesions is represented by macrophages. Neutrophils are detected at the necrotic stage, but our study is focused on pre-necrotic stages to reveal the earlier mechanisms predisposing to the necrotization. We never observed neutrophils or T cells expressing iNOS in our studies.

- It's mentioned in the method section that controls in the IF staining were only fixed for 10min, while the infected cells were fixed for 30min. Consistency is important as the PFA fixation might impact on the fluorescence signal. Therefore, controls should be repeated with the same fixation time.

We have carefully considered the impact of fixation time on fluorescence and have separately analyzed the non-infected and infected samples to address this concern. For the non-infected samples, we examined the effect of TNF in both B6 and B6.Sst1S backgrounds, ensuring that a consistent fixation protocol (10 min) was applied across all experiments without Mtb infection.

For the Mtb infection experiments, we employed an optimized fixation protocol (30 min) to ensure that Mtb was killed before handling the plates, which is critical for preserving the integrity of the samples. In this context, we compared B6 and B6.Sst1S samples to evaluate the effects of fixation and Mtb infection on lipid peroxidation (LPO) induction.

We believe this approach balances the need for experimental consistency with the specific requirements for handling infected cells, and we have revised the manuscript to reflect this clarification.

- Reactive oxygen species levels should be determined in B6 and B6.Sst1S BMDMs (stimulated and unstimulated), as they are very important for oxidative stress.

We have conducted experiments to measure ROS production in both B6 and B6.Sst1S BMDMs and demonstrated higher levels of ROS in the susceptible BMDMs after prolonged TNF stimulation (new Fig.3I-J and Suppl. Fig. 3G). Additionally, we have previously published a comparison of ROS production between B6 and B6.Sst1S by FACS (PMID: 33301427), which also supports the findings presented here.

- Sup. Fig 2C: The inclusion of an unstimulated control would be advisable in order to evaluate if there are already difference in the beginning.

We have included the untreated control to the Suppl. Fig. 2C (currently Suppl. Fig. 2D) in the revised manuscript.

- Sup. Fig. 3F: Why is the fold change now lower than in Fig. 4D (fold change of around 28 compared to 120 in 4D)?

The data in Fig.4D (Fig.4E in the revised manuscript) and Suppl.Fig.3F (currently Suppl.Fig.4C) represent separate experiments and this variation between experiments is commonly observed in qRT-PCR that is affected by slight variations in the expression in unsimulated controls used for the normalization and the kinetics of the response. This 2-4 fold difference between same treatments in separate experiments, as compared to 30 – 100 fold and higher induction by TNF does not affect the data interpretation.

- Sup. Fig. 5C, D: The data seems very interesting as you even observe an increase in gene expression. Data for the B6 mice should be evaluated for increase to a similar level as the TNF treated mutants. Data on the viability of the cells are necessary, as they no longer receive MCSF and might be dying at this point already.

To ensure that the observed effects were not confounded by cytotoxicity, we determined non-toxic concentrations of the CSF1R inhibitors during 48h of incubation and used them in our experiments that lasted for 24h. To address this valid comment, we have included cell viability data in the revised manuscript to confirm that the treatments did not result in cell death (Suppl. Fig. 6D). This experiment rejected our hypothesis that CSF1 driven Myc expression could be involved in the *Ifnb* superinduction. Other effects of CSF1R inhibitors on type I IFN pathway are intriguing but are beyond the scope of this study.

- Sup. Fig 12: the phospho-c-Jun picture for (P) is not the same as in the merged one with Iba1. Double positive cells are mentioned to be analyzed, but from the staining it appears that P-c-Jun is expressed by other cells. You do not indicate how many replicates were counted and if the P and M lesions were evaluated within the same animal. What does the error bar indicate? It seems unlikely from the plots that the double positive cells are significant. Please provide the p values and statistical analysis.

We thank the reviewer for bringing this inadvertent field replacement in the single phospho-cJun channel to our attention. However, the quantification of Iba1+phospho-cJun+ double positive cells in Suppl.Fig.12 and our conclusions were not affected. In the revised manuscript, images and quantification of phospho-cJun and Iba1 co-expression are shown in new Suppl.Fig.13B and C, respectively. We have also updated the figure legends to denote the number of lesions analyzed and statistical tests. Specifically, lesions from 6–8 mice per group (paucibacillary and multibacillary) were evaluated. Each dot in panels Suppl.Fig.13 represent individual lesions.

- Sup. Fig. 13D (suppl.Fig.15D now): What about the expression of MYC itself? Other parts of the signaling pathway should be analyzed(e.g. IFNb, JNK)?

The difference in *MYC* mRNA expression tended to be higher in TB patients with poor outcomes, but it was not statistically significant after correction for multiple testing. The upregulation of Myc pathway in the blood transcriptome associated with TB treatment failure most likely reflects greater proportion of immature cells in peripheral blood, possibly due to increased myelopoiesis. Pathway analysis of the differentially expressed genes revealed that treatment failures were associated with the following pathways relevant to this study: NF-kB Signaling, Flt3 Signaling in Hematopoietic Progenitor Cells (indicative of common myeloid progenitor cell proliferation), SAPK/JNK Signaling and Senescence (possibly indicative of oxidative stress). The upregulation of these pathways in human patients with poor TB treatment outcomes correlates with our findings in TB susceptible mice.

- In the mfIHC you he usage of anti-mouse antibodies is mentioned. Pictures of sections incubated with the secondary antibody alone are required to exclude the possibility that the staining is not specific. Especially, as this data is essential to the manuscript and mouse-antimouse antibodies are notorious for background noise.

We are well aware of the technical difficulties associated with using mouse on mouse staining. In those cases, we use rabbit anti-mouse isotype specific antibodies specifically developed to avoid non-specific background (Abcam cat#ab133469). Each antibody panel for fluorescent multiplexed IHC is carefully optimized prior to studies. We did not use any primary mouse antibodies in the final version of the manuscript and, hence, removed this mention from the Methods.

- In order to tie the story together, it would be interesting to treat infected mice with an INFAR antibody, as well as perform this experiment with a Myc antibody. According to your data, you might expect the survival of the mice to be increased or bacterial loads to be affected.

In collaboration with the Vance laboratory, we tested effects of type I IFN pathway inhibition in B6.Sst1S mice on TB susceptibility: either type I receptor knockout or blocking antibodies increased their resistance to virulent Mtb (published in Ji et al., 2019; PMID 31611644). Unfortunately, blocking Myc using neutralizing antibodies in vivo is not currently achievable. Specifically blocking Myc using small molecule inhibitors in vivo is notoriously difficult, as recognized in oncology literature. We consider using small molecule inhibitors of either Myc translation or specific pathways downstream of Myc in the future.

- It is surprising that you not even once cite or mention your previous study on bioRxiv considering the similarity of the results and topic (https://doi.org/10.1101/2020.12.14.422743). Is not even your Figure 1I and Figure 2 J, K the same as in that study depicted in Figure 4?

The reviewer refers to the first version of this manuscript uploaded to BioRxiv, but it has never been published. We continued this work and greatly expanded our original observations, as presented in the current manuscript. Therefore, we do not consider the previous version as an independent manuscript and, therefore, do not cite it.

- Please revise spelling of the manuscript and pay attention to write gene names in italics

Thank you, we corrected the gene and protein names according to current nomenclature.

Minor points:- Fig. 1: Please provide some DEGs that explain why you used this resolution for the clustering of the scRNAseq data and that these clusters are truly distinct from each other.

Differential gene expression in clusters is presented in Suppl.Fig.1C (interferon response) and Suppl.Fig.1D (stress markers and interferon response previously established in our studies).

- Fig. 1F: What do the two lines represent (magenta, green)?

The lines indicate pseudotime trajectories of B6 (magenta) and B6.Sst1S (green) BMDMs.

- Fig. 1F, G: Why was cluster 6 excluded?

This cluster was not different between B6 and B6.Sst1S, so it was not useful for drawing the strain-specific trajectories.

- Fig. 1E, G, H: The intensity scales are missing. They are vital to understand the data.

We have included the scale in revised manuscript (Fig.1E,G,H and Suppl.Fig.1C-D).

- Fig. 2G-I: please revise order, as you first refer to Fig. 2H and I

We revised the panels’ order accordingly

- Fig. 5: You say the data represents three samples but at least in D and E you have more. Please revise. Why do you only include at (G) the inhibitor only control?

We added the inhibitor only controls to Fig. 5D - H. We also indicated the number of replicates in the updated Fig.5 legend.

- Figure 7A, Sup. Fig. 8: Are these maximum intensity projection? Or is one z-level from the 3D stack depicted?

The Fig. 7A shows 3D images with all the stacks combined.

- Fig. 7B: What do the white boxes indicate?

We have removed this panel in the revised version and replaced it with better images.

- Sup. Fig. 1A: The legend for the staining is missing

The Suppl. Fig.1A shows the relative proportions of either naïve (R and S) or TNFstimulated (RT and ST) B6 or B6.Sst1S macrophages within individual single cell clusters depicted in Fig.1B. The color code is shown next to the graph on the right.

- Sup. Fig. 1B: The feature plots are not clear: The legend for the expression levels is missing. What does the heading means?

We updated the headings, as in Fig.1C. The dots represent individual cells expressing *Sp110* mRNA (upper panels) and *Sp140* mRNA (lower panels).

- Sup. Fig. 3C: The scale bar is barely visible.

We resized the scale bar to make it visible and presented in Suppl. Fig.3E (previously Suppl. Fig.3C).

- Sup. Fig. 3D: There is not figure legend or the legend to C-E is wrong.- Sup. Fig. 3F, G: You do not state to what the data is relative to.

We identified an error in the Suppl.Fig.3 legend referring to specific panels. The Suppl.Fig.3 legend has been updated accordingly. New panels were added and Suppl.Fig.3-G panels are now Suppl.Fig.4C-D.

- Sup. Fig. 3H: It seems you used a two-way ANOVA, yet state it differently. Please revise the figure legend, as Dunnett's multiple comparison would only check for significances compared to the control.

Following the reviewer’s comment, we repeated statistical analysis to include correction for multiple comparisons and revised the figure and legend accordingly.

- Sup. Fig. 4A, B: It is not clear what the lines depict as the legend is not explained. Names that are not required should be changed to make it clear what is depicted (e.g. "TE@" what does this refer to?)

This previous Sup. Fig 4 is now Sup. Fig. 5. The “TE@” is a leftover label from the bioinformatics pipeline, referring to “Transposable Element”. We apologize for this confusion and have removed these extraneous labels. We have also added transposon names of the LTR (MMLV30 and RTLV4) and L1Md to Suppl.Fig.5A and 5B legend, respectively.

- Sup. 4B: What does the y-scale on the right refer to?

We apologize for the missing label for the y-scale on the right which represents the mRNA expression level for the SetDB1 gene, which has a much lower steady state level than the LINE L1Md, so we plotted two Y-scales to allow both the gene and transposon to be visualized on this graph.

- Sup. 4C: Interpretation of the data is highly hindered by the fact that the scales differ between the B6 and B6.Sst1. The scales are barely visible.

We apologize for the missing labels for the y-scales of these coverage plots, which were originally meant to just show a qualitative picture of the small RNA sequencing that was already quantitated by the total amounts in Sup. 4B. We have added thee auto-scaled Y-scales to Sup. 4C and improved the presentation of this figure.

- Sup. Fig. 5A, B: Is the legend correct? Did you add the antibody for 2 days or is the quantification from day 3?

We recognize that the reviewer refers to Suppl.Fig.6A-B (Suppl.Fig.7A-B in the revised manuscript). We did not add antibodies to live cells. The figure legend describes staining with 4HNE-specific antibodies 3 days post Mtb infection.

- Sup. Fig. 8A: Are the "early" and "intermediate" lesions from the same time points? What are the definitions for these stages?

We discussed our lesion classification according to histopathology and bacterial loads above. Of note, in the revised manuscript we simplified our classification to denote paucibacillary and multibacillary lesions only. We agree with reviewers that designation lesions as early, intermediate and advanced lesions were based on our assumptions regarding the time course of their progression from low to high bacterial loads.

- Sup. Fig. 8E: You should state that the bottom picture is an enlargement of an area in the top one. Scale bars are missing.

We replaced this panel with clearer images in Suppl.Fig.12B.

- Sup. Fig. 11A: The IF staining is only visible for Iba and iNOS. Please provide single channels in order to make the other staining visible.

Suppl.Fig.11A (now Suppl.Fig.13B) shows the low-magnification images of TB lesions. In the Fig. 7 and Suppl. Fig. 13F of the revised manuscript we provided images for individual markers.

- Sup. Fig. 13A (Suppl.Fig.15A now): Your axis label is not clear. What do the numbers behind the genes indicate? Why did you choose oncogene signatures and not inflammatory markers to check for a correlation with disease outcome?

X axis of Suppl.Fig.15A represent pre-defined molecular signature gene sets MSigDB in Gene Set Enrichment Analysis (GSEA) database (https://www.gseamsigdb.org/gsea/msigdb). On Y axis is area under curve (AUC) score for each gene set.

- Sup. 13D(Suppl.Fig.15D now): Maybe you could reorder the patients, so that the impression is clearer, as right now only the top genes seem to show a diverging gene signature, while the rest gives the impression of an equal distribution.

The Myc upregulated gene set myc_up was identified among top gene sets associated with treatment failure using unbiased ssGSEA algorithm. We agree with the reviewer that not every gene in the myc_up gene set correlates with the treatment outcome. But the association of the gene set is statistically significant, as presented in Suppl.Fig.15B – C.

- The scale bars for many microscopy pictures are missing.

We have included clearly visible scale bars to all the microscopy images in the revised version.

- The black bar plots should be changed (e.g. in color), since the single data points cannot be seen otherwise.- It would be advisable that a consistent color scheme would be used throughout the manuscript to make it easier to identify similar conditions, as otherwise many different colours are not required and lead right now rather to confusion (e.g. sometimes a black bar refers to BMDMs with and sometimes without TNF stimulation, or B6 BMDMs). Furthermore, plot sizes and fonts should be consistent within the manuscript (including the supplemental data)

We followed this useful suggestion and selected consistent color codes for B6 and B6.Sst1S groups to enhance clarity throughout the revised manuscript.

Within the methods section:- At which concentration did you use the IFNAR antibody and the isotype?

We updated method section by including respective concentrations in the revised manuscript.

- Were mice maintained under SPF conditions? At what age where they used?

Yes, the mice are specific pathogen free. We used 10 - 14 week old mice for Mtb infection.

- The BMDM cultivation is not clear. According to your cited paper you use LCCM but can you provide how much M-CSF it contains? How do you make sure that amounts are the same between experiments and do not vary? You do not mention how you actually obtain this conditioned medium. Is there the possibility of contamination or transferred fibroblasts that would impact on the data analysis? Is LCCM also added during stimulation and inhibitor treatment?

We obtain LCCM by collecting the supernatant from L929 cell line that form confluent monolayer according to well-established protocols for LCCM collection. The supernatants are filtered through 0.22 micron filters to exclude contamination with L929 cells and bacteria. The medium is prepared in 500 ml batches that are sufficient for multiples experiments. Each batch of L929-conditioned medium is tested for biological activity using serial dilutions.

- How was the BCG infection performed? How much bacteria did you use? Which BCG strain was used?

We infected mice with *M. bovis* BCG Pasteur subcutaneously in the hock using 10^6^ CFU per mouse.

- At what density did you seed the BMDMs for stimulation and inhibitor experiments?

In 96 well plates, we seed 12,000 cells per well and allow the cells to grow for 4 days to reach confluency (approximately 50,000 cells per well). For a 6-well plate, we seed 2.5 × 10^5^ cells per well and culture them for 4 days to reach confluency. For a 24-well plate, we seed 50,000 cells per well and keep the cells in media for 4 days before starting any treatments. This ensures that the cells are in a proliferative or near-confluent state before beginning the stimulation or inhibitor treatments. Our detailed protocol is published in STAR Protocols (Yabaji et al., 2022; PMID 35310069).

- What machine did you use to perform the bulk RNA sequencing? How many replicates did you include for the sequencing?

For bulk sequencing we used 3 RNA samples for each condition. The samples were sequenced at Boston University Microarray & Sequencing Resource service using Illumina NextSeq 2000 instrument.

- How many replicates were used for the scRNA sequencing? Why is your threshold for the exclusion of mitochondrial DNA so high? A typical threshold of less than 5% has been reported to work well with mouse tissue.

We used one sample per condition. For the mitochondrial cutoff, we usually base it off of the total distribution. There is no "universal" threshold that can be applied to all datasets. Thresholds must be determined empirically.

- You do not mention how many PCAs were considered for the scRNA sequencing analysis.

We considered 50 PCAs, this information was added to Methods

- You should name all the package versions you used for the scRNA sequencing (e.g. for the slingshot, VAM package)

The following package versions were used: Seurat v4.0.4, VAM v1.0.0, Slingshot v2.3.0, SingleCellTK v2.4.1, Celda v1.10.0, we added this information to Methods.

- You mention two batches for the human samples. Can you specify what the two batches are?

Human blood samples were collected at five sites, as described in the updated Methods section and two RNAseq batches were processed separately that required batch correction.

- At which temperature was the IF staining performed?

We performed the IF at 4oC. We included the details in revised version.

**Reviewer #2 (Significance):**
Overall, the manuscript has interesting findings with regard to macrophage responses in Mycobacteria tuberculosis infection.

However, in its current form there are several shortcomings, both with respect to the precision of the experiments and conclusions drawn.

**Reviewer #3 (Evidence, reproducibility and clarity):**
SummaryThe authors use a mouse model designed to be more susceptible to M.tb (addition of sst1 locus) which has granulomatous lesions more similar to human granulomas, making this mouse highly relevant for M.tb pathogenesis studies. Using WT B6 macrophages or sst1B6 macrophages, the authors seek to understand the how the sst1 locus affects macrophage response to prolonged TNFa exposure, which can occur during a pro-inflammatory response in the lungs. Using single cell RNA-seq, revealed clusters of mutant macrophages with upregulated genes associated with oxidative stress responses and IFN-I signaling pathways when treated with TNF compared to WT macs. The authors go on to show that mutant macrophages have decreased NRF2, decreased antioxidant defense genes and less Sp110 and Sp140. Mutant macrophages are also more susceptible to lipid peroxidation and ironmediated oxidative stress. The IFN-I pathway hyperactivity is caused by the dysregulation of iron storage and antioxidant defense. These mutant macrophages are more susceptible to M.tb infection, showing they are less able to control bacterial growth even in the presence of T cells from BCG vaccinated mice. The transcription factor Myc is more highly expressed in mutant macs during TNF treatment and inhibition Myc led to better control of M.tb growth. Myc is also more abundant in PBMCs from M.tb infected humans with poor outcomes, suggesting that Myc should be further investigated as a target for host-directed therapies for tuberculosis.Major CommentsIsotypes for IF imaging and confocal IF imaging are not listed, or not performed. It is a concern that the microscopy images throughout the manuscript do not have isotype controls for the primary antibodies.Fig 4 (and later) the anti-IFNAR Ab is used along with the Isotype antibody, Fig 4I does not show the isotype. Use of the isotype antibody is also missing in later figures as well as Fig 3J. Why was this left off as the proper control for the Ab?

We addressed the comment in revised manuscript as described above in summary and responses to reviewers 1 and 2. Isotype controls for IFNAR1 blockade were included in Fig.3M (previously 3J), Fig. 4I, Suppl.Fig.4G (previously Fig.4I), and updated Fig.4C-E, Fig.6L-M, Suppl.Fig.4F-G, 7I.

Conclusions drawn by the authors from some of the WB data are worded strongly, yet by eye the blots don't look as dramatically different as suggested. It would be very helpful to quantify the density of bands when making conclusions. (for example, Fig 4A).

We added the densitometry of Western blot values after normalization above each lane in Fig.2A-C, Fig.3C-D and 3K; Fig.4A-B, Fig.5B,C,I,J.

Fig 5A is not described clearly. If the gene expression is normalized to untreated B6 macs, then the level of untreated B6 macs should be 1. In the graph the blue bars are slightly below 1, which would not suggest that levels "initially increased and subsequently downregulated" as stated in the text. It seems like the text describes the protein expression but not the RNA expression. Please check this section and more clearly describe the results.

We appreciate the reviewer’s comment and modified the text to specify the mRNA and protein expression data, as follows:

“We observed that Myc was regulated in an *sst1*-dependent manner: in TNF-stimulated B6 wild type BMDMs, *c-Myc* mRNA was downregulated, while in the susceptible macrophages c-*Myc* mRNA was upregulated (Fig.5A). The c-Myc protein levels were also higher in the B6.Sst1S cells in unstimulated BMDMs and 6 – 12 h of TNF stimulation (Fig.5B)”.

Also, why look at RNA through 24h but protein only through 12h? If c-myc transcripts continue to increase through 24h, it would be interesting to see if protein levels also increase at this later time point.

The time-course of Myc expression up to 24 h is presented in new panels Fig. 5I-5J It demonstrates the decrease of Myc protein levels at 24 h. In the wild type B6 BMDMs the levels of Myc protein significantly decreased in parallel with the mRNA suppression presented in Fig.5A. In contrast , we observed the dissociation of the mRNA and protein levels in the _sst1_mutant BMDMs at 12 and 24 h, most likely, because the mutant macrophages develop integrated stress response (as shown in our previous publication by Bhattacharya et al., JCI, 2021) that is known to inhibit Myc mRNA translation.

Fig 5J the bands look smaller after D-JNK1 treatment at 6 and 12h though in the text is says no change. Quantifying the bands here would be helpful to see if there really is no difference.

This experiment was repeated twice, and the average normalized densitometry values are presented in the updated Fig.5J. The main question addressed in this experiment was whether the hyperactivity of JNK in TNF-stimulated *sst1* mutant macrophages contributed to Myc upregulation, as was previously shown in cancer. Comparing effects of JNK inhibition on phospho-cJun and c-Myc protein levels in TNF stimulated B6.Sst1S macrophages (updated Fig.5J), we concluded that JNK did not have a major role in c-Myc upregulation in this context.

Section 4, third paragraph, the conclusion that JNK activation in mutant macs drives pathways downstream of Myc are not supported here. Are there data or other literature from the lab that supports this claim?

This statement was based on evidence from available literature where JNK was shown to activate oncogens, including Myc. In addition, inhibition of Myc in our model upregulated ferritin (Fig.Fig.5C), reduced the labile iron pool, prevented the LPO accumulation (Fig.5D - G) and inhibited stress markers (Fig.5H). However, we do not have direct experimental evidence in our model that Myc inhibition reduces ASK1 and JNK activities. Hence, we removed this statement from the text and plan to investigate this in the future.

Fig 6N Please provide further rationale for the BCG in vivo experiment. It is unclear what the hypothesis was for this experiment.

In the current version BCG vaccination data is presented in Suppl.Fig.14B. We demonstrate that stressed BMDMs do not respond to activation by BCG-specific T cells (Fig.6J) and their unresponsiveness is mediated by type I interferon (Fig.6L and 6M). The observed accumulation of the stressed macrophages in pulmonary TB lesions of the *sst1*-susceptible mice (Fig.7E, Suppl.Fig.13 and 14A) and the upregulation of type I interferon pathway (Fig.1E,1G, 7C), (Suppl.Fig.1C and 11) suggested that the effect of further boosting T lymphocytes using BCG in Mtb-infected mice will be neutralized due to the macrophage unresponsiveness. This experiment provides a novel insight explaining why BCG vaccine may not be efficient against pulmonary TB in susceptible hosts.

The in vitro work is all concerning treatment with TNFa and how this exposure modifies the responses in B6 vs sst1B6 macrophages; however, this is not explored in the in vivo studies. Are there differences in TNFa levels in the pauci- vs multi-bacillary lesions that lead to (or correlate with) the accumulation of peroxidation products in the intralesional macrophages. How to the experiments with TNFa in vitro relate back to how the macrophages are responding in vivo during infection?

Our investigation of mechanisms of necrosis of TB granulomas stems from and supported by in vivo studies as summarized below.

This work started with the characterization necrotic TB granulomas in C3HeB/FeJ mice in vivo followed by a classical forward genetic analysis of susceptibility to virulent Mtb in vivo.

That led to the discovery of the *sst1* locus and demonstration that it plays a dominant role in the formation of necrotic TB granulomas in mouse lungs in vivo. Using genetic and immunological approaches we demonstrated that the *sst1* susceptibility allele controls macrophage function in vivo (Yan, et al., J.Immunol. 2007) and an aberrant macrophage activation by TNF and increased production of Ifn-b in vitro (He et al. Plos Pathogens, 2013). In collaboration with the Vance lab we demonstrated that the type I IFN receptor inactivation reduced the susceptibility to intracellular bacteria of the *sst1*-susceptible mice in vivo (Ji et al., Nature Microbiology, 2019). Next, we demonstrated that the *Ifnb1* mRNA superinduction results from combined effects of TNF and JNK leading to integrated stress response in vitro (Bhattacharya, JCI, 2021). Thus, our previous work started with extensive characterization of the in vivo phenotype that led to the identification of the underlying macrophage deficiency that allowed for the detailed characterization of the macrophage phenotype in vitro presented in this manuscript. In a separate study, the Sher lab confirmed our conclusions and their in vivo relevance using *Bach1* knockout in the *sst1*-susceptible B6.Sst1S background, where boosting antioxidant defense by Bach1 inactivation resulted in decreased type I interferon pathway activity and reduced granuloma necrosis. We have chosen TNF stimulation for our in vitro studies because this cytokine is most relevant for the formation and maintenance of the integrity of TB granulomas in vivo as shown in mice, non-human primates and humans. Here we demonstrate that although TNF is necessary for host resistance to virulent Mtb, its activity is insufficient for full protection of the susceptible hosts, because of altered macrophages responsiveness to TNF. Thus, our exploration of the necrosis of TB granulomas encompass both in vitro and extensive in vivo studies.

Minor commentsIntroduction, while well written, is longer than necessary. Consider shortening this section. Throughout figures, many graphs show a fold induction/accumulation/etc, but it is rarely specified what the internal control is for each graph. This needs to be added.Paragraph one, authors use the phrase "the entire IFN pathway was dramatically upregulated..." seems to be an exaggeration. How do you know the "entire" IFN pathway was upregulated in a dramatic fashion?

(1) We shortened the introduction and discussion; (2) verified that figure legends internal controls that were used to calculate fold induction; (3) removed the word “entire” to avoid overinterpretation.

Figures 1E, G and H and supp fig 1C, the heat maps are missing an expression key Section 2 second paragraph refers to figs 2D, E as cytoplasmic in the text, but figure legend and y-axis of 2E show total protein.

The expression keys were added to Fig.1E,G,H, Fig.7C, Suppl.Fig.1C and 1D and Suppl.Fig.11A of the revised manuscript.

Section 3 end of paragraph 1 refers to Fig 3h. Does this also refer to Supp Fig 3E?

Yes, Fig.3H shows microscopy of 4-HNE and Suppl.Fig.3H shows quantification of the image analysis. In the revised manuscript these data are presented in Fig.3H and Suppl.Fig.3F. The text was modified to reflect this change.

Supplemental Fig 3 legend for C-E seems to incorrectly also reference F and G.

We corrected this error in the figure legend. New panels were added to Suppl.Fig.3 and previous Suppl.Fig.3F and G were moved to Suppl.Fig.4 panels C and D of the revise version.

Fig 3K, the p-cJun was inhibited with the JNK inhibitor, however it’s unclear why this was done or the conclusion drawn from this experiment. Use of the JNK inhibitor is not discussed in the text.

The JNK inhibitor was used to confirm that c-Jun phosphorylation in our studies is mediated by JNK and to compare effects of JNK inhibition on phospho-cJun and Myc expression. This experiment demonstrated that the JNK inhibitor effectively inhibited c-Jun phosphorylation but not Myc upregulation, as shown in Fig.5I-J of the revised manuscript.

Fig 4 I and Supp Fig 3 H seem to have been swapped? The graph in Fig 4I matches the images in Supp Fig 3I. Please check.

We reorganized the panels to provide microscopy images and corresponding quantification together in the revised the panels Fig. 4H and Fig. 4I, as well as in Suppl. Fig. 4F and Suppl. Fig. 4G.

Fig 6, it is unclear what % cell number means. Also for bacterial growth, the data are fold change compared to what internal control?

We updated Fig.6 legend to indicate that the cell number percentages were calculated based on the number of cells at Day 0 (immediately after Mtb infection). We routinely use fixable cell death staining to enumerate cell death. Brief protocol containing this information is included in Methods section. The detailed protocol including normalization using BCG spike has been published – Yabaji et al, STAR Protocols, 2022. Here we did not present dead cell percentage as it remained low and we did not observe damage to macrophage monolayers. This allows us to exclude artifacts due to cell loss. The fold change of Mtb was calculated after normalization using Mtb load at Day 0 after infection and washes.

Fig 7B needs an expression key

The expression keys was added to Fig.7C (previously Fig. 7B).

Supp Fig 7 and Supp Fig 8A, what do the arrows indicate?

In Suppl.Fig.8 (previously Suppl.Fig.7) the arrows indicate acid fast bacilli (Mtb). In figures Fig.7A and Suppl.Fig.9A arrows indicate Mtb expressing fluorescent reporter mCherry. Corresponding figure legends were updated in the revised version.

Supp Fig 9A, two ROI appear to be outlined in white, not just 1 as the legend says Methods:

We updated the figure legend.

Certain items are listed in the Reagents section that are not used in the manuscript, such as necrostatin-1 or Z-VAD-FMK. Please carefully check the methods to ensure extra items or missing items does not occur.

These experiments were performed, but not included in the final manuscript. Hence, we removed the “necrostatin-1 or Z-VAD-FMK” from the reagents section in methods of revised version.

Western blot, method of visualizing/imaging bands is not provided, method of quantifying density is not provided, though this was done for fig 5C and should be performed for the other WBs.

We used GE ImageQuant LAS4000 Multi-Mode Imager to acquire the Western blot images and the densitometric analyses were performed by area quantification using ImageJ. We included this information in the method section. We added the densitometry of Western blot values after normalization above each lane in Fig.2A-C, Fig.3C-D and 3K; Fig.4A-B, Fig.5B,C,I,J.

**Reviewer #3 (Significance):**
The work of Yabaji et al is of high significance to the field of macrophage biology and M.tb pathogenesis in macrophages. This work builds from previously published work (Bhattacharya 2021) in which the authors first identified the aberrant response induced by TNF in sst1 mutant macrophages. Better understanding how macrophages with the sst1 locus respond not only to bacterial infection but stimulation with relevant ligands such as TNF will aid the field in identifying biomarkers for TB, biomarkers that can suggest a poor outcome vs. "cure" in response to antibiotic treatment or design of host-directed therapies.This work will be of interest to those who study macrophage biology and who study M.tb pathogenesis and tuberculosis in particular. This study expands the knowledge already gained on the sst1 locus to further determine how early macrophage responses are shaped that can ultimately determine disease progression.Strengths of the study include the methodologies, employing both bulk and single cell-RNA seq to answer specific questions. Data are analyze using automated methods (such as HALO) to eliminated bias. The experiments are well planned and designed to determine the mechanisms behind the increased iron-related oxidative stress found in the mutant macrophages following TNF treatment. Also, in vivo studies were performed to validate some of the in vitro work. Examining pauci-bacillary lesions vs multi-bacillary lesions and spatial transcriptomics is a significant strength of this work. The inclusion of human data is another strength of the study, showing increased Myc in humans with poor response to antibiotics for TB.Limitations include the fact that the work is all done with BMDMs. Use of alveolar macrophages from the mice would be a more relevant cell type for M.tb studies. AMs are less inflammatory, therefore treatment with TNF of AMs could result in different results compared to BMDMs. Reviewer's field of expertise: macrophage activation, M.tb pathogenesis in human and mouse models, cell signaling.Limitations: not qualified to evaluate single cell or bulk RNA-seq technical analysis/methodology or spatial transcriptomics analysis.